# Exploration from a Primal-Dual Lens: Value-Incentivized Actor-Critic Methods for Sample-Efficient Online RL

**Tong Yang**[*]
CMU

**Bo Dai**[†]
Georgia Tech

**Lin Xiao**[‡]
Meta

**Yuejie Chi**[§]
Meta & Yale

## Abstract

Online reinforcement learning (RL) with complex function approximations such as transformers and deep neural networks plays a significant role in the modern practice of artificial intelligence. Despite its popularity and importance, balancing the fundamental trade-off between exploration and exploitation remains a long-standing challenge; in particular, we are still in lack of efficient and practical schemes that are backed by theoretical performance guarantees. Motivated by recent developments in exploration via optimistic regularization, this paper provides an interpretation of the principle of optimism through the lens of primal-dual optimization. From this fresh perspective, we set forth a new value-incentivized actor-critic (VAC) method, which optimizes a single easy-to-optimize objective integrating exploration and exploitation — it promotes state-action and policy estimates that are both consistent with collected data transitions and result in higher value functions. Theoretically, the proposed VAC method has near-optimal regret guarantees under linear Markov decision processes (MDPs) in both finite-horizon and infinite-horizon settings, which can be extended to the general function approximation setting under appropriate assumptions.

## 1 Introduction

In online reinforcement learning (RL) [Sutton et al., 1998], an agent learns to update their policy in an adaptive manner while interacting with an unknown environment to maximize long-term cumulative rewards. In conjunction with complex function approximation such as large neural networks and foundation models to reduce dimensionality, online RL has achieved remarkable performance in a wide variety of applications such as game playing [Silver et al., 2017], control [Mnih et al., 2015], language model post-training [OpenAI, 2023, Team et al., 2023] and reasoning [Guo et al., 2025], and many others.

Despite its popularity, advancing beyond current successes is severely bottlenecked by the cost and constraints associated with data collection. While simulators can subsidize data acquisition in certain domains, many real-world applications—such as clinical trials, recommendation systems and autonomous driving—operate under conditions where gathering interaction data is expensive, time-consuming or potentially risky. In these high-stake scenarios, managing the fundamental yet delicate trade-off between exploration (gathering new information about the environment) and exploitation (leveraging existing knowledge to maximize rewards) requires paramount care. Naive exploration schemes, such as the $\epsilon$-greedy method, are known to be sample-inefficient as they explore randomly

---

[*]Carnegie Mellon University; Emails: `tongyang@andrew.cmu.edu`.

[†]Georgia Institute of Technology; Email: `bodai@cc.gatech.edu`.

[‡]Fundamental AI Research, Meta; Email: `linx@meta.com`.

[§]Yale University; Emails: `yuejie.chi@yale.edu`.

39th Conference on Neural Information Processing Systems (NeurIPS 2025).

without strategic information gathering [Dann et al., 2022]. Arguably, it is still an open challenge to develop **practical** online RL algorithms that come with **provable** sample-efficiency guarantees, especially in the presence of function approximation.

Addressing this limitation, significant research attempts have been made to develop statistically efficient approaches, often guided by the principle of optimism in the face of uncertainty [Lattimore and Szepesvári, 2020]. Prominent approaches include constructing optimistic estimates with data-driven confidence sets [Auer et al., 2008, Agarwal et al., 2023, Chen et al., 2025, Foster et al., 2021], as well as employing Bayesian methods like Thompson sampling [Russo et al., 2018] and its optimistic variants [Agrawal and Jia, 2017, Zhang, 2022]. While appealing theoretically, translating them into practical algorithms compatible with general function approximators often proves difficult. Many such theoretically-grounded approaches either suffer from prohibitive computational complexity or exhibit underwhelming empirical performance when scaled to complex problems.

Recently, Liu et al. [2024] introduced an intriguing framework termed Maximize to Explore (MEX) for online RL, which optimizes a single objective function over the state-action value function (i.e., $Q$-function), elegantly unifying estimation, planning and exploration in one framework. In addition, MEX comes with appealing sub-linear regret guarantees under function approximation. However, the practical optimization of the MEX objective presents significant challenges due to its inherent bi-level structure. Specifically, it incorporates the optimal value function derived from the target $Q$-function as a regularizer [Kumar and Becker, 1982], which is not directly amenable to first-order optimization toolkits. As a result, nontrivial modifications are introduced in the said implementation of MEX, making it challenging to ablate the benefit of the MEX framework. This practical hurdle raises a crucial question:

*Can we design a sample-efficient model-free online RL algorithm that optimizes a unifying objective function, but without resorting to complex bilevel optimization?*

## 1.1 Our contribution

In this paper, we answer this question in the affirmative, introducing a novel actor-critic method that achieves near-optimal regret guarantees by optimizing a single non-bilevel objective. Our contributions are summarized as follows.

- *Incentivizing exploration from the primal-dual perspective.* We start by offering a new interpretation of MEX, where optimistic regularization—central to MEX—arises naturally from a Lagrangian formulation within a primal-dual optimization perspective [Dai et al., 2018, Nachum and Dai, 2020]. Specifically, we demonstrate that the seemingly complex MEX objective function can be derived as the regularized Lagrangian of a canonical value maximization problem, subject to the constraint that the $Q$-function satisfies the *Bellman optimality equation*. This viewpoint allows deeper understanding of the structure of the MEX objective and its exploration mechanism.

- *VAC: Value-incentivized actor-critic method.* Motivated by this Lagrangian interpretation, we develop the value-incentivized actor-critic (VAC) method for online RL, which jointly optimizes the $Q$-function and the policy under function approximation over a single objective function. Different from MEX, VAC optimizes a regularized Lagrangian constructed with respect to the *Bellman consistency equation* as the constraint, naturally accommodating the interplay between the $Q$-function and the policy. This formulation preserves the crux of optimistic regularization, while allowing differentiable optimization of the $Q$-function and the policy simultaneously under general function approximation.

- *Theoretical guarantees of VAC.* We substantiate the efficacy of VAC with rigorous theoretical analysis, by proving it achieves a rate of $\widetilde{O}(dH^2\sqrt{T})$ regret under the setting of episodic linear Markov decision processes (MDPs) [Jin et al., 2020], where $d$ is the feature dimension, $H$ is the horizon length, and $T$ is the number of episodes. We further extend the analysis to the infinite-horizon discounted setting and the general function approximation setting under similar assumptions of prior art [Liu et al., 2024].

In summary, our work bridges the gap between theoretically efficient exploration principles and practical applicability in challenging online RL settings with function approximation.

## 1.2 Related work

We discuss a few lines of research that are closely related to our setting, focusing on those with theoretical guarantees under function approximation.

**Regret bounds for online RL under function approximation.** Balancing the exploration-exploitation trade-off is of fundamental importance in the design of online RL algorithms. Most existing methods with provable guarantees rely on the construction of confidence sets and perform constrained optimization within the confident sets, including model-based [Wang et al., 2025, Foster et al., 2023b, Chen et al., 2025], value-based [Agarwal et al., 2023, Jin et al., 2021, Xie et al., 2023], policy optimization [Liu et al., 2023], and actor-critic [Tan et al., 2025] approaches, to name a few. Regret guarantees for approaches based on posterior sampling [Osband and Van Roy, 2017] are provided in [Zhong et al., 2022, Li and Luo, 2024, Agarwal and Zhang, 2022] under function approximation. Regret analysis under the linear MDP model [Jin et al., 2020] has also been actively established for various methods, e.g., for the episodic setting [Zanette et al., 2020, Jin et al., 2020, Papini et al., 2021] and for the infinite-horizon setting [Zhou et al., 2021, Moulin et al., 2025]. However, the confident sets computation and posterior estimation are usually intractable with general function approximator, making the algorithm difficult to be applied.

**Exploration via optimistic estimation.** Exploration via optimistic estimation has been actively studied recently due to its promise in practice, which has been examined over a wide range of settings such as bandits [Kumar and Becker, 1982, Liu et al., 2020, Hung et al., 2021], RL with human feedback [Cen et al., 2024, Xie et al., 2024, Zhang et al., 2024], single-agent RL [Mete et al., 2021, Liu et al., 2024, Chen et al., 2025], and Markov games [Foster et al., 2023a, Xiong et al., 2024, Yang et al., 2025]. Tailored to online RL, most of the optimistic estimation algorithms are model-based, with a few exceptions such as the model-free variant of MEX in [Liu et al., 2020], but still with computationally challenges.

**Primal-dual optimization in RL.** Primal-dual formulation has been exploited in RL for handling the "double-sampling" issue [Dai et al., 2017, 2018] from an optimization perspective. By connecting through the linear programming view of MDP [De Farias and Van Roy, 2004, Puterman, 2014, Wang, 2017, Neu et al., 2017, Lakshminarayanan et al., 2017, Bas-Serrano et al., 2021], a systematic framework [Nachum et al., 2019b] has been developed for offline RL, which induces concrete algorithms for off-policy evaluation [Nachum et al., 2019a, Uehara et al., 2020, Yang et al., 2020], confidence interval evaluation [Dai et al., 2020], imitation learning [Kostrikov et al., 2019, Zhu et al., 2020, Ma et al., 2022, Sikchi et al., 2023], and policy optimization [Nachum et al., 2019b, Lee et al., 2021]. However, how to exploit the primal-dual formulation in online RL setting has not been investigated formally to the best of our knowledge.

**Paper organization and notation.** The rest of this paper is organized as follows. We describe the background, and illuminate the connection between exploration and primal-dual optimization in Section 2. We present the proposed VAC method, and state its regret guarantee in Section 3. Section 4 provide numerical experiments to corroborate the effectiveness of the proposed method. Finally, we conclude in Section 5. The proofs and generalizations to the infinite-horizon and general function approximation settings are deferred to the appendix.

**Notation.** Let $\Delta(\mathcal{A})$ be the probability simplex over the set $\mathcal{A}$, and $[n]$ denote the set $\{1, \ldots, n\}$. For any $x \in \mathbb{R}^n$, we let $\|x\|_p$ denote the $\ell_p$ norm of $x$, where $p \in [1, \infty]$. The $d$-dimensional $\ell_2$ ball of radius $R$ is denoted by $\mathbb{B}_2^d(R)$, and the $d \times d$ identity matrix is denoted by $I_d$.

## 2 Background and Motivation

### 2.1 Background

**Episodic Markov decision processes.** Let $\mathcal{M} = (\mathcal{S}, \mathcal{A}, P, r, H)$ be a finite-horizon episodic MDP, where $\mathcal{S}$ and $\mathcal{A}$ denote the state space and the action space, respectively, $H \in \mathbb{N}^+$ is the horizon length, and $P = \{P_h\}_{h \in [H]}$ and $r = \{r_h\}_{h \in [H]}$ are the inhomogeneous transition kernel and the reward function: for each time step $h \in [H]$, $P_h : \mathcal{S} \times \mathcal{A} \mapsto \Delta(\mathcal{S})$ specifies the probability

distribution over the next state given the current state and action at step $h$, and $r_h : \mathcal{S} \times \mathcal{A} \mapsto [0, 1]$ is the reward function at step $h$. We let $\pi = \{\pi_h\}_{h \in [H]} : \mathcal{S} \times [H] \mapsto \Delta(A)$ denote the policy of the agent, where $\pi_h(\cdot|s) \in \Delta(\mathcal{A})$ specifies an action selection rule at time step $h$.

For any given policy $\pi$, the value function at step $h$, denoted by $V_h^\pi : \mathcal{S} \mapsto \mathbb{R}$, is given as

$$\forall s \in \mathcal{S}, \ h \in [H]: \quad V_h^\pi(s) := \mathbb{E}\left[\sum_{i=h}^H r_i(s_i, a_i)|s_h = s\right], \tag{1}$$

which measures the expected cumulative reward starting from state $s$ at time step $h$ until the end of the episode. The expectation is taken over the randomness of the trajectory generated following $a_i \sim \pi_i(\cdot|s_i)$ and the MDP dynamics $s_{i+1} \sim P_i(\cdot|s_i, a_i)$ for $i = h, \ldots, H$. We define $V_H^\pi(s) := 0$ for all $s \in \mathcal{S}$. The value function at the beginning of the episode, when $h = 1$, is often denoted simply as $V^\pi(s) := V_1^\pi(s)$. Given an initial state distribution $s_1 \sim \rho$ over $\mathcal{S}$, we also define $V^\pi(\rho) := \mathbb{E}_{s_1 \sim \rho}[V_1^\pi(s_1)]$.

Similarly, the $Q$-function of policy $\pi$ at step $h$, denoted by $Q_h^\pi : \mathcal{S} \times \mathcal{A} \mapsto \mathbb{R}$, is defined as

$$\forall (s, a) \in \mathcal{S} \times \mathcal{A}, \ h \in [H]: \quad Q_h^\pi(s, a) := \mathbb{E}\left[\sum_{i=h}^H r_i(s_i, a_i)|s_h = s, a_h = a\right], \tag{2}$$

which measures the expected discounted cumulative reward starting from state $s$ and taking action $a$ at time step $h$, and following policy $\pi$ thereafter, according to the time-dependent transitions. We define $Q_{H+1}^\pi(s, a) := 0$ and $Q^\pi(s, a) := Q_1^\pi(s, a)$ for all $(s, a) \in \mathcal{S} \times \mathcal{A}$. They satisfy the Bellman consistency equation, given by, for all $(s, a) \in \mathcal{S} \times \mathcal{A}, \ h \in [H]$:

$$Q_h^\pi(s, a) = r_h(s, a) + \mathbb{E}_{s_{h+1} \sim P_h(\cdot|s,a), a_{h+1} \sim \pi_{h+1}(\cdot|s_{h+1})}[Q_{h+1}^\pi(s_{h+1}, a_{h+1})]. \tag{3}$$

It is known that there exists at least one optimal policy $\pi^\star = (\pi_1^\star, \ldots, \pi_H^\star)$ that maximizes the value function $V^\pi(s)$ for all initial states $s \in \mathcal{S}$ [Puterman, 2014]. The corresponding optimal value function and Q-function are denoted as $V^\star$ and $Q^\star$, respectively. In particular, they satisfy the Bellman optimality equation, given by, for all $(s, a) \in \mathcal{S} \times \mathcal{A}, \ h \in [H]$:

$$Q_h^\star(s, a) = r_h(s, a) + \mathbb{E}_{s_{h+1} \sim P_h(\cdot|s,a), a_{h+1} \sim \pi_{h+1}^\star(\cdot|s_{h+1})}[Q_{h+1}^\star(s_{h+1}, a_{h+1})]. \tag{4}$$

**Goal: regret minimization in online RL.** In this paper, we are interested in the online RL setting, where the agent interacts with the episodic MDP sequentially for $T$ episodes, where in the $t$-th episode ($t \geqslant 1$), the agent executes a policy $\pi_t = \{\pi_{t,h}\}_{h=1}^H$ learned based on the data collected up to the $(t-1)$-th episode. To evaluate the performance of the learned policy, our goal is to minimize the cumulative regret, defined as

$$\mathsf{Regret}(T) = \sum_{t=1}^T (V^\star(\rho) - V^{\pi_t}(\rho)), \tag{5}$$

which measures the sub-optimality gap between the values of the optimal policy and the learned policies over $T$ episodes. In particular, we would like the regret to scale sub-linearly in $T$, so the sub-optimality gap is amortized over time.

## 2.2 Motivation: revisiting MEX from primal-dual lens

Recently, MEX [Liu et al., 2024] emerges as a promising framework for online RL, which balances exploration and exploitation in a single objective while naturally enabling function approximation. Consider a function class $\mathcal{Q} = \prod_{h=1}^H \mathcal{Q}_h$ of the Q-function. For any $f = \{f_h\}_{h \in [H]} \in \mathcal{Q}$, we denote the corresponding Q-function $Q_f = \{Q_{f,h}\}_{h \in [H]}$ with $Q_{f,h} = f_h$. At the beginning of the $t$-th episode, given the collection $\mathcal{D}_{t-1,h}$ of transition tuples $(s_h, a_h, s_{h+1})$ at step $h$ up to the $(t-1)$-th episode, MEX [Liu et al., 2024] (more precisely, its model-free variant) updates the Q-function estimate as

$$f_t = \arg\sup_{f \in \mathcal{Q}} \mathbb{E}_{s_1 \sim \rho}\left[\max_{a \in \mathcal{A}} Q_{f,1}(s_1, a)\right] - \alpha \mathcal{L}_t(f), \tag{6}$$

where $\alpha \geqslant 0$ is some regularization parameter, and $\mathcal{L}_t(f)$ is

$$\mathcal{L}_t(f) = \sum_{h=1}^H \left[\sum_{\xi_h \in \mathcal{D}_{t-1,h}} \left(r_h(s_h, a_h) + \max_{a \in \mathcal{A}} Q_{f,h+1}(s_{h+1}, a) - Q_{f,h}(s_h, a_h)\right)^2 \right. \tag{7}$$

$$- \inf_{g_h \in \mathcal{Q}_h} \sum_{\xi_h \in \mathcal{D}_{t-1,h}} \Big( r_h(s_h, a_h) + \max_{a \in \mathcal{A}} Q_{f,h+1}(s_{h+1}, a) - g_h(s_h, a_h) \Big)^2 \Big],$$

where $\xi_h = (s_h, a_h, s_{h+1})$ is the transition tuple. The first term in (6) promotes exploration by searching for $Q$-functions with higher values, while the second term ensures the Bellman consistency of the $Q$-function with the collected data transitions. The policy is then updated greedily from $Q_{f_t}$ to collect the next batch of data. While Liu et al. [2024] offered strong regret guarantees of MEX, there is little insight provided into the design of (6), which is deeply connected to the reward-biased framework in Kumar and Becker [1982].

**Interpretation from primal-dual lens.** We offer a new interpretation of MEX, where optimistic regularization arises naturally from a regularized Lagrangian formulation of certain constrained value maximization problem within a primal-dual optimization perspective. As a brief detour to build intuition, we consider a value maximization problem over the Q-function with the exact (i.e., population) Bellman optimality equation as the constraints:

$$\sup_{f \in \mathcal{Q}} \ \mathbb{E}_{s_1 \sim \rho} \Big[ \max_{a \in \mathcal{A}} Q_{f,1}(s_1, a) \Big] \tag{8}$$

$$\text{s.t.} \ \ Q_{f,h}(s,a) = r_h(s,a) + \mathbb{E}_{s' \sim P_h(\cdot|s,a)} \Big[ \max_{a \in \mathcal{A}} Q_{f,h+1}(s',a) \Big], \quad \forall (s,a,h) \in \mathcal{S} \times \mathcal{A} \times [H],$$

with the boundary condition $Q_{f,H+1} = 0$. When the optimal $Q$-function is realizable, i.e., $Q^\star \in \mathcal{Q}$, the unique solution of (8) recovers the true optimal $Q$-function $Q^\star$.

How is this connected to the MEX objective? Introducing the dual variables $\{\lambda_h\}_{h \in [H]}$, the regularized Lagrangian of (8) can be written as

$$\sup_{f \in \mathcal{Q}} \mathbb{E}_{s_1 \sim \rho} \Big[ \max_{a \in \mathcal{A}} Q_{f,1}(s_1, a) \Big] \tag{9}$$

$$+ \inf_{\{\lambda_h\}_{h \in [H]}} \sum_{h=1}^{H} \mathbb{E}_{(s,a,s') \sim \mathcal{D}_h} \Big\{ \lambda_h(s,a) \Big( r_h(s,a) + \max_{a \in \mathcal{A}} Q_{f,h+1}(s',a) - Q_{f,h}(s,a) \Big) + \frac{\beta}{2} \lambda_h(s,a)^2 \Big\},$$

where $\beta > 0$ is the regularization parameter of the dual variable,[5] and $\mathcal{D}_h$ denotes a properly defined joint distribution over the transition tuples that covers the state-action space over $(s,a)$. We invoke the trick in Dai et al. [2018], Baird [1995], which deals with the *double-sampling issue*, and reparameterize the dual variable

$$\lambda_h(s,a) = \frac{Q_{f,h}(s,a) - g_h(s,a)}{\beta}, \tag{10}$$

which satisfies

$$\forall \delta_h(s,a) : \qquad \lambda_h(s,a) \big( \delta_h(s,a) - Q_{f,h}(s,a) \big) + \frac{\beta}{2} \lambda_h(s,a)^2$$

$$= \frac{1}{2\beta} \Big[ \big( \delta_h(s,a) - Q_{f,h}(s,a) \big)^2 - \big( \delta_h(s,a) - g_h(s,a) \big)^2 \Big]. \tag{11}$$

Consequently, by setting $\delta_h(s,a) := r_h(s,a) + \max_{a \in \mathcal{A}} Q_{f,h+1}(s',a)$ in (11), the Lagrangian objective (9) becomes

$$\sup_{f \in \mathcal{Q}} \mathbb{E}_{s_1 \sim \rho} \Big[ \max_{a \in \mathcal{A}} Q_{f,1}(s_1, a) \Big] - \sum_{h=1}^{H} \frac{1}{2\beta} \sup_{g_h \in \mathcal{Q}_h} \mathbb{E}_{(s,a,s') \sim \mathcal{D}_h} \Big[ \big( r_h(s,a) + \max_{a \in \mathcal{A}} Q_{f,h+1}(s',a) - Q_{f,h}(s,a) \big)^2$$

$$- \big( r_h(s,a) + \max_{a \in \mathcal{A}} Q_{f,h+1}(s',a) - g_h(s,a) \big)^2 \Big]. \tag{12}$$

By replacing the population distribution $\mathcal{D}_h$ with its samples in $\mathcal{D}_{t-1,h}$ at each round, then we recover the model-free MEX algorithm in (7).

However, (6) is a bilevel optimization problem where in the lower level, another optimization problem $\max_{a \in \mathcal{A}} Q_{f,h}(s,a)$ needs to be computed in (7). This can be can be computationally intensive if not intractable. In this paper, inspired from this primal-dual view, we derive a more implementable algorithm.

---

[5] It is possible to use an $(s, a, h)$-dependent regularization too.

# 3 Value-incentivized Actor-Critic Method

## 3.1 Algorithm development

We now develop the proposed value-incentivized actor-critic method. In contrast to the model-free MEX for (12), we consider a value maximization problem over both the $Q$-function and the policy with the exact (i.e., population) Bellman *consistency* equation as the constraints:

$$\sup_{f \in \mathcal{Q}, \pi \in \mathcal{P}} \mathbb{E}_{s_1 \sim \rho, a_1 \sim \pi_1(\cdot|s_1)} \big[ Q_{f,1}(s_1, a_1) \big] \tag{13}$$

$$\text{s.t.} \quad Q_{f,h}(s, a) = r_h(s, a) + \mathbb{E}_{\substack{s' \sim P_h(\cdot|s,a) \\ a' \sim \pi_{h+1}(\cdot|s')}} \big[ Q_{f,h+1}(s', a') \big], \qquad \forall \, (s, a, h) \in \mathcal{S} \times \mathcal{A} \times [H],$$

where $\mathcal{P} = \prod_{h=1}^{H} \mathcal{P}_h$ is the policy class. This formulation explicits the joint optimization over the $Q$-function (critic) and the policy (actor), and uses the Bellman's consistency equation as the constraint, rather than the Bellman's optimality equation, which is key to obtain a more tractable optimization problem.

Similar as (9), we can write the regularized Lagrangian of (13) as

$$\sup_{f \in \mathcal{Q}, \pi \in \mathcal{P}} \mathbb{E}_{s_1 \sim \rho, a_1 \sim \pi_1(\cdot|s_1)} \big[ Q_{f,1}(s_1, a_1) \big] \tag{14}$$

$$+ \inf_{\{\lambda_h\}_{h=1}^{H}} \sum_{h=1}^{H} \mathbb{E}_{\substack{(s,a,s') \sim \mathcal{D}_h \\ a' \sim \pi_{h+1}(\cdot|s')}} \left\{ \lambda_h(s, a) \Big( r_h(s, a) + Q_{f,h+1}(s', a') - Q_{f,h}(s, a) \Big) + \frac{\beta}{2} \lambda_h(s, a)^2 \right\}.$$

Similar to earlier discussion, we also consider the reparameterization (10) which gives

$$\sup_{f, \pi \in \mathcal{P}} \left\{ V_f^\pi(\rho) - \sum_{h=1}^{H} \frac{1}{2\beta} \sup_{g_h \in \mathcal{Q}_h} \mathbb{E}_{\substack{(s,a,s') \sim \mathcal{D}_h \\ a' \sim \pi_{h+1}(\cdot|s')}} \Big[ \big( r_h(s, a) + Q_{f,h+1}(s', a') - Q_{f,h}(s, a) \big)^2 \right.$$

$$\left. - \big( r_h(s, a) + Q_{f,h+1}(s', a') - g_h(s, a) \big)^2 \Big] \right\}, \tag{15}$$

where we define

$$V_f^\pi(s) := \mathbb{E}_{a \sim \pi_1(\cdot|s)} \big[ Q_{f,1}(s, a) \big], \quad \text{and} \quad V_f^\pi(\rho) := \mathbb{E}_{s \sim \rho} \big[ V_f^\pi(s) \big]. \tag{16}$$

Note that, the objective function (15) is easier to optimize over both $Q_f$ and $\pi$. Replacing the population distribution $\mathcal{D}_h$ of $\xi = (s, a, s')$ by its empirical samples leads to the proposed algorithm, which is termed value-incentivized actor-critic (VAC) method; see Algorithm 1 for a summary.

---

**Algorithm 1** Value-incentivized Actor-Critic (VAC) for finite-horizon MDPs

---

1: **Input:** regularization coefficient $\alpha > 0$.
2: **Initialization:** dataset $\mathcal{D}_{0,h} := \emptyset$ for all $h \in [H]$.
3: **for** $t = 1, \cdots, T$ **do**
4:  Update Q-function estimation and policy:

$$(f_t, \pi_t) \leftarrow \arg \sup_{f \in \mathcal{Q}, \pi \in \mathcal{P}} \left\{ V_f^\pi(\rho) - \alpha \mathcal{L}_t(f, \pi) \right\}. \tag{17}$$

5:  Data collection: run $\pi_t$ to obtain a trajectory $\{s_{t,1}, a_{t,1}, s_{t,2}, \ldots, s_{t,H+1}\}$, and update the dataset $\mathcal{D}_{t,h} \leftarrow \mathcal{D}_{t-1,h} \cup \{(s_{t,h}, a_{t,h}, s_{t,h+1})\}$ for all $h \in [H]$.
6: **end for**

---

In Algorithm 1, at $t$-th iteration, given dataset $\mathcal{D}_{t-1,h}$ of transitions $(s_h, a_h, s_{h+1})$ collected from the previous iterations for all $h \in [H]$, and use the current policy $\pi_t$ to collect new action $a'$ for each tuples, we define the loss function as follows:

$$\mathcal{L}_t(f, \pi) = \sum_{h=1}^{H} \left\{ \sum_{\xi_h \in \mathcal{D}_{t-1,h}} \mathbb{E}_{a' \sim \pi_{h+1}(\cdot|s_{h+1})} \big( r_h(s_h, a_h) + Q_{f,h+1}(s_{h+1}, a') - Q_{f,h}(s_h, a_h) \big)^2 \right.$$

$$\left. - \inf_{g_h \in \mathcal{Q}_h} \sum_{\xi_h \in \mathcal{D}_{t-1,h}} \mathbb{E}_{a' \sim \pi_{h+1}(\cdot|s_{h+1})} \big( r_h(s_h, a_h) + Q_{f,h+1}(s_{h+1}, a') - g_h(s_h, a_h) \big)^2 \right\}, \tag{18}$$

where $\xi_h = (s_h, a_h, s_{h+1})$ is the transition tuple. To approximately solve the optimization problem (17), which is the sample version of (15), we can, in practice, employ first-order method, *i.e.*,

- **Critic evaluation:** Given the policy $\pi_{t-1}$ fixed, we solve the saddle-point problem for $f_t$ as *biased policy evaluation* for $\pi_{t-1}$, *i.e.*,

$$f_t = \arg\max_{f \in \mathcal{Q}} \ V_f^{\pi_{t-1}}(\rho) - \alpha \mathcal{L}_t(f, \pi_{t-1}). \tag{19}$$

- **Policy update:** Given the critic $f$ is fixed, we can update the policy $\pi$ through policy gradient following the gradient calculation in Nachum et al. [2019b].

Clearly, the proposed VAC recovers an actor-critic style algorithm, therefore, demonstrating the practical potential of the proposed algorithm. However, we emphasize the critic evaluation step is different from the vanilla policy evaluation, where we have $V_f^{\pi}(\rho)$ to bias the policy value. In contrast, MEX only admits an actor-critic implementation for $\alpha = 0$ (corresponding to vanilla actor-critic when there is no exploration) since their data loss term requires the *optimal* value function, while the data loss term $\mathcal{L}_t(f, \pi)$ is policy-dependent in VAC.

### 3.2 Theoretical guarantees

The design of VAC is versatile and can be implemented with arbitrary function approximation. To corroborate its efficacy, however, we focus on understanding its theoretical performance in the linear MDP model, which is popular in the literature [Jin et al., 2020, Lu et al., 2021].

**Assumption 1** (linear MDP, Jin et al. [2020]). *There exist* unknown *vectors $\zeta_h \in \mathbb{R}^d$ and* unknown *(signed) measures $\mu_h = (\mu_h^{(1)}, \cdots, \mu_h^{(d)})$ over $\mathcal{S}$ such that*

$$r_h(s, a) = \phi_h(s, a)^\top \zeta_h \qquad \text{and} \qquad P_h(s'|s, a) = \phi_h(s, a)^\top \mu_h(s'), \tag{20}$$

*where $\phi_h : \mathcal{S} \times \mathcal{A} \mapsto \mathbb{R}^d$ is a* known *feature map satisfying $\|\phi_h(s, a)\|_2 \leqslant 1$, and $\max\{\|\zeta_h\|_2, \|\mu_h(\mathcal{S})\|_2\} \leqslant \sqrt{d}$, for all $(s, a, s') \in \mathcal{S} \times \mathcal{A} \times \mathcal{S}$ and all $h \in [H]$.*

We also need to specify the function class $\mathcal{Q}$ for the $Q$-function and the policy class $\mathcal{P}$ for the policy. Under the linear MDP, it suffices to represent $Q$-function linearly w.r.t. $\phi_h(s, a)$, *i.e.*, $Q_h(s, a) = \phi_h(s, a)^\top \theta_h$, and the log-linear function approximation for the policy derived from the max-entropy policy [Ren et al., 2022], with the following two regularization assumptions on the weights.

**Assumption 2** (linear $Q$-function class). *The function class $\mathcal{Q} = \prod_{h=1}^H \mathcal{Q}_h$ is*

$$\forall h \in [H] : \ \mathcal{Q}_h := \left\{ f_{\theta,h} := \phi_h(\cdot, \cdot)^\top \theta : \|\theta\|_2 \leqslant (H + 1 - h)\sqrt{d}, \ \|f_{\theta,h}\|_\infty \leqslant H + 1 - h \right\}.$$

**Assumption 3** (log-linear policy class). *The policy class $\mathcal{P} = \prod_{h=1}^H \mathcal{P}_h$ is*

$$\forall h \in [H] : \quad \mathcal{P}_h := \left\{ \pi_{\omega,h} : \pi_{\omega,h}(a|s) = \frac{\exp\left(\phi_h(s, a)^\top \omega\right)}{\sum_{a' \in \mathcal{A}} \exp\left(\phi_h(s, a')^\top \omega\right)} \ with \ \|\omega\|_2 \leqslant BH\sqrt{d} \right\}$$

*with some constant $B > 0$.*

Under these assumptions, we first state the regret bound of Algorithm 1 in Theorem 1.

**Theorem 1.** *Suppose Assumptions 1-3 hold. We let $B = \frac{T \log |\mathcal{A}|}{dH}$ in Assumption 3, and set*

$$\alpha = \left( \frac{1}{H^2 T \log\left(\log |\mathcal{A}| T/\delta\right)} \log\left(1 + \frac{T^{3/2}}{d}\right) \right)^{1/2}. \tag{21}$$

*Then for any $\delta \in (0, 1)$, with probability at least $1 - \delta$, the regret of VAC (cf. Algorithm 1) satisfies*

$$\text{Regret}(T) = \mathcal{O}\left( dH^2 \sqrt{T} \sqrt{\log\left(\frac{\log(|\mathcal{A}|)T}{\delta}\right) \log\left(1 + \frac{T^{3/2}}{d}\right)} \right). \tag{22}$$

Theorem 1 shows that by choosing $B = \widetilde{O}(T/dH)$ and $\alpha = \widetilde{O}\left(\frac{1}{H\sqrt{T}}\right)$, the regret of VAC is no larger than the order of $\widetilde{O}(dH^2\sqrt{T})$ up to log-factors. Compared to the minimax lower bound $\widetilde{\Omega}(d\sqrt{H^3 T})$ [He et al., 2023], this suggests that our bound is near-optimal up to a factor of $\sqrt{H}$, but with practical implementation generalizable to arbitrary function approximator.

**Extension to the infinite-horizon setting.** Our algorithm and theory can be extended to the infinite-horizon discounted setting leveraging the sampling procedure in Yuan et al. [2023, Algorithm 3]. We demonstrate that the sample complexity of VAC is no larger than $\widetilde{O}\left(\frac{d^2}{(1-\gamma)^5\varepsilon^2}\right)$ to return an $\varepsilon$-optimal policy, where $\gamma$ is the discount factor. This rate is near-optimal up to polynomial factors of $\frac{1}{1-\gamma}$ and logarithmic factors. We leave the details to the appendix.

**Extension to the general function approximation.** Our theoretical analysis can also be extended to general function approximation, under standard assumptions for general function approximation such as low *generalized Eluder coefficient* (GEC) [Zhong et al., 2022, Liu et al., 2024]. The corresponding tight regret bound is provided in Appendix B.3, which matches the bound given in Liu et al. [2024, Corollary 5.2] under similar assumptions.

**Extension to KL-regularized MDPs.** Recently, MDPs regularized by the Kullback-Leibler (KL) divergence $\mathsf{KL}(\pi\|\pi_{\mathrm{ref}})$, with respect to a reference policy $\pi_{\mathrm{ref}} = \{\pi_{\mathrm{ref},h}\}_{h\in[H]} : \mathcal{S} \times [H] \mapsto \Delta(\mathcal{A})$, has attracted much attention for preventing over-optimization and increasing stability of the learning process [Ouyang et al., 2022, Yang et al., 2025]. Our framework of VAC can be extended straightforwardly, by invoking the soft Bellman consistency equation in the derivation:

$$Q_{\tau,h}^\pi(s,a) \coloneqq r_h(s,a) + \mathbb{E}_{\substack{s_{h+1}\sim P_h(\cdot|s,a) \\ a_{h+1}\sim\pi_{h+1}(\cdot|s_{h+1})}} \left[Q_{\tau,h+1}^\pi(s_{h+1},a_{h+1}) - \tau\log\frac{\pi_{h+1}(a_{h+1}|s_{h+1})}{\pi_{\mathrm{ref},h+1}(a_{h+1}|s_{h+1})}\right],$$
(23)

where $\tau > 0$ is the regularization parameter. We omit the details for conciseness.

# 4 Experiments

We provide numerical experiments to demonstrate the efficacy of the value-incentivized regularization in the actor-critic framework.

**Setup.** We evaluate on two challenging continuous-control benchmarks in MuJoCo [Todorov et al., 2012]: Ant-v4 and Walker2d-v4. For the base learner, we adopt Soft Actor-Critic (SAC) implemented in Stable-Baselines3 [Raffin et al., 2021] and add a simple sample-based value-incentivized term to its critic objective.

*Critic update.* With two critics $\{Q_{\theta_j}\}_{j=1}^2$ and target networks $\{Q_{\theta_j^-}\}_{j=1}^2$, the SAC target is

$$y = r(s,a) + \gamma\left(\min_j Q_{\theta_j^-}(s',a') - \tau_{\mathrm{ent}}\log\pi(a'\mid s')\right), \quad a'\sim\pi(\cdot\mid s'),$$

Here, $r(s,a)$ denotes the one-step reward, and $\pi$ denotes the current stochastic policy used by SAC for target evaluation (i.e., $a'\sim\pi(\cdot\mid s')$). Our modified critic objective uses minibatch sample averages (replacing population expectations) and reads

$$\widehat{\mathcal{L}}_Q(\{\theta_j\}) = \sum_{(s,a,s')\in\mathcal{B}}\sum_{j=1}^2\left(Q_{\theta_j}(s,a) - y\right)^2 - \frac{1}{|\mathcal{B}|\alpha}\sum_{s\in\mathcal{B}}\sum_{j=1}^2\frac{1}{N}\sum_{i=1}^N Q_{\theta_j}(s,a_i).$$

Here we use a single Monte Carlo sample $\frac{1}{N}\sum_{i=1}^N Q_{\theta_j}(s,a_i)$ to approximate $V_f^\pi(s) = \mathbb{E}_{a\sim\pi(\cdot|s)}[Q_f(s,a)]$. We found that setting $N = 1$, i.e., using a single policy sample is good enough. We use a minibatch $\mathcal{B}$ of size 256 sampled uniformly from a replay buffer of size $10^6$. The buffer stores the historical data: during the first 100 steps we act uniformly at random (warm-up). After warm-up, the current policy selects one action at each step, and the resulting $(s,a,r(s,a),s')$ is appended to the replay buffer. We optimize the critic with Adam (learning rate $3\times10^{-4}$), perform one gradient step, and update target networks every step via Polyak averaging with $\tau_{\mathrm{polyak}} = 0.005$. Training starts after collecting 100 steps. The entropy coefficient is tuned automatically by optimizing a learnable log-temperature to match a target entropy.

*Policy update.* The actor is updated with the standard SAC loss

$$\widehat{\mathcal{L}}_\pi(\omega) = \frac{1}{|\mathcal{B}|}\sum_{s\in\mathcal{B}}\mathbb{E}_{a\sim\pi_\omega(\cdot|s)}\left[\tau_{\mathrm{ent}}\log\pi_\omega(a\mid s) - \min_{j\in\{1,2\}}Q_{\theta_j}(s,a)\right],$$

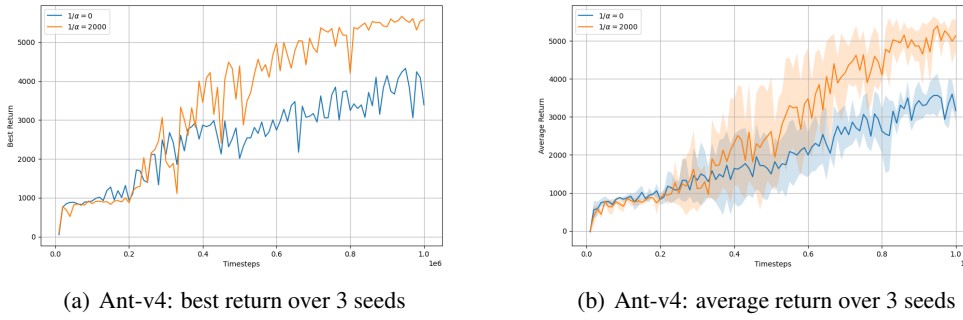

(a) Ant-v4: best return over 3 seeds

(b) Ant-v4: average return over 3 seeds

Figure 1: Ant-v4 with $1/\alpha \in \{0, 2000\}$. Shaded area indicates standard deviation across 3 seeds.

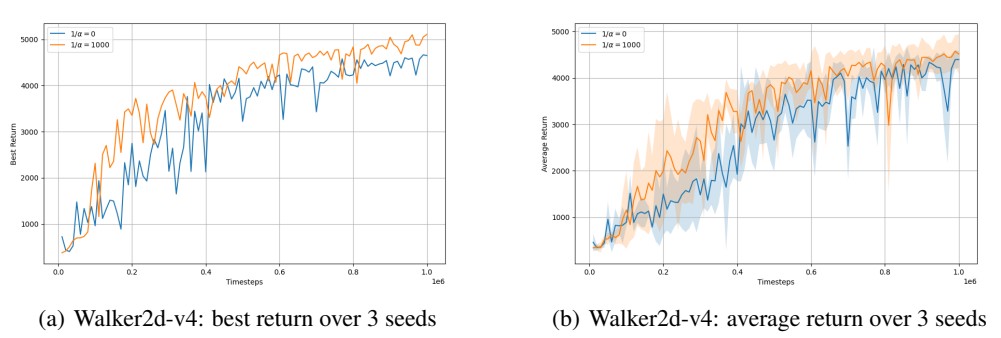

(a) Walker2d-v4: best return over 3 seeds

(b) Walker2d-v4: average return over 3 seeds

Figure 2: Walker2d-v4 with $1/\alpha \in \{0, 1000\}$. Shaded area indicates standard deviation across 3 seeds.

estimated with one reparameterized sample per state using the Tanh-squashed Gaussian policy; we optimize the actor with Adam (learning rate $3 \times 10^{-4}$) in lockstep with the critic. VAC modifies only the critic objective above, leaving the actor update identical to SAC.

*Network architecture.* Both critics are separate MLPs with two hidden layers of 256 ReLU units each ("twin Q"), and the actor is an MLP with the same hidden sizes producing a Gaussian policy with Tanh-squashed actions.

**Results.** We run both experiments for $10^6$ iterations over 3 seeds. Figures 1 and 2 summarize performance. For each task, we plot the best return across the three seeds and the average return over seeds; shaded regions denote standard deviation. The VAC regularization improves sample efficiency compared to SAC.

## 5 Conclusion

In this paper, we develop a provably sample-efficient actor-critic method, called value-incentivized actor-critic (VAC), for online RL with a single easy-to-optimize objective function that avoids complex bilevel optimization in the presence of complex function approximation. We theoretically establish VAC's efficacy by proving it achieves $\widetilde{O}(\sqrt{T})$-regret in both episodic and discounted settings. Our work suggests that a unified Lagrangian-based objective offers a promising direction for principled and practical online RL, allowing many venues for future developments. Further, we empirically validate VAC's performance on MuJoCo tasks. Follow-up efforts will focus on more empirical validation, and extending the algorithm design to multi-agent settings.

## Acknowledgments and Disclosure of Funding

This work of T. Yang and Y. Chi is supported in part by the grants NSF DMS-2134080, CCF-2106778, and ONR N00014-19-1-2404. T. Yang is also graciously supported by the CMU Wei

Shen and Xuehong Zhang Presidential Fellowship. The work of B. Dai is supported in part by the grants NSF ECCS-2401391, IIS-2403240, and ONR N00014-25-1-2173.

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

## A   Technical Lemmas

We provide some technical lemmas that will be used in our proofs.

**Lemma 2** (Freedman's inequality, Lemma D.2 in Liu et al. [2024]). *Let $\{X_t\}_{t \leqslant T}$ be a real-valued martingale difference sequence adapted to filtration $\{\mathcal{F}_t\}_{t \leqslant T}$. If $|X_t| \leqslant R$ almost surely, then for any $\eta \in (0, 1/R)$ it holds that with probability at least $1 - \delta$,*

$$\sum_{t=1}^{T} X_t \leqslant \mathcal{O}\left(\eta \sum_{t=1}^{T} \mathbb{E}[X_t^2 | \mathcal{F}_{t-1}] + \frac{\log(1/\delta)}{\eta}\right).$$

**Lemma 3** (Covering number of $\ell_2$ ball, Lemma D.5 in Jin et al. [2020]). *For any $\epsilon > 0$ and $d \in \mathbb{N}_+$, the $\epsilon$-covering number of the $\ell_2$ ball of radius $R$ in $\mathbb{R}^d$ is bounded by $(1 + 2R/\epsilon)^d$.*

**Lemma 4** (Lemma 11 in Abbasi-Yadkori et al. [2011]). *Let $\{x_s\}_{s \in [T]}$ be a sequence of vectors with $x_s \in \mathcal{V}$ for some Hilbert space $\mathcal{V}$. Let $\Lambda_0$ be a positive definite matrix and define $\Lambda_t = \Lambda_0 + \sum_{s=1}^{t} x_s x_s^\top$. Then it holds that*

$$\sum_{s=1}^{T} \min\left\{1, \|x_s\|_{\Lambda_{s-1}^{-1}}\right\} \leqslant 2 \log\left(\frac{\det(\Lambda_T)}{\det(\Lambda_0)}\right).$$

**Lemma 5** (Lemma F.3 in Du et al. [2021]). *Let $\mathcal{X} \subset \mathbb{R}^d$ and $\sup_{x \in \mathcal{X}} \|x\|_2 \leqslant B_X$. Then for any $n \in \mathbb{N}_+$, we have*

$$\forall \lambda > 0: \quad \max_{x_1, \cdots, x_n \in \mathcal{X}} \log \det\left(I_d + \frac{1}{\lambda} \sum_{i=1}^{n} x_i x_i^\top\right) \leqslant d \log\left(1 + \frac{n B_X^2}{d\lambda}\right).$$

**Lemma 6** (Corollary A.7 in Edelman et al. [2022]). *Define the softmax function as $\mathsf{softmax}(\cdot) : \mathbb{R}^d \to \Delta^d$ by $\mathsf{softmax}(x)_i = \frac{\exp(x_i)}{\sum_{j=1}^{d} \exp(x_j)}$ for all $i \in [d]$ and $x \in \mathbb{R}^d$. Then for any $x, y \in \mathbb{R}^d$, we have*

$$\|\mathsf{softmax}(x) - \mathsf{softmax}(y)\|_1 \leqslant 2\|x - y\|_\infty.$$

## B   Proofs for Episodic MDPs

### B.1   Proof of Theorem 1

**Notation and preparation.**   For notation simplicity, we let $f^\star := Q^\star$ be the optimal Q-function. We let $\Pi := \Delta(\mathcal{A})^{\mathcal{S}}$ denote the whole policy space. We have $\mathcal{P}_h \subset \Pi$ for all $h \in [H]$. We also define the transition tuples

$$\xi := (s, a, s') \in \mathcal{S} \times \mathcal{A} \times \mathcal{S} \quad \text{and} \quad \xi_h := (s_h, a_h, s_{h+1}) \in \mathcal{S} \times \mathcal{A} \times \mathcal{S}. \tag{24}$$

Given any policy profile $\pi = \{\pi_h\}_{h \in [H]}$ and $f = \{f_h : \mathcal{S} \times \mathcal{A} \mapsto \mathbb{R}\}$, we define $\mathbb{P}_h^\pi f$ as

$$\forall (s_h, a_h) \in \mathcal{S} \times \mathcal{A}: \quad \mathbb{P}_h^\pi f(s_h, a_h) := r_h(s_h, a_h) + \mathbb{E}_{\substack{s_{h+1} \sim \mathbb{P}_h(\cdot | s_h, a_h), \\ a_{h+1} \sim \pi_{h+1}(\cdot | s_{h+1})}} [f_{h+1}(s_{h+1}, a_{h+1})], \tag{25}$$

and let $\mathbb{P}^\pi f := \{\mathbb{P}_h^\pi f\}_{h \in [H]}$. Let

$$\Theta_h := \{\theta : f_{\theta, h} \in \mathcal{Q}_h\}, \quad \Omega := \left\{\omega : \|\omega\|_2 \leqslant BH\sqrt{d}\right\} \tag{26}$$

be the parameter space of $\mathcal{Q}_h$ and $\mathcal{P}_h$, respectively for all $h \in [H]$. We also define

$$V_{f,h}^\pi(s) := \mathbb{E}_{a \sim \pi(\cdot | s)}[Q_{f,h}(s, a)] \quad \text{and} \quad V_{f,h}^\pi(\rho) := \mathbb{E}_{s \sim \rho}[V_{f,h}^\pi(s)], \quad \forall f \in \mathcal{Q}, \pi \in \mathcal{P}, s \in \mathcal{S}, h \in [H]. \tag{27}$$

We'll repeatedly use the following lemma, which guarantees that under Assumption 1, the optimal Q-function $Q^\star$ is in $\mathcal{Q}$, and $\mathbb{P}^\pi f \in \mathcal{Q}$ for any $f \in \mathcal{Q}$ and $\pi \in \Pi^H$. Similar results can be found in the literature, e.g., Jin et al. [2020]. For completeness, we include the proof of Lemma 7 in Appendix B.2.1.

**Lemma 7** (Linear MDP $\Rightarrow$ Bellman completeness + realizability). *Under Assumption 1, we have*

- *(realizability) $Q^\star \in \mathcal{Q}$;*

- *(Bellman completeness) $\forall \pi \in \Pi$ and $f \in \mathcal{Q}$, $\mathbb{P}^\pi f \in \mathcal{Q}$.*

We also use the following lemma, which bounds the difference between the optimal value function $V^\star$ and $\max_{\pi \in \mathcal{P}} V^\pi$ — the optimal value over the policy class $\mathcal{P}$, where we let

$$\widetilde{\pi}_h^\star := \arg \max_{\pi_h \in \mathcal{P}_h} V^\pi_{f^\star, h}(\rho), \quad \forall h \in [H], \tag{28}$$

and $\widetilde{\pi}^\star = \{\widetilde{\pi}_h^\star\}_{h \in [H]}$ be the optimal policy within the policy class $\mathcal{P}$. The proof of Lemma 8 is deferred to Appendix B.2.2.

**Lemma 8** (model error with log-linear policies). *Under Assumptions 1-3, we have*

$$\forall s \in \mathcal{S}, h \in [H]: \quad 0 \leqslant V_h^\star(s) - V_{f^\star, h}^{\widetilde{\pi}^\star}(s) \leqslant \frac{\log |\mathcal{A}|}{B}, \tag{29}$$

*where $B$ is defined in Assumption 3.*

**Main proof.** We first decompose the regret (cf. (5)) as follows:

$$\mathsf{Regret}(T) = \sum_{t=1}^T (V^\star(\rho) - V^{\pi_t}(\rho)) = \underbrace{\sum_{t=1}^T \left( V^\star(\rho) - V_{f_t}^{\pi_t}(\rho) \right)}_{(\mathrm{i})} + \underbrace{\sum_{t=1}^T \left( V_{f_t}^{\pi_t}(\rho) - V^{\pi_t}(\rho) \right)}_{(\mathrm{ii})}, \tag{30}$$

where recall we define $V_f^\pi = V_{f,1}^\pi$ in (16). We will bound the two terms separately.

**Step 1: bounding term (i).** The linear MDP assumption guarantees that $Q^\star \in \mathcal{Q}$ by Lemma 7, and by definition (28), $\widetilde{\pi}^\star$ is in $\mathcal{P}$. Thus by our update rule (17), we have

$$\forall t \in \mathbb{N}_+: \quad V_{f^\star}^{\widetilde{\pi}^\star}(\rho) - \alpha \mathcal{L}_t(f^\star, \widetilde{\pi}^\star) \leqslant V_{f_t}^{\pi_t}(\rho) - \alpha \mathcal{L}_t(f_t, \pi_t),$$

which gives

$$V_{f^\star}^{\widetilde{\pi}^\star}(\rho) - V_{f_t}^{\pi_t}(\rho) \leqslant \alpha \left( \mathcal{L}_t(f^\star, \widetilde{\pi}^\star) - \mathcal{L}_t(f_t, \pi_t) \right).$$

Invoking Lemma 8, we have

$$V^\star(\rho) - V_{f_t}^{\pi_t}(\rho) \leqslant \alpha \left( \mathcal{L}_t(f^\star, \widetilde{\pi}^\star) - \mathcal{L}_t(f_t, \pi_t) \right) + \frac{\log |\mathcal{A}|}{B}. \tag{31}$$

Thus to bound (i), it suffices to bound $\mathcal{L}_t(f^\star, \widetilde{\pi}^\star) - \mathcal{L}_t(f_t, \pi_t)$ for each $t \in [T]$. To introduce our lemmas, we define $\ell_h : \mathcal{Q}_h \times \mathcal{S} \times \mathcal{A} \times \Pi \mapsto \mathbb{R}$ for all $h \in [H]$ as

$$\ell_h(f, s, a, \pi) := \left( \mathbb{E}_{\substack{s' \sim \mathbb{P}_h(\cdot|s,a), \\ a' \sim \pi_{h+1}(\cdot|s')}} [r_h(s,a) + f_{h+1}(s', a') - f_h(s, a)] \right)^2. \tag{32}$$

We give the following lemma that bounds (i), whose proof is given in Appendix B.2.3.

**Lemma 9.** *Suppose Assumptions 1-3 hold. For any $\delta \in (0, 1)$, with probability at least $1 - \delta$, for any $t \in [T]$, we have*

$$\mathcal{L}_t(f^\star, \widetilde{\pi}^\star) - \mathcal{L}_t(f_t, \pi_t) \leqslant -\frac{1}{2} \sum_{i=1}^{t-1} \sum_{h=1}^H \mathbb{E}_{(s_{i,h}, a_{i,h}) \sim d_{\rho,h}^{\pi_i}} [\ell_h(f_t, s_{i,h}, a_{i,h}, \pi_t)]$$

$$+ CH^3 \left( d \log \left( \frac{BHdT}{\delta} \right) + \frac{T \log |\mathcal{A}|}{BH} \right) \tag{33}$$

*for some absolute constant $C > 0$. Here, $d_{\rho,h}^{\pi_i}$ is the state-action visitation distribution induced by policy $\pi_i$ at step $h$.*

By (31) and Lemma 9, we have

$$V^\star(\rho) - V_{f_t}^{\pi_t}(\rho) \leqslant \alpha \left\{ -\frac{1}{2} \sum_{i=1}^{t-1} \sum_{h=1}^{H} \mathbb{E}_{(s_{i,h},a_{i,h}) \sim d_{\rho,h}^{\pi_i}} [\ell_h(f_t, s_{i,h}, a_{i,h}, \pi_t)] + CH^3 d \log \left( \frac{BHdT}{\delta} \right) \right\}$$
$$+ \left( CH^2 \alpha T + 1 \right) \frac{\log |\mathcal{A}|}{B},$$

which gives

$$(i) \leqslant \alpha \left\{ -\frac{1}{2} \sum_{t=1}^{T} \sum_{i=1}^{t-1} \sum_{h=1}^{H} \left( \mathbb{E}_{(s_{i,h},a_{i,h}) \sim d_{\rho,h}^{\pi_i}} [\ell_h(f_t, s_{i,h}, a_{i,h}, \pi_t)] \right) + CTH^3 d \log \left( \frac{BHdT}{\delta} \right) \right\}$$
$$+ \left( CH^2 \alpha T + 1 \right) \frac{T \log |\mathcal{A}|}{B}. \tag{34}$$

**Step 2: bounding term (ii).**  For any $\lambda > 0$, we define

$$d(\lambda) := d \log \left( 1 + \frac{T}{d\lambda} \right). \tag{35}$$

We use the following lemma to bound (ii), whose proof is in Appendix B.2.4.

**Lemma 10.** *Under Assumption 1, for any $\eta > 0$, we have*

$$\sum_{t=1}^{T} \left| V_{f_t}^{\pi_t}(\rho) - V^{\pi_t}(\rho) \right| \leqslant \eta \sum_{t=1}^{T} \sum_{i=1}^{t-1} \sum_{h=1}^{H} \mathbb{E}_{(s_i,a_i) \sim d_{\rho,h}^{\pi_i}} \ell_h(f_t, s_i, a_i, \pi_t) + (6H^2 + H/\eta)d(\lambda) + H^2 \lambda dT.$$

By Lemma 10, we have

$$(ii) \leqslant \eta \sum_{t=1}^{T} \sum_{i=1}^{t-1} \sum_{h=1}^{H} \mathbb{E}_{(s_i,a_i) \sim d_{\rho,h}^{\pi_i}} \ell_h(f_t, s_i, a_i, \pi_t) + (6H^2 + H/\eta)d(\lambda) + H^2 \lambda dT. \tag{36}$$

**Step 3: combining (i) and (ii).**  Substituting (34) and (36) into (30), and letting $\eta = \frac{\alpha}{2}$, we have

$$\mathsf{Regret}(T) \leqslant \alpha CTH^3 d \log \left( \frac{BHdT}{\delta} \right) + \left( CH^2 \alpha T + 1 \right) \frac{T \log |\mathcal{A}|}{B}$$
$$+ (6H^2 + 2H/\alpha)d(\lambda) + H^2 \lambda dT. \tag{37}$$

Setting $\lambda = \frac{1}{\sqrt{T}}$, $\alpha = \left( \frac{1}{H^2 T \log(\log |\mathcal{A}| T/\delta)} log \left( 1 + \frac{T^{3/2}}{d} \right) \right)^{1/2}$, and $B = \frac{T \log |\mathcal{A}|}{dH}$ in the above bound, we have with probability at least $1 - \delta$,

$$\mathsf{Regret}(T) \leqslant C' dH^2 \sqrt{T} \sqrt{\log \left( \frac{\log(|\mathcal{A}|)T}{\delta} \right) \log \left( 1 + \frac{T^{3/2}}{d} \right)}$$

for some absolute constant $C' > 0$. This completes the proof of Theorem 1.

## B.2   Proof of key lemmas

### B.2.1   Proof of Lemma 7

Assumption 1 guarantees that

$$Q_h^\star(s_h, a_h) = r_h(s_h, a_h) + \mathbb{E}_{s_{h+1} \sim \mathbb{P}_h(\cdot|s_h,a_h)} \left[ V_{h+1}^\star(s_{h+1}) \right]$$
$$= \phi_h(s_h, a_h)^\top \zeta_h + \int_{\mathcal{S}} \mathbb{P}_h(s_{h+1}|s_h, a_h) V_{h+1}^\star(s_{h+1}) ds_{h+1}$$
$$= \phi_h(s_h, a_h)^\top \underbrace{\left( \zeta_h + \int_{\mathcal{S}} V_{h+1}^\star(s_{h+1}) d\mu_h(s_{h+1}) \right)}_{:= \nu_h^\star}, \tag{38}$$

where $\nu_h^\star \in \mathbb{R}^d$ satisfies

$$\|\nu_h^\star\|_2 = \left\| \zeta_h + \int_{\mathcal{S}} V_{h+1}^\star(s_{h+1}) d\mu_h(s_{h+1}) \right\|_2$$
$$\leqslant \|\zeta_h\|_2 + \|V_{h+1}^\star\|_\infty \|\mu_h(\mathcal{S})\|_2 \leqslant \sqrt{d} + (H - h)\sqrt{d} = \sqrt{d}(H - h + 1).$$

We also have $\|Q_h^\star\|_\infty \leqslant H + 1 - h$ for all $h \in [H]$. Thus $Q^\star \in \mathcal{Q}$.

Moreover, for any $f \in \mathcal{Q}$, we have

$$\mathbb{P}_h^\pi f(s_h, a_h) = r_h(s_h, a_h) + \mathbb{E}_{\substack{s_{h+1} \sim \mathbb{P}_h(\cdot|s_h, a_h) \\ a_{h+1} \sim \pi_{h+1}(\cdot|s_{h+1})}} [f_{h+1}(s_{h+1}, a_{h+1})]$$

$$= \phi_h(s_h, a_h)^\top \zeta_h + \int_{\mathcal{S}} \mathbb{P}_h(s_{h+1}|s_h, a_h) \mathbb{E}_{a_{h+1} \sim \pi_{h+1}(\cdot|s_{h+1})} [f_{h+1}(s_{h+1}, a_{h+1})] \, ds_{h+1}$$

$$= \phi_h(s_h, a_h)^\top \underbrace{\left( \zeta_h + \int_{\mathcal{S}} \left( \mathbb{E}_{a_{h+1} \sim \pi_{h+1}(\cdot|s_{h+1})} f_{h+1}(s_{h+1}, a_{h+1}) \right) d\mu_h(s_{h+1}) \right)}_{:=\zeta_h},$$

where $\zeta_h \in \mathbb{R}^d$ satisfies

$$\|\zeta_h\|_2 = \left\| \zeta_h + \int_{\mathcal{S}} \left( \mathbb{E}_{a_{h+1} \sim \pi_{h+1}(\cdot|s_{h+1})} f_{h+1}(s_{h+1}, a_{h+1}) \right) d\mu_h(s_{h+1}) \right\|_2$$
$$\leqslant \|\zeta_h\|_2 + \|f_{h+1}\|_\infty \|\mu_h\|_2 \leqslant \sqrt{d} + (H - h)\sqrt{d} = \sqrt{d}(H - h + 1).$$

In addition, we have

$$\|\mathbb{P}_h^\pi f\|_\infty \leqslant \|r_h\|_\infty \|f_{h+1}\|_\infty \leqslant H - h + 1, \quad \forall h \in [H].$$

Thus $\mathbb{P}^\pi f \in \mathcal{Q}$.

### B.2.2  Proof of Lemma 8

From Lemma 7, it is known that for all $h \in [H]$, there exists $\nu_h^\star \in \Theta_h$ such that

$$Q_h^\star(s, a) = \phi_h(s, a)^\top \nu_h^\star, \quad \forall (s, a) \in \mathcal{S} \times \mathcal{A}. \tag{39}$$

Let

$$\pi_h(a|s) := \frac{\exp(B\phi_h(s, a)^\top \nu_h^\star)}{\sum_{a' \in \mathcal{A}} \exp(B\phi_h(s, a')^\top \nu_h^\star)}, \quad \forall (s, a) \in \mathcal{S} \times \mathcal{A}, \tag{40}$$

where $B$ is defined in Assumption 3. It follows that $\pi_h \in \mathcal{P}_h$, and for all $s \in \mathcal{S}$, $\pi_h(\cdot|s)$ is the solution to the following optimization problem [Beck, 2017, Example 3.71]:

$$\max_{p \in \Delta(\mathcal{A})} \quad \langle p, Q_h^\star(s, a) \rangle + \frac{1}{B} \mathcal{H}(p), \quad \text{where} \quad \mathcal{H}(p) := -\sum_{a \in \mathcal{A}} p(a) \log p(a). \tag{41}$$

Here, $\mathcal{H}(\cdot)$ is the entropy function satisfying

$$0 \leqslant \mathcal{H}(p) \leqslant \log |\mathcal{A}|, \quad \forall p \in \Delta(\mathcal{A}). \tag{42}$$

The optimality of $\pi_h$ for (41), together with (42), implies

$$\forall s \in \mathcal{S}: \quad V_{f^\star, h}^\pi(s) + \frac{\log |\mathcal{A}|}{B} \geqslant \langle \pi_h(\cdot|s), Q_h^\star(s, a) \rangle + \frac{1}{B} \mathcal{H}(\pi_h(\cdot|s))$$

$$\geqslant \langle \pi_h^\star(\cdot|s), Q_h^\star(s, a) \rangle + \frac{1}{B} \mathcal{H}(\pi_h^\star(\cdot|s))$$

$$= V_h^\star(s) + \frac{1}{B} \mathcal{H}(\pi_h^\star(\cdot|s)) \geqslant V_h^\star(s), \tag{43}$$

which further indicates

$$\max_{\pi_h' \in \mathcal{P}_h} V_{f^\star, h}^{\pi_h'}(s) \geqslant V_h^\star(s) - \frac{\log |\mathcal{A}|}{B}. \tag{44}$$

The desired bound (29) follows from the above inequality and the fact that $V_h^\star(s) = \max_{a \in \mathcal{A}} Q^\star(s, a) \geqslant V_{f^\star, h}^{\pi'}(s)$ for any policy profile $\pi'$, $s \in \mathcal{S}$ and $h \in [H]$.

### B.2.3 Proof of Lemma 9

We bound the two terms $\mathcal{L}_t(f^\star, \widetilde{\pi}^\star)$ and $-\mathcal{L}_t(f_t, \pi_t)$ on the left-hand side of (33) separately.

**Step 1: bounding $-\mathcal{L}_t(f_t, \pi_t)$.** Given $f, f' \in \mathcal{Q}$, data tuple $\xi = (s, a, s')$ and policy profile $\pi = \{\pi_h\}_{h=1}^H \in \Pi^H$, we define the random variable

$$l_h(f, f', \xi, \pi) := r_h(s, a) + f_{h+1}(s', a') - f'_h(s, a), \quad \forall h \in [H], \tag{45}$$

where $a' \sim \pi_{h+1}(\cdot|s')$. Then we have (recall we define $\mathbb{P}^\pi f$ in (25))

$$l_h(f, \mathbb{P}^\pi f, \xi, \pi) = f_{h+1}(s', a') - \mathbb{E}_{\substack{s' \sim \mathbb{P}_h(\cdot|s,a) \\ a' \sim \pi_{h+1}(\cdot|s')}} [f_{h+1}(s', a')], \tag{46}$$

which indicates that for any $f, f' \in \mathcal{Q}, \xi$ and $\pi$,

$$l_h(f, f', \xi, \pi) - l_h(f, \mathbb{P}^\pi f, \xi, \pi) = \mathbb{E}_{\substack{s' \sim \mathbb{P}_h(\cdot|s,a) \\ a' \sim \pi_{h+1}(\cdot|s')}} [l_h(f, f', \xi, \pi)]. \tag{47}$$

For any $f \in \mathcal{Q}, \pi \in \Pi^H$ and $t \in [T]$, we define $X_{f,\pi,h}^t$ as

$$X_{f,\pi,h}^t := \mathbb{E}_{a' \sim \pi_{h+1}(\cdot|s_{t,h+1})} \left[ l_h(f, f, \xi_{t,h}, \pi)^2 - l_h(f, \mathbb{P}^\pi f, \xi_{t,h}, \pi)^2 \right], \tag{48}$$

where $\xi_{t,h} := (s_{t,h}, a_{t,h}, s_{t,h+1})$ is the transition tuple collected at time $t$ and step $h$. Then we have for any $f \in \mathcal{Q}$:

$$
\begin{aligned}
\sum_{i=1}^{t-1} X_{f,\pi,h}^i &= \sum_{i=1}^{t-1} \mathbb{E}_{a' \sim \pi_{h+1}(\cdot|s_{i,h+1})} l_h(f, f, \xi_{i,h}, \pi)^2 - \sum_{i=1}^{t-1} \mathbb{E}_{a' \sim \pi_{h+1}(\cdot|s'_{i,h+1})} l_h(f, \mathbb{P}^\pi f, \xi_{i,h}, \pi)^2 \\
&\leqslant \sum_{i=1}^{t-1} \mathbb{E}_{a' \sim \pi_{h+1}(\cdot|s'_{i,h+1})} l_h(f, f, \xi_{i,h}, \pi)^2 - \inf_{g \in \mathcal{Q}} \sum_{i=1}^{t-1} \mathbb{E}_{a' \sim \pi_{h+1}(\cdot|s'_{i,h+1})} l_h(f, g, \xi_{i,h}, \pi)^2 = \mathcal{L}_{t,h}(f, \pi),
\end{aligned}
\tag{49}
$$

where the inequality uses the fact that $\mathbb{P}^\pi f \in \mathcal{Q}$, which is guaranteed by Lemma 7. Here, we define

$$
\begin{aligned}
\mathcal{L}_{t,h}(f, \pi) &:= \sum_{i=1}^{t-1} \mathbb{E}_{a' \sim \pi_{h+1}(\cdot|s_{i,h+1})} \left[ \left( r_h(s_{i,h}, a_{i,h}) + f_{h+1}(s_{i,h+1}, a') - f_h(s_{i,h}, a_{i,h}) \right)^2 \right] \\
&\quad - \inf_{g \in \mathcal{Q}} \sum_{i=1}^{t-1} \mathbb{E}_{a' \sim \pi_{h+1}(\cdot|s_{i,h+1})} \left[ \left( r_h(s_{i,h}, a_{i,h}) + f_{h+1}(s_{i,h+1}, a') - g(s_{i,h}, a_{i,h}) \right)^2 \right].
\end{aligned}
\tag{50}
$$

Therefore, to upper bound $-\mathcal{L}_t(f_t, \pi_t) = -\sum_{h=1}^H \mathcal{L}_{t,h}(f_t, \pi_t)$, it suffices to bound $-\sum_{i=1}^{t-1} X_{f_t,\pi_t,h}^i$ for all $h \in [H]$. In what follows, we use Freedman's inequality (Lemma 2) and a covering number argument similar to that in Yang et al. [2025] to give the desired bound.

**Step 1.1: building the covering argument.** We start with some basic preparation on the covering argument. For any $\mathcal{X} \subset \mathbb{R}^d$, let $\mathcal{N}(\mathcal{X}, \epsilon, \|\cdot\|)$ be the $\epsilon$-covering number of $\mathcal{X}$ with respect to the norm $\|\cdot\|$. Assumption 2 and Assumption 3 guarantee that (cf. (26)) $\Theta_h \subset \mathbb{B}_2^d(H\sqrt{d})$ and $\Omega = \mathbb{B}_2^d(BH\sqrt{d})$ for all $h$, where we use $\mathbb{B}_2^d(R)$ to denote the $\ell_2$ ball of radius $R$ in $\mathbb{R}^d$. Thus by Lemma 3 we have

$$\log \mathcal{N}(\Theta_h, \epsilon, \|\cdot\|_2) \leqslant \log \mathcal{N}\left(\mathbb{B}_2^d(H\sqrt{d}), \epsilon, \|\cdot\|_2\right) \leqslant d \log\left(1 + \frac{2H\sqrt{d}}{\epsilon}\right), \tag{51a}$$

$$\log \mathcal{N}(\Omega, \epsilon, \|\cdot\|_2) = \log \mathcal{N}\left(\mathbb{B}_2^d(BH\sqrt{d}), \epsilon, \|\cdot\|_2\right) \leqslant d \log\left(1 + \frac{2BH\sqrt{d}}{\epsilon}\right) \tag{51b}$$

for any $\epsilon > 0$. This suggests that for any $\epsilon > 0$, there exists an $\epsilon$-net $\Theta_{h,\epsilon} \subset \Theta_h$ and an $\epsilon$-net $\Omega_\epsilon \subset \Omega$ such that

$$\log |\Theta_{h,\epsilon}| \leqslant d \log \left(1 + \frac{2H\sqrt{d}}{\epsilon}\right), \quad \text{and} \quad \log |\Omega_\epsilon| \leqslant d \log \left(1 + \frac{2BH\sqrt{d}}{\epsilon}\right). \tag{52}$$

For any $f_h = f_{\theta,h} \in \mathcal{Q}_h$ with $\theta_h \in \Theta_h$, there exists $\theta_{h,\epsilon} \in \Theta_{h,\epsilon}$ such that $\|\theta_h - \theta_{h,\epsilon}\|_2 \leqslant \epsilon$, and we let $f_{h,\epsilon} := f_{\theta_{h,\epsilon}}$ and define

$$\mathcal{Q}_{h,\epsilon} := \{f_{h,\epsilon} : \theta_{h,\epsilon} \in \Theta_{h,\epsilon}\}, \qquad \mathcal{Q}_\epsilon = \prod_{h=1}^{H} \mathcal{Q}_{h,\epsilon} \tag{53}$$

In addition, for any $\pi_h \in \mathcal{P}_h$, there exists $\omega_h \in \Omega$ and $\omega_{h,\epsilon} \in \Omega_\epsilon$ such that $\|\omega_h - \omega_{h,\epsilon}\|_2 \leqslant \epsilon$, such that

$$\pi_h(a|s) = \frac{\exp(\phi_h(s,a)^\top \omega_h)}{\sum_{a' \in \mathcal{A}} \exp(\phi_h(s,a')^\top \omega_h)}, \qquad \pi_{h,\epsilon}(a|s) := \frac{\exp(\phi_h(s,a)^\top \omega_{h,\epsilon})}{\sum_{a' \in \mathcal{A}} \exp(\phi_h(s,a')^\top \omega_{h,\epsilon})}, \quad \forall (s,a) \in \mathcal{S} \times \mathcal{A}.$$

We define

$$\mathcal{P}_{h,\epsilon} := \{\pi_{h,\epsilon} : \omega_{h,\epsilon} \in \Omega_\epsilon\}, \qquad \mathcal{P}_\epsilon = \prod_{h=1}^{H} \mathcal{P}_{h,\epsilon}. \tag{54}$$

We claim that for any $f \in \mathcal{Q}$ and $\pi \in \mathcal{P}$, there exists $f_\epsilon \in \mathcal{Q}_\epsilon$ and $\pi_\epsilon \in \mathcal{P}_\epsilon$ such that

$$\left| X_{f_\epsilon, \pi_\epsilon, h}^t - X_{f, \pi, h}^t \right| \leqslant 24 H^2 \epsilon. \tag{55}$$

The proof of (55) is deferred to the end of this proof.

**Step 1.2: bounding the mean and variance.** Assumption 1 ensures $X_{f,\pi_h}^t$ is bounded:

$$\forall f \in \mathcal{Q}, \pi \in \mathcal{P}, h \in [H]: \quad |X_{f,\pi,h}^t| \leqslant 4H^2. \tag{56}$$

We now bound $\mathbb{E}_{s_{t,h+1} \sim \mathbb{P}_h(\cdot|s_{t,h}, a_{t,h})} \left[ X_{f,\pi,h}^t \right]$. Notice that

$$l_h(f, f, \xi, \pi)^2 = (l_h(f, f, \xi, \pi) - l_h(f, \mathbb{P}^\pi f, \xi, \pi) + l_h(f, \mathbb{P}^\pi f, \xi, \pi))^2$$

$$\stackrel{(47)}{=} \left( \mathbb{E}_{\substack{s' \sim \mathbb{P}_h(\cdot|s,a) \\ a' \sim \pi_{h+1}(\cdot|s')}} [l_h(f, f, \xi, \pi)] + l_h(f, \mathbb{P}^\pi f, \xi_h, \pi) \right)^2$$

$$= \left( \mathbb{E}_{\substack{s' \sim \mathbb{P}_h(\cdot|s,a) \\ a' \sim \pi_{h+1}(\cdot|s')}} [l_h(f, f, \xi, \pi)] \right)^2 + l_h(f, \mathbb{P}^\pi f, \xi, \pi)^2 + 2 \mathbb{E}_{\substack{s' \sim \mathbb{P}_h(\cdot|s,a) \\ a' \sim \pi_{h+1}(\cdot|s')}} [l_h(f, f, \xi, \pi)] l_h(f, \mathbb{P}^\pi f, \xi, \pi),$$

$$\tag{57}$$

where the expectation of the last term satisfies

$$\mathbb{E}_{\substack{s' \sim \mathbb{P}_h(\cdot|s,a) \\ a' \sim \pi_{h+1}(\cdot|s')}} \left[ \mathbb{E}_{\substack{s' \sim \mathbb{P}_h(\cdot|s,a) \\ a' \sim \pi_{h+1}(\cdot|s')}} [l_h(f, f, \xi, \pi)] l_h(f, \mathbb{P}^\pi f, \xi, \pi) \right]$$

$$= \mathbb{E}_{\substack{s' \sim \mathbb{P}_h(\cdot|s,a) \\ a' \sim \pi_{h+1}(\cdot|s')}} [l_h(f, f, \xi, \pi)] \, \mathbb{E}_{\substack{s' \sim \mathbb{P}_h(\cdot|s,a) \\ a' \sim \pi_{h+1}(\cdot|s')}} [l_h(f, \mathbb{P}^\pi f, \xi, \pi)] \stackrel{(46)}{=} 0. \tag{58}$$

Combining (48), (57) and (58), we have

$$\mathbb{E}_{s_{t,h+1} \sim \mathbb{P}_h(\cdot|s_{t,h}, a_{t,h})} \left[ X_{f,\pi,h}^t \right] = \left( \mathbb{E}_{\substack{s_{t,h+1} \sim \mathbb{P}_h(\cdot|s_{t,h}, a_{t,h}) \\ a' \sim \pi_{h+1}(\cdot|s_{t,h+1})}} [l_h(f, f, \xi_{t,h}, \pi)] \right)^2 \stackrel{(32)}{=} \ell_h(f, s_{t,h}, a_{t,h}, \pi). \tag{59}$$

Now we consider the martingale variance term. Define the filtration $\mathcal{F}_t := \sigma(\mathcal{D}_t)$ (the $\sigma$-algebra generated by the dataset $\mathcal{D}_t := \cup_{h=1}^{H} \mathcal{D}_{t,h}$). We have

$$\forall f \in \mathcal{Q}, h \in [H]: \quad \mathbb{E}\left[ X_{f,\pi,h}^t | \mathcal{F}_{t-1} \right] = \mathbb{E}\left[ \mathbb{E}_{s_{t,h+1} \sim \mathbb{P}_h(\cdot|s_{t,h}, a_{t,h})} \left[ X_{f,\pi,h}^t \right] | \mathcal{F}_{t-1} \right]$$

$$\stackrel{(59)}{=} \mathbb{E}_{(s_{t,h}, a_{t,h}) \sim d_{\rho,h}^{\pi_t}} [\ell_h(f, s_{t,h}, a_{t,h}, \pi)], \tag{60}$$

where we define $d_{\rho,h}^{\pi}$ to be the state-action visitation distribution at step $h$ and time $t$ under policy profile $\pi$ and initial state distribution $\rho$, i.e.,

$$d_{\rho,h}^{\pi}(s,a) := \mathbb{E}_{s_1 \sim \rho} \mathbb{P}^{\pi}(s_h = s, a_h = a | s_1). \tag{61}$$

Furthermore, we have

$$\mathsf{Var}\left[X_{f,\pi,h}^t | \mathcal{F}_{t-1}\right] \leqslant \mathbb{E}\left[\left(X_{f,\pi,h}^t\right)^2 | \mathcal{F}_{t-1}\right]$$

$$= \mathbb{E}\left[\left(\mathbb{E}_{a' \sim \pi_{h+1}(\cdot|s_{t,h+1})}\left[\left(r_h(s_{t,h},a_{t,h}) + f_{h+1}(s_{t,h+1},a') - f_h(s_{t,h},a_{t,h})\right)^2\right.\right.\right.$$

$$\left.\left.\left. - \left(f_{h+1}(s_{t,h+1},a') - \mathbb{E}_{\substack{s' \sim \mathbb{P}_h(\cdot|s_{t,h},a_{t,h}) \\ a' \sim \pi_{h+1}(\cdot|s')}}\left[f_{h+1}(s',a')\right]\right)^2\right]\right)^2 \middle| \mathcal{F}_{t-1}\right]$$

$$\leqslant \mathbb{E}\left[\left(r_h(s_{t,h},a_{t,h}) + 2f_{h+1}(s_{t,h+1},a') - f_h(s_{t,h},a_{t,h}) - \mathbb{E}_{\substack{s' \sim \mathbb{P}_h(\cdot|s_{t,h},a_{t,h}) \\ a' \sim \pi_{h+1}(\cdot|s')}}\left[f_{h+1}(s',a')\right]\right)^2\right.$$

$$\left. \cdot \left(r_h(s_{t,h},a_{t,h}) + \mathbb{E}_{\substack{s' \sim \mathbb{P}_h(\cdot|s_{t,h},a_{t,h}) \\ a' \sim \pi_{h+1}(\cdot|s')}}\left[f_{h+1}(s',a')\right] - f_h(s_{t,h},a_{t,h})\right)^2 \middle| \mathcal{F}_{t-1}\right]$$

$$\leqslant 16H^2 \mathbb{E}_{(s_{t,h},a_{t,h}) \sim d_{\rho,h}^{\pi_t}}\left[\ell_h(f,s_{t,h},a_{t,h},\pi)\right], \quad \forall f \in \mathcal{Q}, \tag{62}$$

where the first equality follows from (45) and (46), and the second inequality follows from Jenson's inequality.

**Step 1.3: applying Freedman's inequality and finishing up.** By Lemma 2, (56), (60) and (62), and noticing that $\ell_h(f,s,a,\pi)$ is only related to $f_h, f_{h+1}$ and $\pi_{h+1}$, we have with probability at least $1-\delta$, for all $t \in [T]$, $h \in [H]$, $f_\epsilon \in \mathcal{Q}_\epsilon$ and $\pi_\epsilon \in \mathcal{P}_\epsilon$,

$$\sum_{i=1}^{t-1} \mathbb{E}_{(s_{i,h},a_{i,h}) \sim d_{\rho,h}^{\pi_i}}\left[\ell_h(f_\epsilon,s_{i,h},a_{i,h},\pi_\epsilon)\right] - \sum_{i=1}^{t-1} X_{f_\epsilon,\pi_\epsilon,h}^i$$

$$\leqslant \frac{1}{2}\sum_{i=1}^{t-1} \mathbb{E}_{(s_{i,h},a_{i,h}) \sim d_{\rho,h}^{\pi_i}}\left[\ell_h(f_\epsilon,s_{i,h},a_{i,h},\pi_\epsilon)\right] + C_1 H^2 \log(TH|\Theta_{h,\epsilon}||\Theta_{h+1,\epsilon}||\Omega_\epsilon|/\delta)$$

$$\overset{(52)}{\leqslant} \frac{1}{2}\sum_{i=1}^{t-1} \mathbb{E}_{(s_{i,h},a_{i,h}) \sim d_{\rho,h}^{\pi_i}}\left[\ell_h(f_\epsilon,s_{i,h},a_{i,h},\pi_\epsilon)\right] + C_1' H^2 \left(d\log\left(\frac{BHd}{\epsilon}\right) + \log(T/\delta)\right), \quad (63)$$

where $C_1, C_1' > 0$ are absolute constants. From (63) we deduce that for all $t \in [T]$, $f_\epsilon \in \mathcal{Q}_\epsilon$, and $\pi_\epsilon \in \mathcal{P}_\epsilon$, we have with probability at least $1-\delta$,

$$-\sum_{i=1}^{t-1}\sum_{h=1}^{H} X_{f_\epsilon,\pi_\epsilon,h}^i \leqslant -\frac{1}{2}\sum_{i=1}^{t-1}\sum_{h=1}^{H} \mathbb{E}_{(s_{i,h},a_{i,h}) \sim d_{\rho,h}^{\pi_i}}\left[\ell_h(f_\epsilon,s_{i,h},a_{i,h},\pi_\epsilon)\right] + C_1' H^3 \left(d\log\left(\frac{BHd}{\epsilon}\right) + \log(T/\delta)\right). \tag{64}$$

Note that for any $t \in [T]$ and $h \in [H]$, there exist $\theta_{t,h} \in \Theta_h$ and $\omega_{t,h} \in \Omega$ such that $f_{t,h} = f_{\theta_{t,h}} \in \mathcal{Q}_h$ and $\pi_{t,h} = \pi_{\omega_{t,h}} \in \mathcal{P}_h$. We can choose $\theta_{t,h,\epsilon} \in \Theta_{h,\epsilon}$ and $\omega_{t,h,\epsilon} \in \Omega_\epsilon$ such that $\|\theta_{t,h} - \theta_{t,h,\epsilon}\|_2 \leqslant \epsilon$ and $\|\omega_{t,h} - \omega_{t,h,\epsilon}\|_2 \leqslant \epsilon$. We let $f_{t,\epsilon} := \{f_{\theta_{t,h,\epsilon}}\}_{h \in [H]} \in \mathcal{Q}_\epsilon$ and $\pi_{t,\epsilon} := \{\pi_{\omega_{t,h,\epsilon}}\}_{h \in [H]} \in \mathcal{P}_\epsilon$. Then by (64) we have for all $t \in [T]$,

$$-\mathcal{L}_t(f_t,\pi_t)$$

$$\overset{(49)}{\leqslant} -\sum_{i=1}^{t-1}\sum_{h=1}^{H} X_{f_t,\pi_t,h}^i$$

$$\overset{(55)}{\leqslant} -\sum_{i=1}^{t-1}\sum_{h=1}^{H} X_{f_{t,\epsilon},\pi_{t,\epsilon},h}^i + 24H^3\epsilon T$$

$$\overset{(64)}{\leqslant} -\frac{1}{2}\sum_{i=1}^{t-1}\sum_{h=1}^{H} \mathbb{E}_{(s_{i,h},a_{i,h}) \sim d_{\rho,h}^{\pi_i}}\left[\ell_h(f_{t,\epsilon},s_{i,h},a_{i,h},\pi_{t,\epsilon})\right] + C_1' H^3 \left(d\log\left(\frac{BHd}{\epsilon}\right) + \log(T/\delta)\right) + 24H^3\epsilon T$$

$$
\leqslant -\frac{1}{2} \sum_{i=1}^{t-1} \sum_{h=1}^{H} \mathbb{E}_{(s_{i,h},a_{i,h})\sim d_{\rho,h}^{\pi_i}} \left[ \ell_h(f_t, s_{i,h}, a_{i,h}, \pi_t) \right] + C_1' H^3 \left( d \log \left( \frac{BHd}{\epsilon} \right) + \log(T/\delta) \right) + 36 H^3 \epsilon T,
$$

(65)

where the last line follows from (55) and (59).

**Step 2: bounding $\mathcal{L}_t(f^\star, \widetilde{\pi}^\star)$.**   For any $f \in \mathcal{Q}$ and $t \in [T]$, we define

$$
Y_{f,h}^t := \mathbb{E}_{a'\sim\widetilde{\pi}_{h+1}^\star(\cdot|s_{t,h})} \left[ l_h(f^\star, f, \xi_{t,h}, \widetilde{\pi}^\star)^2 - l_h(f^\star, \widetilde{f}^\star, \xi_{t,h}, \widetilde{\pi}^\star)^2 \right] \qquad \text{where} \quad \widetilde{f}^\star := \mathbb{P}^{\widetilde{\pi}^\star} f^\star.
$$

(66)

Note that for any tuple $\xi = (s, a, s')$, we have

$$
\left| l_h(f^\star, f^\star, \xi, \widetilde{\pi}^\star)^2 - l_h(f^\star, \widetilde{f}^\star, \xi, \widetilde{\pi}^\star)^2 \right|
$$
$$
= \left| l_h(f^\star, f^\star, \xi, \widetilde{\pi}^\star) + l_h(f^\star, \widetilde{f}^\star, \xi, \widetilde{\pi}^\star) \right| \left| l_h(f^\star, f^\star, \xi, \widetilde{\pi}^\star) - l_h(f^\star, \widetilde{f}^\star, \xi, \widetilde{\pi}^\star) \right|
$$
$$
\leqslant 4H \left| \mathbb{E}_{\substack{s'\sim\mathbb{P}_h(\cdot|s,a) \\ a'\sim\widetilde{\pi}_{h+1}^\star(\cdot|s')}} \left[ l_h(f^\star, f^\star, \xi, \widetilde{\pi}^\star) \right] \right|,
$$

(67)

where the last line follows from (47). Furthermore, we have

$$
\mathbb{E}_{\substack{s'\sim\mathbb{P}_h(\cdot|s,a) \\ a'\sim\widetilde{\pi}_{h+1}^\star(\cdot|s')}} \left[ l_h(f^\star, f^\star, \xi, \widetilde{\pi}^\star) \right] \overset{(45)}{=} \mathbb{E}_{\substack{s'\sim\mathbb{P}_h(\cdot|s,a) \\ a'\sim\widetilde{\pi}_{h+1}^\star(\cdot|s')}} \left[ r_h(s,a) + f_{h+1}^\star(s',a') - f_h^\star(s,a) \right]
$$
$$
= r_h(s,a) + \mathbb{E}_{s'\sim\mathbb{P}_h(\cdot|s,a)} \left[ V_{f^\star,h+1}^{\widetilde{\pi}^\star}(s') \right] - f_h^\star(s,a)
$$
$$
= \mathbb{E}_{s'\sim\mathbb{P}_h(\cdot|s,a)} \left[ V_{f^\star,h+1}^{\widetilde{\pi}^\star}(s') \right] - \mathbb{E}_{s'\sim\mathbb{P}_h(\cdot|s,a)} \left[ V_{h+1}^\star(s') \right], \quad (68)
$$

where the last line follows from Bellman's optimality equation:

$$
r_h(s,a) + \mathbb{E}_{s'\sim\mathbb{P}_h(\cdot|s,a)} \left[ V_{h+1}^\star(s') \right] - f_h^\star(s,a) = 0.
$$

Note that by Lemma 8, we have

$$
\mathbb{E}_{s'\sim\mathbb{P}_h(\cdot|s,a)} \left[ V_{h+1}^\star(s') \right] - \frac{\log|\mathcal{A}|}{B} \leqslant \mathbb{E}_{s'\sim\mathbb{P}_h(\cdot|s,a)} \left[ V_{f^\star,h+1}^{\widetilde{\pi}^\star}(s') \right] \leqslant \mathbb{E}_{s'\sim\mathbb{P}_h(\cdot|s,a)} \left[ V_{h+1}^\star(s') \right].
$$

(69)

Plugging the above inequality into (67) and (68) leads to

$$
\left| l_h(f^\star, f^\star, \xi, \widetilde{\pi}^\star)^2 - l_h(f^\star, \widetilde{f}^\star, \xi, \widetilde{\pi}^\star)^2 \right| \leqslant 4H \frac{\log|\mathcal{A}|}{B}.
$$

(70)

The above bounds (70) and (50) imply that

$$
\mathcal{L}_{t,h}(f^\star, \widetilde{\pi}^\star) = \sum_{i=1}^{t-1} \mathbb{E}_{a'\sim\widetilde{\pi}_{h+1}^\star(\cdot|s_i')} l_h(f^\star, f^\star, \xi_{i,h}, \widetilde{\pi}^\star)^2 - \inf_{g\in\mathcal{Q}} \sum_{i=1}^{t-1} \mathbb{E}_{a'\sim\widetilde{\pi}_{h+1}^\star(\cdot|s_i')} l_h(f^\star, g, \xi_{i,h}, \widetilde{\pi}^\star)^2
$$
$$
\leqslant \sup_{f\in\mathcal{Q}} \sum_{i=1}^{t-1} \left( -Y_{f,h}^i \right) + \frac{4HT \log|\mathcal{A}|}{B},
$$

(71)

where we also use the definitions of $Y_{f,h}^t$ (c.f. (66)) and $\widetilde{f}^\star$ (c.f. (66)). Thus to bound $\mathcal{L}_t(f^\star, \widetilde{\pi}^\star)$, below we bound the sum $\sum_{i=1}^{t-1} Y_{f,h}^i$ for any $f \in \mathcal{Q}$, $t \in [T]$ and $h \in [H]$ by applying Freedman's inequality and the covering argument. By a similar argument as earlier, we have for any $f \in \mathcal{Q}$, there exists $f_\epsilon \in \mathcal{Q}_\epsilon$ such that

$$
Y_{f_\epsilon,h}^t - Y_{f,h}^t \leqslant 4H\epsilon,
$$

(72)

whose proof is deferred to the end. We next compute the key quantities required to apply Freedman's inequality.

- Repeating a similar derivation of (59), we have

$$\mathbb{E}_{s'\sim\mathbb{P}_h(\cdot|s,a)}\left[Y^t_{f,h}\right] = \left(\mathbb{E}_{\substack{s'\sim\mathbb{P}_h(\cdot|s,a)\\a'\sim\widetilde{\pi}^\star_{h+1}(\cdot|s')}}\left[l_h(f^\star,f,\xi_t,\widetilde{\pi}^\star)\right]\right)^2,\tag{73}$$

which implies

$$\forall f\in\mathcal{Q}:\quad \mathbb{E}\left[Y^t_{f,h}|\mathcal{F}_{t-1}\right]=\mathbb{E}_{(s_{t,h},a_{t,h})\sim d^{\pi_t}_{\rho,h}}\left[\left(\mathbb{E}_{\substack{s_{t,h+1}\sim\mathbb{P}_h(\cdot|s_{t,h},a_{t,h})\\a'\sim\widetilde{\pi}^\star_{h+1}(\cdot|s_{t,h+1})}}\left[l_h(f^\star,f,\xi_{t,h},\widetilde{\pi}^\star)\right]\right)^2\right].\tag{74}$$

- We have

$$\mathsf{Var}\left[Y^t_{f,h}|\mathcal{F}_{t-1}\right]$$
$$\leqslant \mathbb{E}\left[\left(Y^t_{f,h}\right)^2|\mathcal{F}_{t-1}\right]$$
$$= \mathbb{E}\left[\left(\mathbb{E}_{a'\sim\widetilde{\pi}^\star_{h+1}(\cdot|s_{t,h})}\left[\left(r_h(s_{t,h},a_{t,h})+f^\star_{h+1}(s_{t,h+1},a')-f_h(s_{t,h},a_{t,h})\right)^2\right.\right.\right.$$
$$\left.\left.\left.-\left(f^\star_{h+1}(s_{t,h+1},a')-\mathbb{E}_{\substack{s_{t,h+1}\sim\mathbb{P}_h(\cdot|s_{t,h},a_{t,h})\\a'\sim\widetilde{\pi}^\star_{h+1}(\cdot|s_{t,h+1})}}\left[f^\star_{h+1}(s_{t,h+1},a')\right]\right)^2\right]\right)^2\Bigg|\mathcal{F}_{t-1}\right]$$
$$\leqslant \mathbb{E}\left[\left(r_h(s_{t,h},a_{t,h})+2f^\star_{h+1}(s_{t,h+1},a')-f_h(s_{t,h},a_{t,h})-\mathbb{E}_{\substack{s_{t,h+1}\sim\mathbb{P}_h(\cdot|s_{t,h},a_{t,h})\\a'\sim\widetilde{\pi}^\star_{h+1}(\cdot|s_{t,h+1})}}\left[f^\star_{h+1}(s_{t,h+1},a')\right]\right)^2\right.$$
$$\left.\cdot\left(r_h(s_{t,h},a_{t,h})+\mathbb{E}_{\substack{s_{t,h+1}\sim\mathbb{P}_h(\cdot|s_{t,h},a_{t,h})\\a'\sim\widetilde{\pi}^\star_{h+1}(\cdot|s_{t,h+1})}}\left[f^\star_{h+1}(s_{t,h+1},a')\right]-f_h(s_{t,h},a_{t,h})\right)^2\Bigg|\mathcal{F}_{t-1}\right]$$
$$\leqslant 16H^2\mathbb{E}_{(s_{t,h},a_{t,h})\sim d^{\pi_t}_{\rho,h}}\left[\left(\mathbb{E}_{\substack{s_{t,h+1}\sim\mathbb{P}_h(\cdot|s_{t,h},a_{t,h})\\a'\sim\widetilde{\pi}^\star_{h+1}(\cdot|s_{t,h+1})}}\left[l_h(f^\star,f,\xi_{t,h},\widetilde{\pi}^\star)\right]\right)^2\right],\tag{75}$$

where the first line uses (by (46))

$$l_h(f^\star,\widetilde{f}^\star,\xi_{t,h},\pi^\star)=f^\star_{h+1}(s_{t,h+1},a')-\mathbb{E}_{\substack{s_{t,h+1}\sim\mathbb{P}_h(\cdot|s_{t,h},a_{t,h})\\a'\sim\widetilde{\pi}^\star_{h+1}(\cdot|s_{t,h+1})}}\left[f^\star_{h+1}(s_{t,h+1},a')\right]\tag{76}$$

where $a'\sim\widetilde{\pi}^\star_{h+1}(\cdot|s_{t,h+1})$ and the second inequality uses Jenson's inequality.
- Last but not least, it's easy to verify that

$$|Y^t_f|\leqslant 4H^2.\tag{77}$$

Invoking Lemma 2, and setting $\eta$ as

$$\eta=\min\left\{\frac{1}{4H^2},\sqrt{\frac{\log(|\Theta_{h,\epsilon}||\Theta_{h+1,\epsilon}|HT/\delta)}{\sum_{i=1}^{t-1}\mathsf{Var}\left[Y^i_{f,h}|\mathcal{F}_{i-1}\right]}}\right\},$$

we have with probability at least $1-\delta$, for all $f_\epsilon\in\mathcal{Q}_\epsilon,t\in[T],h\in[H]$,

$$\sum_{i=1}^{t-1}\left(-Y^i_{f_\epsilon,h}+\mathbb{E}_{(s_{i,h},a_{i,h})\sim d^{\pi_i}_{\rho,h}}\left[\left(\mathbb{E}_{\substack{s_{i,h+1}\sim\mathbb{P}_h(\cdot|s_{i,h},a_{i,h})\\a'\sim\widetilde{\pi}^\star_{h+1}(\cdot|s_{i,h+1})}}\left[l_h(f^\star,f_\epsilon,\xi_{i,h},\widetilde{\pi}^\star)\right]\right)^2\right]\right)$$
$$\lesssim H\sqrt{\log(|\Theta_{h,\epsilon}||\Theta_{h+1,\epsilon}|HT/\delta)\sum_{i=1}^{t-1}\mathbb{E}_{(s_{i,h},a_{i,h})\sim d^{\pi_i}_{\rho,h}}\left[\left(\mathbb{E}_{\substack{s_{i,h+1}\sim\mathbb{P}_h(\cdot|s_{i,h},a_{i,h})\\a'\sim\widetilde{\pi}^\star_{h+1}(\cdot|s_{i,h+1})}}\left[l_h(f^\star,f_\epsilon,\xi_{i,h},\widetilde{\pi}^\star)\right]\right)^2\right]}$$
$$+H^2\log(|\Theta_{h,\epsilon}||\Theta_{h+1,\epsilon}|HT/\delta).\tag{78}$$

Reorganizing the above inequality, we have for any $f_\epsilon\in\mathcal{Q}_\epsilon,t\in[T]$:

$$\sum_{i=1}^{t-1}\left(-Y^i_{f_\epsilon,h}\right)$$

$$\lesssim -\sum_{i=1}^{t-1} \mathbb{E}_{(s_{i,h},a_{i,h})\sim d_{\rho,h}^{\pi_i}} \left[ \left( \mathbb{E}_{\substack{s_{i,h+1}\sim \mathbb{P}_h(\cdot|s_{i,h},a_{i,h}) \\ a'\sim \tilde{\pi}_{h+1}^\star(\cdot|s_{i,h+1})}} [l_h(f^\star, f_\epsilon, \xi_{i,h}, \tilde{\pi}^\star)] \right)^2 \right] + H^2 \log(|\Theta_{h,\epsilon}||\Theta_{h+1,\epsilon}|HT/\delta)$$

$$+ H\sqrt{\log(|\Theta_{h,\epsilon}||\Theta_{h+1,\epsilon}|HT/\delta) \sum_{i=1}^{t-1} \mathbb{E}_{(s_{i,h},a_{i,h})\sim d_{\rho,h}^{\pi_i}} \left[ \left( \mathbb{E}_{\substack{s_{i,h+1}\sim \mathbb{P}_h(\cdot|s_{i,h},a_{i,h}) \\ a'\sim \tilde{\pi}_{h+1}^\star(\cdot|s_{i,h+1})}} [l_h(f^\star, f_\epsilon, \xi_{i,h}, \tilde{\pi}^\star)] \right)^2 \right]}$$

$$\lesssim H^2 \log(|\Theta_{h,\epsilon}||\Theta_{h+1,\epsilon}|HT/\delta), \tag{79}$$

where the last line makes use of the fact that $-x^2 + bx \leqslant b^2/4$.

Combining (79) and (72), we have with probability at least $1 - \delta$, for any $t \in [T]$ and $f \in \mathcal{Q}$,

$$\sum_{i=1}^{t-1} \sum_{h=1}^{H} \left(-Y_{f,h}^i\right) \leqslant \sum_{i=1}^{t-1} \sum_{h=1}^{H} \left(-Y_{f_\epsilon,h}^i\right) + 4H^2\epsilon T$$

$$\overset{(52)}{\leqslant} C_2 H^3 \left( d\log\left(\frac{Hd}{\epsilon}\right) + \log(T/\delta) \right) + 4H^2\epsilon T, \tag{80}$$

where $C_2 > 0$ is an absolute constant. Plugging this into (71), we have

$$\mathcal{L}_t(f^\star, \tilde{\pi}^\star) \leqslant C_2 H^3 \left( d\log\left(\frac{Hd}{\epsilon}\right) + \log(T/\delta) \right) + 4H^2\epsilon T + \frac{4H^2 T \log|\mathcal{A}|}{B}. \tag{81}$$

**Step 3: combining the two bounds.** Combining (65) and (81), we have for any $t \in [T]$,

$$\mathcal{L}_t(f^\star, \tilde{\pi}^\star) - \mathcal{L}_t(f_t, \pi_t) \leqslant -\frac{1}{2} \sum_{i=1}^{t-1} \sum_{h=1}^{H} \mathbb{E}_{(s_{i,h},a_{i,h})\sim d_{\rho,h}^{\pi_i}} [\ell_h(f_t, s_{i,h}, a_{i,h}, \pi_t)]$$

$$+ CH^3 \left( d\log\left(\frac{BHd}{\epsilon}\right) + \log(T/\delta) + \epsilon T + \frac{T\log|\mathcal{A}|}{BH} \right) \tag{82}$$

for some absolute constant $C > 0$. Letting $\epsilon = \frac{1}{T}$, we obtain the desired result.

**Proof of** (55) **and** (72). By Assumption 1, we have

$$\forall (s,a) \in \mathcal{S} \times \mathcal{A}: \quad |f_h(s,a) - f_{h,\epsilon}(s,a)| \leqslant \|\phi_h(s,a)\|_2 \|\theta_h - \theta_{h,\epsilon}\|_2 \leqslant \epsilon, \tag{83}$$

and thus for any $f \in \mathcal{Q}$ and $\pi \in \mathcal{P}$, we have

$$\left| X_{f_\epsilon,\pi,h}^t - X_{f,\pi,h}^t \right|$$

$$= \left| \mathbb{E}_{a'\sim\pi_{h+1}(\cdot|s_{t,h+1})} \left[ (r_h(s_{t,h},a_{t,h}) + f_{h+1,\epsilon}(s_{t,h+1},a') - f_{h,\epsilon}(s_{t,h},a_{t,h}))^2 \right. \right.$$

$$- \left( f_{h+1,\epsilon}(s_{t,h+1},a') - \mathbb{E}_{\substack{s'\sim\mathbb{P}_h(\cdot|s_{t,h},a_{t,h}) \\ a'\sim\pi_{h+1}(\cdot|s')}} [f_{h+1}(s',a')] \right)^2 \right]$$

$$- \mathbb{E}_{a'\sim\pi_{h+1}(\cdot|s_{t,h+1})} \left[ (r_h(s_{t,h},a_{t,h}) + f_{h+1}(s_{t,h+1},a') - f_h(s_{t,h},a_{t,h}))^2 \right.$$

$$\left. \left. - \left( f_{h+1}(s_{t,h+1},a') - \mathbb{E}_{\substack{s'\sim\mathbb{P}_h(\cdot|s_{t,h},a_{t,h}) \\ a'\sim\pi_{h+1}(\cdot|s')}} [f_{h+1}(s',a')] \right)^2 \right] \right|$$

$$= \left| \mathbb{E}_{a'\sim\pi_{h+1}(\cdot|s_{t,h+1})} \left[ (2r_h(s_{t,h},a_{t,h}) + f_{h+1,\epsilon}(s_{t,h+1},a') - f_{h,\epsilon}(s_{t,h},a_{t,h}) + f_{h+1}(s_{t,h+1},a') - f_h(s_{t,h},a_{t,h})) \right. \right.$$

$$\cdot \left( f_{h+1,\epsilon}(s_{t,h+1},a') - f_{h+1}(s_{t,h+1},a') - \mathbb{E}_{\substack{s'\sim\mathbb{P}_h(\cdot|s_{t,h},a_{t,h}) \\ a'\sim\pi_{h+1}(\cdot|s')}} [f_{h+1}(s',a') - f_{h+1,\epsilon}(s',a')] \right) \right]$$

$$+ \mathbb{E}_{a'\sim\pi_{h+1}(\cdot|s_{t,h+1})} \left[ \left( f_{h+1}(s_{t,h+1},a') - f_{h+1,\epsilon}(s_{t,h+1},a') - \mathbb{E}_{\substack{s'\sim\mathbb{P}_h(\cdot|s_{t,h},a_{t,h}) \\ a'\sim\pi_{h+1}(\cdot|s')}} [f_{h+1}(s',a') - f_{h+1,\epsilon}(s',a')] \right) \right.$$

$$\left. \left. \cdot \left( f_{h+1}(s_{t,h+1},a') - \mathbb{E}_{\substack{s'\sim\mathbb{P}_h(\cdot|s_{t,h},a_{t,h}) \\ a'\sim\pi_{h+1}(\cdot|s')}} [f_{h+1}(s',a')] + f_{h+1,\epsilon}(s_{t,h+1},a') - \mathbb{E}_{\substack{s'\sim\mathbb{P}_h(\cdot|s_{t,h},a_{t,h}) \\ a'\sim\pi_{h+1}(\cdot|s')}} [f_{h+1,\epsilon}(s',a')] \right) \right] \right|$$

$$\leqslant 8H\epsilon + 8H\epsilon = 16H\epsilon, \tag{84}$$

where in the last inequality we use (83).

Similarly, by Lemma 6, we have

$$\forall s \in \mathcal{S}, h \in [H]: \quad \|\pi_h(\cdot|s) - \pi_{h,\epsilon}(\cdot|s)\|_1 \leqslant 2 \max_{s,a} \|\phi_h(s,a)\|_2 \|\omega_h - \omega_{h,\epsilon}\|_2 \leqslant 2\epsilon. \tag{85}$$

Therefore, we have

$$
\left| X_{f,\pi_\epsilon,h}^t - X_{f,\pi,h}^t \right| = \left| \mathbb{E}_{a'\sim\pi_{h+1}(\cdot|s_{t,h+1})} \left[ \left( r_h(s_{t,h}, a_{t,h}) + f_{h+1}(s_{t,h+1}, a') - f_h(s_{t,h}, a_{t,h}) \right)^2 \right. \right.
$$
$$
\left. - \left( f_{h+1}(s_{t,h+1}, a') - \mathbb{E}_{\substack{s'\sim\mathbb{P}_h(\cdot|s_{t,h},a_{t,h})\\a'\sim\pi_{h+1}(\cdot|s')}} [f_{h+1}(s', a')] \right)^2 \right]
$$
$$
- \mathbb{E}_{a'\sim\pi_{h+1,\epsilon}(\cdot|s_{t,h+1})} \left[ \left( r_h(s_{t,h}, a_{t,h}) + f_{h+1}(s_{t,h+1}, a') - f_h(s_{t,h}, a_{t,h}) \right)^2 \right.
$$
$$
\left. \left. - \left( f_{h+1}(s_{t,h+1}, a') - \mathbb{E}_{\substack{s'\sim\mathbb{P}_h(\cdot|s_{t,h},a_{t,h})\\a'\sim\pi_{h+1,\epsilon}(\cdot|s')}} [f_{h+1}(s', a')] \right)^2 \right] \right|
$$
$$
\leqslant 4H^2 \|\pi_{h+1}(\cdot|s_{t,h+1}) - \pi_{h+1,\epsilon}(\cdot|s_{t,h+1})\|_1 \overset{(85)}{\leqslant} 8H^2\epsilon, \tag{86}
$$

where the first inequality follows from Hölder's inequality and the fact that

$$
\left| (r_h(s,a) + f_{h+1}(s', a') - f_h(s,a))^2 - \left( f_{h+1}(s', a') - \mathbb{E}_{\substack{s'\sim\mathbb{P}_h(\cdot|s,a)\\a'\sim\pi_{h+1}(\cdot|s')}} [f_{h+1}(s', a')] \right)^2 \right| \leqslant 4H^2
$$

for all $(s,a) \in \mathcal{S} \times \mathcal{A}$, $f \in \mathcal{Q}$ and $\pi \in \mathcal{P}$.

Combining (84) and (86), we have the desired bound in (55):

$$
\left| X_{f_\epsilon,\pi_\epsilon,h}^t - X_{f,\pi,h}^t \right| \leqslant \left| X_{f_\epsilon,\pi_\epsilon,h}^t - X_{f_\epsilon,\pi,h}^t \right| + \left| X_{f_\epsilon,\pi,h}^t - X_{f,\pi,h}^t \right| \leqslant 16H\epsilon + 8H^2\epsilon = 24H^2\epsilon.
$$

Similarly, we have (72) follows by

$$
Y_{f_\epsilon,h}^t - Y_{f,h}^t = \mathbb{E}_{a'\sim\widetilde{\pi}_{h+1}^\star(\cdot|s_{t,h})} \left[ \left( r_h(s_{t,h}, a_{t,h}) + f_{h+1}^\star(s_{t,h+1}, a') - f_{\epsilon,h}(s_{t,h}, a_{t,h}) \right)^2 \right.
$$
$$
\left. - \left( r_h(s_{t,h}, a_{t,h}) + f_{h+1}^\star(s_{t,h+1}, a') - f_h(s_{t,h}, a_{t,h}) \right)^2 \right]
$$
$$
= \mathbb{E}_{a'\sim\widetilde{\pi}_{h+1}^\star(\cdot|s_{t,h})} \left[ \left( 2r_h(s_{t,h}, a_{t,h}) + 2f_{h+1}^\star(s_{t,h+1}, a') - f_{\epsilon,h}(s_{t,h}, a_{t,h}) - f_h(s_{t,h}, a_{t,h}) \right) \right.
$$
$$
\left. \cdot \left( f_h(s_{t,h}, a_{t,h}) - f_{\epsilon,h}(s_{t,h}, a_{t,h}) \right) \right]
$$
$$
\leqslant 4H\epsilon,
$$

where the last inequality uses (83).

### B.2.4   Proof of Lemma 10

First note that for any policy profile $\pi \in \Pi^H$, any $f \in \mathcal{Q}$ and $h \in [H]$, we have (note that $V_{f,H+1} = 0$)

$$
V_{f_h}^\pi(\rho) = \mathbb{E}_{\substack{s_1\sim\rho, a_h\sim\pi_{h+1}(\cdot|s_h)\\s_{h+1}\sim\mathbb{P}_h(\cdot|s_h,a_h), \forall h\in[H]}} \left[ \sum_{h=1}^H \left( V_{f,h}^\pi(s_h) - V_{f,h+1}^\pi(s_{h+1}) \right) \right]
$$
$$
= \mathbb{E}_{\substack{s_1\sim\rho, a_h\sim\pi_{h+1}(\cdot|s_h)\\s_{h+1}\sim\mathbb{P}_h(\cdot|s_h,a_h), \forall h\in[H]}} \left[ \sum_{h=1}^H \left( Q_{f,h}(s_h, a_h) - V_{f,h+1}^\pi(s_{h+1}) \right) \right], \tag{87}
$$

and

$$
V^\pi(\rho) = \mathbb{E}_{\substack{s_1\sim\rho, a_h\sim\pi(\cdot|s_h)\\s_{h+1}\sim\mathbb{P}_h(\cdot|s_h,a_h), \forall h\in[H]}} \left[ \sum_{h=1}^H r_h(s_h, a_h) \right]. \tag{88}
$$

The above two expressions (87) and (88) together give that

$$V_f^\pi(\rho) - V^\pi(\rho) = \mathbb{E}_{\substack{s_1\sim\rho, a_h\sim\pi_{h+1}(\cdot|s_h) \\ s_{h+1}\sim\mathbb{P}_h(\cdot|s_h,a_h), \forall h\in[H]}} \left[ \sum_{h=1}^{H} \left( Q_{f,h}(s_h,a_h) - r_h(s_h,a_h) - V_{f,h+1}^\pi(s_{h+1}) \right) \right]$$

$$= \sum_{h=1}^{H} \mathbb{E}_{(s_h,a_h)\sim d_{\rho,h}^\pi} \left[ \underbrace{\left( Q_{f,h}(s_h,a_h) - r_h(s_h,a_h) - \mathbb{P}_h V_f^\pi(s_h,a_h) \right)}_{=:\mathcal{E}_h(f,s_h,a_h,\pi)} \right], \qquad (89)$$

where we define

$$\mathbb{P}_h V_f^\pi(s,a) := \mathbb{E}_{s'\sim\mathbb{P}_h(\cdot|s,a)} \left[ V_{f,h+1}^\pi(s') \right], \qquad (90)$$

and

$$\mathcal{E}_h(f,s,a,\pi) := Q_{f,h}(s,a) - r_h(s,a) - \mathbb{P}_h V_f^\pi(s,a). \qquad (91)$$

By Assumption 1, for any $f \in \mathcal{Q}$, there exists $\theta_f \in \Theta$ such that $f_h(s,a) = \langle \theta_{f,h}, \phi_h(s,a) \rangle$. Thus we have

$$\mathcal{E}_h(f,s,a,\pi) = \phi_h(s,a)^\top \underbrace{\left( \theta_{f,h} - \zeta_h - \int_{\mathcal{S}} V_{f,h+1}^\pi(s') d\mu_h(s') \right)}_{=:W_h(f,\pi)}, \qquad (92)$$

where $W_h(f,\pi)$ satisfies

$$\forall f \in \mathcal{Q}, \pi \in \Pi, h \in [H]: \quad \|W_h(f,\pi)\|_2 \leqslant 2H\sqrt{d} \qquad (93)$$

under Assumption 1. We define

$$x_h(\pi) := \mathbb{E}_{(s,a)\sim d_{\rho,h}^\pi} \left[ \phi_h(s,a) \right]. \qquad (94)$$

Then we have

$$V_f^\pi(\rho) - V^\pi(\rho) = \sum_{h=1}^{H} \mathbb{E}_{(s,a)\sim d_{\rho,h}^\pi} \left[ \mathcal{E}_h(f,s,a,\pi) \right] = \sum_{h=1}^{H} \langle x_h(\pi), W_h(f,\pi) \rangle. \qquad (95)$$

For all $t \in [T]$ and $h \in [H]$, we define

$$\Lambda_{t,h}(\lambda) := \lambda I_d + \sum_{i=1}^{t-1} x_h(\pi_i) x_h(\pi_i)^\top, \ \forall\lambda > 0, \qquad (96)$$

where $I_d$ is the $d \times d$ identity matrix. Then by Lemma 4, we have

$$\sum_{i=1}^{t} \min \left\{ \|x_h(\pi_i)\|_{\Lambda_{i,h}(\lambda)^{-1}}, 1 \right\} \leqslant 2\log \left( \det \left( I_d + \frac{1}{\lambda} \sum_{i=1}^{t-1} x_h(\pi_i) x_h(\pi_i)^\top \right) \right). \qquad (97)$$

Further, we could use Lemma 5 to bound the last term in (97), and obtain

$$\forall t \in [T]: \quad \sum_{i=1}^{t} \min \left\{ \|x_h(\pi_i)\|_{\Lambda_{i,h}(\lambda)^{-1}}, 1 \right\} \leqslant 2d(\lambda), \qquad (98)$$

where in the last line, we use the definition of $d(\lambda)$ (c.f. (35)) and the fact that

$$\|x_h(\pi)\|_2 \leqslant 1, \qquad (99)$$

which is ensured by Assumption 1.

Observe that

$$\sum_{t=1}^{T} \left| V_{f_t}^{\pi_t}(\rho) - V^{\pi_t}(\rho) \right| \overset{(95)}{\leqslant} \sum_{t=1}^{T} \sum_{h=1}^{H} |\langle x_h(\pi_t), W_h(f_t,\pi_t) \rangle|$$

$$= \underbrace{\sum_{t=1}^{T}\sum_{h=1}^{H}|\langle x_h(\pi_t), W_h(f_t, \pi_t)\rangle| \, \mathbf{1}\left\{\|x_h(\pi_t)\|_{\Lambda_{t,h}(\lambda)^{-1}} \leqslant 1\right\}}_{(a)}$$

$$+ \underbrace{\sum_{t=1}^{T}\sum_{h=1}^{H}|\langle x_h(\pi_t), W_h(f_t, \pi_t)\rangle| \, \mathbf{1}\left\{\|x_h(\pi_t)\|_{\Lambda_{t,h}(\lambda)^{-1}} > 1\right\}}_{(b)}, \quad (100)$$

where $\mathbf{1}\{\cdot\}$ is the indicator function.

To give the desired bound, we will bound (a) and (b) separately.

**Bounding (a).**  We have for any $\lambda > 0$,

$$(a) \leqslant \sum_{t=1}^{T}\sum_{h=1}^{H}\|W_h(f_t, \pi_t)\|_{\Lambda_{t,h}(\lambda)}\|x_h(\pi_t)\|_{\Lambda_{t,h}(\lambda)^{-1}} \, \mathbf{1}\left\{\|x_h(\pi_t)\|_{\Lambda_{t,h}(\lambda)^{-1}} \leqslant 1\right\}$$

$$\leqslant \sum_{t=1}^{T}\sum_{h=1}^{H}\|W_h(f_t, \pi_t)\|_{\Lambda_{t,h}(\lambda)}\min\left\{\|x_h(\pi_t)\|_{\Lambda_{t,h}(\lambda)^{-1}}, 1\right\}. \quad (101)$$

Note that $\|W_h(f_t, \pi_t)\|_{\Lambda_{t,h}(\lambda)}$ can be bounded as follows:

$$\|W_h(f_t, \pi_t)\|_{\Lambda_{t,h}(\lambda)} \leqslant \sqrt{\lambda}\cdot 2H\sqrt{d} + \left(\sum_{i=1}^{t-1}|\langle x_h(\pi_i), W_h(f_t, \pi_t)\rangle|^2\right)^{1/2}, \quad (102)$$

where we use (93), (96) and the fact that $\sqrt{a+b} \leqslant \sqrt{a} + \sqrt{b}$ for any $a, b \geqslant 0$.

The above two bounds (101) and (102) together give

$$(a) \leqslant \sum_{t=1}^{T}\sum_{h=1}^{H}\left(\sqrt{\lambda}\cdot 2H\sqrt{d} + \left(\sum_{i=1}^{t-1}|\langle x_h(\pi_i), W_h(f_t, \pi_t)\rangle|^2\right)^{1/2}\right)\min\left\{\|x_h(\pi_t)\|_{\Lambda_{t,h}(\lambda)^{-1}}, 1\right\}$$

$$\leqslant \underbrace{\left(\sum_{t=1}^{T}\sum_{h=1}^{H}\lambda\cdot 4dH^2\right)^{1/2}\left(\sum_{t=1}^{T}\sum_{h=1}^{H}\min\left\{\|x_h(\pi_t)\|_{\Lambda_{t,h}(\lambda)^{-1}}, 1\right\}\right)^{1/2}}_{(a\text{-}i)}$$

$$+ \underbrace{\left(\sum_{t=1}^{T}\sum_{i=1}^{t-1}\sum_{h=1}^{H}|\langle x_h(\pi_i), W_h(f_t, \pi_t)\rangle|^2\right)^{1/2}\left(\sum_{t=1}^{T}\sum_{h=1}^{H}\min\left\{\|x_h(\pi_t)\|_{\Lambda_{t,h}(\lambda)^{-1}}, 1\right\}\right)^{1/2}}_{(a\text{-}ii)}, \quad (103)$$

where in the second inequality we use Cauchy-Schwarz inequality and the fact that

$$\forall t \in [T]: \quad \min\left\{\|x_h(\pi_t)\|_{\Lambda_{t,h}(\lambda)^{-1}}, 1\right\}^2 \leqslant \min\left\{\|x_h(\pi_t)\|_{\Lambda_{t,h}(\lambda)^{-1}}, 1\right\}. \quad (104)$$

The first term (a-i) in (103) could be bounded as follows:

$$(a\text{-}i) \overset{(98)}{\leqslant} 2H^2\sqrt{2\lambda dTd(\lambda)}. \quad (105)$$

To bound (a-ii), note that for any $\pi, \pi' \in \Pi^H$, we have

$$|\langle x_h(\pi'), W_h(f, \pi)\rangle|^2 = \left|\mathbb{E}_{(s,a)\sim d_{\rho,h}^{\pi'}}\left[Q_{f,h}(s,a) - r_h(s,a) - \mathbb{P}_h V_f^\pi(s,a)\right]\right|^2$$

$$\leqslant \mathbb{E}_{(s,a)\sim d_{\rho,h}^{\pi'}}\left[\ell_h(f, s, a, \pi)\right], \quad (106)$$

where the inequality follows from Jenson's inequality, and recall $\ell_h(f, s, a, \pi)$ is defined in (32). Combining (106) and (98), we could bound (a-ii) in (103) as follows:

$$\text{(a-ii)} \leqslant \left( 2Hd(\lambda) \sum_{t=1}^{T} \sum_{i=1}^{t-1} \sum_{h=1}^{H} \mathbb{E}_{(s_i, a_i) \sim d_{\rho,h}^{\pi_i}} \ell_h(f_t, s_i, a_i, \pi_t) \right)^{1/2} \tag{107}$$

Plugging (105) and (107) into (103), we have

$$\text{(a)} \leqslant 2H^2 \sqrt{2\lambda dT d(\lambda)} + \left( 2Hd(\lambda) \sum_{t=1}^{T} \sum_{i=1}^{t-1} \sum_{h=1}^{H} \mathbb{E}_{(s_i, a_i) \sim d_{\rho,h}^{\pi_i}} \ell_h(f_t, s_i, a_i, \pi_t) \right)^{1/2}. \tag{108}$$

**Bounding (b).** By Assumption 1 and (95), we have

$$\forall \pi \in \Pi: \quad |\langle x_h(\pi), W_h(f, \pi) \rangle| \leqslant 2H. \tag{109}$$

Combining the above inequality with (98), we have

$$\text{(b)} \leqslant 4H^2 d(\lambda). \tag{110}$$

**Combining (a) and (b).** Plugging (108) and (110) into (100), we have

$$\sum_{t=1}^{T} \left| V_{f_t}^{\pi_t}(\rho) - V^{\pi_t}(\rho) \right|$$

$$\leqslant 2H^2 \sqrt{2\lambda dT d(\lambda)} + \left( 2Hd(\lambda) \sum_{t=1}^{T} \sum_{i=1}^{t-1} \sum_{h=1}^{H} \mathbb{E}_{(s_i, a_i) \sim d_{\rho,h}^{\pi_i}} \ell_h(f_t, s_i, a_i, \pi_t) \right)^{1/2} + 4H^2 d(\lambda). \tag{111}$$

The first term in the right hand side of (111) could be bounded as

$$2H^2 \sqrt{2\lambda dT d(\lambda)} \leqslant H^2 \left( \lambda dT + 2d(\lambda) \right), \tag{112}$$

and the second term in the right hand side of (111) could be bounded as

$$\left( 2Hd(\lambda) \sum_{t=1}^{T} \sum_{i=1}^{t-1} \sum_{h=1}^{H} \mathbb{E}_{(s_i, a_i) \sim d_{\rho,h}^{\pi_i}} \ell_h(f_t, s_i, a_i, \pi_t) \right)^{1/2} \leqslant \frac{Hd(\lambda)}{\eta} + \eta \sum_{t=1}^{T} \sum_{i=1}^{t-1} \sum_{h=1}^{H} \mathbb{E}_{(s_i, a_i) \sim d_{\rho,h}^{\pi_i}} \ell_h(f_t, s_i, a_i, \pi_t) \tag{113}$$

for any $\eta > 0$, where in both (112) and (113), we use the fact that $\sqrt{ab} \leqslant \frac{a+b}{2}$ for any $a, b \geqslant 0$.

Substituting (112) and (113) into (111) and reorganizing the terms, we have

$$\sum_{t=1}^{T} \left| V_{f_t}^{\pi_t}(\rho) - V^{\pi_t}(\rho) \right| \leqslant \eta \sum_{t=1}^{T} \sum_{i=1}^{t-1} \sum_{h=1}^{H} \mathbb{E}_{(s_i, a_i) \sim d_{\rho,h}^{\pi_i}} \ell_h(f_t, s_i, a_i, \pi_t) + (6H^2 + H/\eta)d(\lambda) + H^2 \lambda dT. \tag{114}$$

This gives the desired result.

## B.3 Extension to general function approximation

We now extend the analysis to finite-horizon MDPs with general function approximation. We first state our assumptions in this section.

**Assumption 4** (Q-function class). *The Q-function class $\mathcal{Q} = \prod_{h=1}^{H} \mathcal{Q}_h$ satisfies*

- *(realizability) $Q^\star \in \mathcal{Q}$.*
- *(Bellman completeness) $\forall \pi \in \mathcal{P}$ and $f \in \mathcal{Q}$, $\mathbb{P}^\pi f \in \mathcal{Q}$.*
- *(boundedness) $\forall f_h \in \mathcal{Q}_h$, $\|f_h\|_\infty \leqslant H + 1 - h$.*

Assumption 4 is a standard condition in prior literature involving general function approximation [Liu et al., 2024, Assumption 3.1], [Jin et al., 2021, Assumption 2.1]. In particular, Assumption 4 holds under linear MDPs (c.f. Assumption 1), as established inLemma 7. Under Assumption 4, we set the policy class $\mathcal{P}$ as follows.

**Assumption 5** (Policy class). *The policy class* $\mathcal{P} = \prod_{h=1}^{H} \mathcal{P}_h$ *is*

$$\forall h \in [H] : \quad \mathcal{P}_h := \left\{ \pi_h : \pi_h(s,a) = \frac{\exp\left(BQ_h(s,a)\right)}{\sum_{a' \in \mathcal{A}} \exp\left(BQ_h(s,a')\right)}, \ \forall Q_h \in \mathcal{Q}_h \right\} \qquad (115)$$

*with some constant* $B > 0$.

Moreover, drawing upon the work of Zhong et al. [2022], Liu et al. [2024], we require the MDP to feature a low *generalized Eluder coefficient* (GEC). This characteristic is essential for ensuring that the minimization of in-sample prediction error, based on historical data, also effectively limits out-of-sample prediction error.

**Assumption 6** (Generalized Eluder coefficient, Assumption 4.2 in Liu et al. [2024]). *Given any* $\widetilde{\lambda} > 0$, *there exists* $\widetilde{d}(\widetilde{\lambda}) \in \mathbb{R}_+$ *such that for any sequence* $\{f_t\}_{t=1}^{T} \subset \mathcal{Q}$, $\{\pi_t\}_{t=1}^{T} \subset \mathcal{P}$, *we have*

$$\sum_{t=1}^{T} \left( V_{f_t}^{\pi_t}(\rho) - V^{\pi_t}(\rho) \right) \leqslant \inf_{\eta > 0} \eta \sum_{t=1}^{T} \sum_{i=1}^{t-1} \sum_{h=1}^{H} \mathbb{E}_{(s_i,a_i) \sim d_{\rho,h}^{\pi_i}} \ell_h(f_t, s_i, a_i, \pi_t) + \frac{\widetilde{d}(\widetilde{\lambda})}{\eta} + \sqrt{\widetilde{d}(\widetilde{\lambda}) H T} + \widetilde{\lambda} H T.$$
(116)

*For each* $\widetilde{\lambda} > 0$, *we denote the smallest* $\widetilde{d}(\widetilde{\lambda}) \in \mathbb{R}_+$ *that makes* (116) *hold as* $d_{\mathsf{GEC}}(\widetilde{\lambda})$.

From Lemma 10 we can see that under linear MDPs (c.f. Assumption 1), Assumption 6 holds with $d_{\mathsf{GEC}}(\widetilde{\lambda}) \lesssim H d\left(\frac{\widetilde{\lambda}}{dH}\right)$, where $d(\cdot)$ is defined in (35). Moreover, as demonstrated by Zhong et al. [2022], RL problems characterized by a low Generalized Eluder Coefficient (GEC) constitute a significantly broad category, such as linear MDPs [Yang and Wang, 2019, Jin et al., 2020], linear mixture MDPs [Ayoub et al., 2020], MDPs of bilinear classes [Du et al., 2021], MDPs with low witness rank [Sun et al., 2019], and MDPs with low Bellman Eluder dimension [Jin et al., 2021], see Zhong et al. [2022] for a more detailed discussion.

We let $\mathcal{N}(\mathcal{Q}_h, \epsilon, \|\cdot\|_\infty)$ denote the $\epsilon$-covering number of $\mathcal{Q}_h$ w.r.t. the $\ell_\infty$ norm, and assume the $\epsilon$-nets $\mathcal{Q}_{h,\epsilon}$ are finite.

**Assumption 7** (Finite $\epsilon$-nets). $\mathcal{N}(\epsilon) := \max_{h \in [H]} \mathcal{N}(\mathcal{Q}_h, \epsilon, \|\cdot\|_\infty) < +\infty$.

The following theorem gives the regret bound under the above more general assumptions.

**Theorem 11** (Regret under general function approximation). *Suppose Assumptions 4, 5, 6, 7 hold. We let* $B = \frac{T \log |\mathcal{A}|}{H}$ *in Assumption 5, and set*

$$\alpha = \left( \frac{1}{T H^3 \log\left(\frac{\mathcal{N}(\epsilon/B) H T}{\delta}\right)} d_{\mathsf{GEC}}\left(\sqrt{\frac{H}{T}}\right) \right)^{1/2}. \qquad (117)$$

*Then for any* $\delta \in (0,1)$, *with probability at least* $1 - \delta$, *the regret of Algorithm 1 satisfies*

$$\mathsf{Regret}(T) = \mathcal{O}\left( H^{3/2} \sqrt{T} \sqrt{\left( \log\left(\frac{HT}{\delta}\right) + \log\left(\mathcal{N}\left(\frac{H\epsilon}{T \log |\mathcal{A}|}\right)\right) \right) d_{\mathsf{GEC}}\left(\sqrt{\frac{H}{T}}\right)} \right). \quad (118)$$

Under linear MDPs, (118) reduces to (22) given in Theorem 1. Besides, this bound also matches (is slightly tighter than) the bound given in Corollary 5.2 of Liu et al. [2024] under similar assumptions.

## B.4 Proof of Theorem 11

In this proof, we use the same notations as in the proof of Theorem 1 in Appendix B.1. First, we define

$$\widetilde{\pi}_h^\star := \arg\max_{\pi_h \in \mathcal{P}_h} V_{f^\star,h}^\pi(\rho), \quad \forall h \in [H], \qquad (119)$$

and $\widetilde{\pi}^\star = \{\widetilde{\pi}_h^\star\}_{h \in [H]}$. Using the same argument as Lemma 8, we have the following lemma.

**Lemma 12** (model error with log linear policies). *Under Assumption 4 and 5, we have*

$$\forall s \in \mathcal{S}, h \in [H]: \quad 0 \leqslant V_h^\star(s) - V_{f^\star,h}^{\widetilde{\pi}^\star}(s) \leqslant \frac{\log|\mathcal{A}|}{B},\tag{120}$$

*where $B$ is defined in Assumption 5.*

We bound the two terms in the regret decomposition (30) separately.

**Bounding term (i).** Following the same analysis as (31), we have

$$V^\star(\rho) - V_{f_t}^{\pi_t}(\rho) \leqslant \alpha\left(\mathcal{L}_t(f^\star, \widetilde{\pi}^\star) - \mathcal{L}_t(f_t, \pi_t)\right) + \frac{\log|\mathcal{A}|}{B}.\tag{121}$$

It boils down to bound $\mathcal{L}_t(f^\star, \widetilde{\pi}^\star) - \mathcal{L}_t(f_t, \pi_t)$ for each $t \in [T]$. Recall the definition of $\ell_h(f, s, a, \pi)$ in (32), we give the following lemma, whose proof is deferred to Appendix B.2.3.

**Lemma 13.** *Suppose Assumption 4, 5, 7 hold. For any $\delta \in (0, 1)$, with probability at least $1 - \delta$, for any $t \in [T]$, we have*

$$\mathcal{L}_t(f^\star, \widetilde{\pi}^\star) - \mathcal{L}_t(f_t, \pi_t) \leqslant -\frac{1}{2}\sum_{i=1}^{t-1}\sum_{h=1}^{H}\mathbb{E}_{(s_{i,h}, a_{i,h}) \sim d_{\rho,h}^{\pi_i}}\left[\ell_h(f_t, s_{i,h}, a_{i,h}, \pi_t)\right]$$
$$+ CH^3\left(\log\left(\mathcal{N}\left(\epsilon/B\right)\right) + \log(TH/\delta) + \frac{T\log|\mathcal{A}|}{BH}\right)\tag{122}$$

*for some absolute constant $C > 0$.*

By (121) and Lemma 13, we have

$$\text{(i)} \leqslant \alpha\left\{-\frac{1}{2}\sum_{t=1}^{T}\sum_{i=1}^{t-1}\sum_{h=1}^{H}\left(\mathbb{E}_{(s_{i,h}, a_{i,h}) \sim d_{\rho,h}^{\pi_i}}\left[\ell_h(f_t, s_{i,h}, a_{i,h}, \pi_t)\right]\right) + CTH^3\log\left(\frac{\mathcal{N}\left(\epsilon/B\right)HT}{\delta}\right)\right\}$$
$$+ \left(CH^2\alpha T + 1\right)\frac{T\log|\mathcal{A}|}{B}.\tag{123}$$

**Bounding term (ii).** By Assumption 6, we have for any $\widetilde{\lambda} > 0, \eta > 0$,

$$\text{(ii)} \leqslant \eta\sum_{t=1}^{T}\sum_{i=1}^{t-1}\sum_{h=1}^{H}\mathbb{E}_{(s_i, a_i) \sim d_{\rho,h}^{\pi_i}}\ell_h(f_t, s_i, a_i, \pi_t) + \frac{\widetilde{d}(\widetilde{\lambda})}{\eta} + \sqrt{\widetilde{d}(\widetilde{\lambda})HT} + \widetilde{\lambda}HT.\tag{124}$$

**Combining (i) and (ii).** Substituting (123) and (124) into (30), and letting $\eta = \frac{\alpha}{2}$, we have

$$\mathsf{Regret}(T) \leqslant \alpha CTH^3\log\left(\frac{\mathcal{N}\left(\epsilon/B\right)HT}{\delta}\right) + \left(CH^2\alpha T + 1\right)\frac{T\log|\mathcal{A}|}{B} + \frac{2d_{\mathsf{GEC}}(\widetilde{\lambda})}{\alpha} + \sqrt{d_{\mathsf{GEC}}(\widetilde{\lambda})HT} + \widetilde{\lambda}HT.$$

Setting

$$\widetilde{\lambda} = \sqrt{\frac{H}{T}}, \quad \alpha = \left(\frac{d_{\mathsf{GEC}}\left(\sqrt{\frac{H}{T}}\right)}{TH^3\log\left(\frac{\mathcal{N}(\epsilon/B)HT}{\delta}\right)}\right)^{1/2}, \quad \text{and} \quad B = \frac{T\log|\mathcal{A}|}{H}\tag{125}$$

in the above bound, we have with probability at least $1 - \delta$,

$$\mathsf{Regret}(T) \leqslant C'H^{3/2}\sqrt{T}\sqrt{\left(\log\left(\frac{HT}{\delta}\right) + \log\left(\mathcal{N}\left(\frac{H\epsilon}{T\log|\mathcal{A}|}\right)\right)\right)d_{\mathsf{GEC}}\left(\sqrt{\frac{H}{T}}\right)}$$

for some absolute constant $C' > 0$. This completes the proof of Theorem 11.

### B.4.1 Proof of Lemma 13

The proof is similar to the proof of Lemma 9 given in Appendix B.2.3. We use the same notations as in Appendix B.2.3, and also bound the two terms $\mathcal{L}_t(f^\star, \widetilde{\pi}^\star)$ and $-\mathcal{L}_t(f_t, \pi_t)$ in the left-hand side of (122) separately.

**Bounding $-\mathcal{L}_t(f_t, \pi_t)$.** Same as in (48), here we also define

$$X^t_{f,\pi,h} := \mathbb{E}_{a' \sim \pi_{h+1}(\cdot|s_{t,h+1})} \left[ l_h(f, f, \xi_{t,h}, \pi)^2 - l_h(f, \mathbb{P}^\pi f, \xi_{t,h}, \pi)^2 \right], \tag{126}$$

then for any $f \in \mathcal{Q}$:

$$\sum_{i=1}^{t-1} X^i_{f,\pi,h} = \sum_{i=1}^{t-1} \mathbb{E}_{a' \sim \pi_{h+1}(\cdot|s_{i,h+1})} l_h(f, f, \xi_{i,h}, \pi)^2 - \sum_{i=1}^{t-1} \mathbb{E}_{a' \sim \pi_{h+1}(\cdot|s'_{h,i})} l_h(f, \mathbb{P}^\pi f, \xi_{i,h}, \pi)^2$$

$$\leqslant \sum_{i=1}^{t-1} \mathbb{E}_{a' \sim \pi_{h+1}(\cdot|s'_{h,i})} l_h(f, f, \xi_{i,h}, \pi)^2 - \inf_{g \in \mathcal{Q}} \sum_{i=1}^{t-1} \mathbb{E}_{a' \sim \pi_{h+1}(\cdot|s'_{h,i})} l_h(f, g, \xi_{i,h}, \pi)^2 = \mathcal{L}_{t,h}(f, \pi), \tag{127}$$

where we use the fact that $\mathbb{P}^\pi f \in \mathcal{Q}$ guaranteed by Assumption 4. Therefore, to upper bound $-\mathcal{L}_t(f_t, \pi_t) = -\sum_{h=1}^H \mathcal{L}_{t,h}(f_t, \pi_t)$, it suffices to bound $-\sum_{i=1}^{t-1} X^i_{f_t,\pi_t,h}$ for all $h \in [H]$.

For all $h \in [H]$, there exists an $\epsilon$-net $\mathcal{Q}_{h,\epsilon}$ of $\mathcal{Q}_h$ w.r.t. the $\ell_\infty$ norm such that

$$|Q_{h,\epsilon}| \leqslant \mathcal{N}(\epsilon) < +\infty, \tag{128}$$

where the last relation is due to Assumption 4. Then for any $f \in \mathcal{Q}_h$, there exists $f_{h,\epsilon} \in \mathcal{Q}_{h,\epsilon}$ such that

$$\|f - f_{h,\epsilon}\|_\infty \leqslant \epsilon, \tag{129}$$

and thus for any $f \in \mathcal{Q}$ and $\pi \in \mathcal{P}$, we have

$$\left| X^t_{f_\epsilon,\pi,h} - X^t_{f,\pi,h} \right|$$

$$= \left| \mathbb{E}_{a' \sim \pi_{h+1}(\cdot|s_{t,h+1})} \left[ \left( r_h(s_{t,h}, a_{t,h}) + f_{h+1,\epsilon}(s_{t,h+1}, a') - f_{h,\epsilon}(s_{t,h}, a_{t,h}) \right)^2 \right. \right.$$

$$\left. - \left( f_{h+1,\epsilon}(s_{t,h+1}, a') - \mathbb{E}_{\substack{s' \sim \mathbb{P}_h(\cdot|s_{t,h}, a_{t,h}) \\ a' \sim \pi_{h+1}(\cdot|s')}} \left[ f_{h+1}(s', a') \right] \right)^2 \right]$$

$$- \mathbb{E}_{a' \sim \pi_{h+1}(\cdot|s_{t,h+1})} \left[ \left( r_h(s_{t,h}, a_{t,h}) + f_{h+1}(s_{t,h+1}, a') - f_h(s_{t,h}, a_{t,h}) \right)^2 \right.$$

$$\left. \left. - \left( f_{h+1}(s_{t,h+1}, a') - \mathbb{E}_{\substack{s' \sim \mathbb{P}_h(\cdot|s_{t,h}, a_{t,h}) \\ a' \sim \pi_{h+1}(\cdot|s')}} \left[ f_{h+1}(s', a') \right] \right)^2 \right] \right|$$

$$= \left| \mathbb{E}_{a' \sim \pi_{h+1}(\cdot|s_{t,h+1})} \left[ \left( 2r_h(s_{t,h}, a_{t,h}) + f_{h+1,\epsilon}(s_{t,h+1}, a') - f_{h,\epsilon}(s_{t,h}, a_{t,h}) + f_{h+1}(s_{t,h+1}, a') - f_h(s_{t,h}, a_{t,h}) \right) \right. \right.$$

$$\cdot \left( f_{h+1,\epsilon}(s_{t,h+1}, a') - f_{h+1}(s_{t,h+1}, a') - \mathbb{E}_{\substack{s' \sim \mathbb{P}_h(\cdot|s_{t,h}, a_{t,h}) \\ a' \sim \pi_{h+1}(\cdot|s')}} \left[ f_{h+1}(s', a') - f_{h+1,\epsilon}(s', a') \right] \right) \right]$$

$$+ \mathbb{E}_{a' \sim \pi_{h+1}(\cdot|s_{t,h+1})} \left[ \left( f_{h+1}(s_{t,h+1}, a') - f_{h+1,\epsilon}(s_{t,h+1}, a') - \mathbb{E}_{\substack{s' \sim \mathbb{P}_h(\cdot|s_{t,h}, a_{t,h}) \\ a' \sim \pi_{h+1}(\cdot|s')}} \left[ f_{h+1}(s', a') - f_{h+1,\epsilon}(s', a') \right] \right) \right.$$

$$\left. \left. \cdot \left( f_{h+1}(s_{t,h+1}, a') - \mathbb{E}_{\substack{s' \sim \mathbb{P}_h(\cdot|s_{t,h}, a_{t,h}) \\ a' \sim \pi_{h+1}(\cdot|s')}} \left[ f_{h+1}(s', a') \right] + f_{h+1,\epsilon}(s_{t,h+1}, a') - \mathbb{E}_{\substack{s' \sim \mathbb{P}_h(\cdot|s_{t,h}, a_{t,h}) \\ a' \sim \pi_{h+1}(\cdot|s')}} \left[ f_{h+1,\epsilon}(s', a') \right] \right) \right] \right|$$

$$\leqslant 8H\epsilon + 8H\epsilon = 16H\epsilon, \tag{130}$$

where in the last inequality we use (129) and the boundedness of $f_h$ and $f_{h+1}$ assumed in Assumption 4.

In addition, there exists $\mathcal{Q}_{h,\epsilon/B}$ of $\mathcal{Q}_h$ w.r.t. the $\ell_\infty$ norm such that

$$\left| Q_{h,\epsilon/B} \right| \leqslant \mathcal{N}(\epsilon/B) < +\infty. \tag{131}$$

We define

$$\mathcal{P}_{h,\epsilon} := \left\{ \pi_h : \pi_h(s,a) = \frac{\exp\left(BQ_h(s,a)\right)}{\sum_{a' \in \mathcal{A}} \exp\left(BQ_h(s,a')\right)}, \ \forall Q_h \in \mathcal{Q}_{h,\epsilon/B} \right\}, \tag{132}$$

then we have

$$\left|\mathcal{P}_{h,\epsilon}\right| = \left|\mathcal{Q}_{h,\epsilon/B}\right| \leqslant \mathcal{N}(\epsilon/B), \tag{133}$$

and by Assumption 5, for any $\pi_h \in \mathcal{P}_h$, there exists $Q_h \in \mathcal{Q}_{h,\epsilon/B}$ such that

$$\pi_h(s,a) = \frac{\exp\left(BQ_h(s,a)\right)}{\sum_{a' \in \mathcal{A}} \exp\left(BQ_h(s,a')\right)}. \tag{134}$$

There also exists $Q_{h,\epsilon/B} \in \mathcal{Q}_{h,\epsilon/B}$ such that

$$\left\|Q_h - Q_{h,\epsilon/B}\right\|_\infty \leqslant \epsilon/B. \tag{135}$$

We let

$$\pi_{h,\epsilon}(s,a) = \frac{\exp\left(BQ_{h,\epsilon/B}(s,a)\right)}{\sum_{a' \in \mathcal{A}} \exp\left(BQ_{h,\epsilon/B}(s,a')\right)}. \tag{136}$$

Then by Lemma 6, we have

$$\left\|\pi_h - \pi_{h,\epsilon}\right\|_1 \leqslant 2\epsilon. \tag{137}$$

In other words, we have shown that $\mathcal{P}_{h,\epsilon}$ is an $2\epsilon$-net of $\mathcal{P}_h$ w.r.t. the $\ell_1$ norm.

Therefore, we have

$$\left|X^t_{f,\pi_\epsilon,h} - X^t_{f,\pi,h}\right| = \left| \mathbb{E}_{a' \sim \pi_{h+1}(\cdot|s_{t,h+1})} \left[ \left( r_h(s_{t,h},a_{t,h}) + f_{h+1}(s_{t,h+1},a') - f_h(s_{t,h},a_{t,h}) \right)^2 \right. \right.$$

$$\left. - \left( f_{h+1}(s_{t,h+1},a') - \mathbb{E}_{\substack{s' \sim \mathbb{P}_h(\cdot|s_{t,h},a_{t,h}) \\ a' \sim \pi_{h+1}(\cdot|s')}} \left[ f_{h+1}(s',a') \right] \right)^2 \right]$$

$$- \mathbb{E}_{a' \sim \pi_{h+1,\epsilon}(\cdot|s_{t,h+1})} \left[ \left( r_h(s_{t,h},a_{t,h}) + f_{h+1}(s_{t,h+1},a') - f_h(s_{t,h},a_{t,h}) \right)^2 \right.$$

$$\left. \left. - \left( f_{h+1}(s_{t,h+1},a') - \mathbb{E}_{\substack{s' \sim \mathbb{P}_h(\cdot|s_{t,h},a_{t,h}) \\ a' \sim \pi_{h+1,\epsilon}(\cdot|s')}} \left[ f_{h+1}(s',a') \right] \right)^2 \right] \right|$$

$$\leqslant 4H^2 \left\|\pi_{h+1}(\cdot|s_{t,h+1}) - \pi_{h+1,\epsilon}(\cdot|s_{t,h+1})\right\|_1 \overset{(137)}{\leqslant} 8H^2\epsilon, \tag{138}$$

where the first inequality follows from Hölder's inequality and the fact that

$$\left| \left(r_h(s,a) + f_{h+1}(s',a') - f_h(s,a)\right)^2 - \left( f_{h+1}(s',a') - \mathbb{E}_{\substack{s' \sim \mathbb{P}_h(\cdot|s,a) \\ a' \sim \pi_{h+1}(\cdot|s')}} \left[ f_{h+1}(s',a') \right] \right)^2 \right| \leqslant 4H^2$$

for all $(s,a) \in \mathcal{S} \times \mathcal{A}$, $f \in \mathcal{Q}$ and $\pi \in \mathcal{P}$, which is ensured by Assumption 4.

Combining (130) and (138), we have

$$\left|X^t_{f_\epsilon,\pi_\epsilon,h} - X^t_{f,\pi,h}\right| \leqslant \left|X^t_{f_\epsilon,\pi_\epsilon,h} - X^t_{f_\epsilon,\pi,h}\right| + \left|X^t_{f_\epsilon,\pi,h} - X^t_{f,\pi,h}\right| \leqslant 16H\epsilon + 8H^2\epsilon = 24H^2\epsilon. \tag{139}$$

On the other hand, Assumption 4 ensures $X^t_{f,\pi_h}$ is bounded:

$$\forall f \in \mathcal{Q}, \pi \in \mathcal{P}, h \in [H]: \quad \left|X^t_{f,\pi,h}\right| \leqslant 4H^2. \tag{140}$$

Thus following the same argument as in Appendix B.2.3 that leads to (63), here we could obtain that for any $\delta \in (0,1)$, with probability at least $1 - \delta$, for all $t \in [T]$, $h \in [H]$, $f_\epsilon \in \mathcal{Q}_\epsilon = \prod_{h=1}^{H} \mathcal{Q}_{h,\epsilon}$ and $\pi_\epsilon \in \mathcal{P}_\epsilon = \prod_{h=1}^{H} \mathcal{P}_{h,\epsilon}$,

$$\sum_{i=1}^{t-1} \mathbb{E}_{(s_{i,h},a_{i,h}) \sim d^{\pi_i}_{\rho,h}} \left[\ell_h(f_\epsilon, s_{i,h}, a_{i,h}, \pi_\epsilon)\right] - \sum_{i=1}^{t-1} X^i_{f_\epsilon,\pi_\epsilon,h}$$

$$\leqslant \frac{1}{2} \sum_{i=1}^{t-1} \mathbb{E}_{(s_{i,h}, a_{i,h}) \sim d_{\rho,h}^{\pi_i}} \left[ \ell_h(f_\epsilon, s_{i,h}, a_{i,h}, \pi_\epsilon) \right] + C_1 H^2 \log(TH|\mathcal{Q}_{h,\epsilon}||\mathcal{Q}_{h+1,\epsilon}||\mathcal{P}_{h,\epsilon}|/\delta)$$

$$\leqslant \frac{1}{2} \sum_{i=1}^{t-1} \mathbb{E}_{(s_{i,h}, a_{i,h}) \sim d_{\rho,h}^{\pi_i}} \left[ \ell_h(f_\epsilon, s_{i,h}, a_{i,h}, \pi_\epsilon) \right] + C_1' H^2 \left( \log \left( \mathcal{N} \left( \epsilon/B \right) \right) + \log(TH/\delta) \right), \quad (141)$$

where $C_1, C_1' > 0$ are absolute constants.

From (141) we deduce that for all $t \in [T]$, $f_\epsilon \in \mathcal{Q}_\epsilon$, and $\pi_\epsilon \in \mathcal{P}_\epsilon$, we have with probability at least $1 - \delta$,

$$-\sum_{i=1}^{t-1} \sum_{h=1}^{H} X_{f_\epsilon, \pi_\epsilon, h}^i \leqslant -\frac{1}{2} \sum_{i=1}^{t-1} \sum_{h=1}^{H} \mathbb{E}_{(s_{i,h}, a_{i,h}) \sim d_{\rho,h}^{\pi_i}} \left[ \ell_h(f_\epsilon, s_{i,h}, a_{i,h}, \pi_\epsilon) \right] + C_1' H^3 \left( \log \left( \mathcal{N} \left( \epsilon/B \right) \right) + \log(TH/\delta) \right).$$
$$(142)$$

By (137), for any $t \in [T]$ and $h \in [H]$, we can choose $f_{t,h,\epsilon} \in \mathcal{Q}_{h,\epsilon}$ and $\pi_{t,h,\epsilon} \in \mathcal{P}_{h,\epsilon}$ such that

$$\|f_{t,h} - f_{t,h,\epsilon}\|_\infty \leqslant \epsilon, \quad \|\pi_{t,h} - \pi_{t,h,\epsilon}\|_1 \leqslant 2\epsilon. \quad (143)$$

Then by (142) we have for all $t \in [T]$,

$$- \mathcal{L}_t(f_t, \pi_t)$$

$$\overset{(127)}{\leqslant} -\sum_{i=1}^{t-1} \sum_{h=1}^{H} X_{f_t, \pi_t, h}^i$$

$$\overset{(139)}{\leqslant} -\sum_{i=1}^{t-1} \sum_{h=1}^{H} X_{f_{t,\epsilon}, \pi_{t,\epsilon}, h}^i + 24 H^3 \epsilon T$$

$$\overset{(142)}{\leqslant} -\frac{1}{2} \sum_{i=1}^{t-1} \sum_{h=1}^{H} \mathbb{E}_{(s_{i,h}, a_{i,h}) \sim d_{\rho,h}^{\pi_i}} \left[ \ell_h(f_{t,\epsilon}, s_{i,h}, a_{i,h}, \pi_{t,\epsilon}) \right] + C_1' H^3 \left( \log \left( \mathcal{N} \left( \epsilon/B \right) \right) + \log(TH/\delta) \right) + 24 H^3 \epsilon T$$

$$\leqslant -\frac{1}{2} \sum_{i=1}^{t-1} \sum_{h=1}^{H} \mathbb{E}_{(s_{i,h}, a_{i,h}) \sim d_{\rho,h}^{\pi_i}} \left[ \ell_h(f_t, s_{i,h}, a_{i,h}, \pi_t) \right] + C_1' H^3 \left( \log \left( \mathcal{N} \left( \epsilon/B \right) \right) + \log(TH/\delta) \right) + 36 H^3 \epsilon T,$$
$$(144)$$

where the last line follows from (139) and (59).

**Bounding $\mathcal{L}_t(f^\star, \widetilde{\pi}^\star)$.** Same as in (66), for any $f \in \mathcal{Q}$ and $t \in [T]$, we define

$$Y_{f,h}^t := \mathbb{E}_{a' \sim \widetilde{\pi}_{h+1}^\star(\cdot|s_{t,h})} \left[ l_h(f^\star, f, \xi_{t,h}, \widetilde{\pi}^\star)^2 - l_h(f^\star, \widetilde{f}^\star, \xi_{t,h}, \widetilde{\pi}^\star)^2 \right], \quad (145)$$

where we define

$$\widetilde{f}^\star := \mathbb{P}^{\widetilde{\pi}^\star} f^\star. \quad (146)$$

Then following the same argument that leads to (79), setting $\eta$ in Lemma 2 as

$$\eta = \min \left\{ \frac{1}{4H^2}, \sqrt{\frac{\log(|\mathcal{Q}_{h,\epsilon}||\mathcal{Q}_{h+1,\epsilon}|HT/\delta)}{\sum_{i=1}^{t-1} \mathsf{Var}\left[ Y_{f,h}^i | \mathcal{F}_{i-1} \right]}} \right\}$$

we have with probability at least $1 - \delta$, for any $f_\epsilon \in \mathcal{Q}_\epsilon, t \in [T]$:

$$\sum_{i=1}^{t-1} \left( -Y_{f_\epsilon, h}^i \right)$$

$$\lesssim -\sum_{i=1}^{t-1} \mathbb{E}_{(s_{i,h}, a_{i,h}) \sim d_{\rho,h}^{\pi_i}} \left[ \left( \mathbb{E}_{\substack{s_{i,h+1} \sim \mathbb{P}_h(\cdot|s_{i,h}, a_{i,h}) \\ a' \sim \widetilde{\pi}_{h+1}^\star(\cdot|s_{i,h+1})}} \left[ l_h(f^\star, f_\epsilon, \xi_{i,h}, \widetilde{\pi}^\star) \right] \right)^2 \right] + H^2 \log(|\mathcal{Q}_{h,\epsilon}||\mathcal{Q}_{h+1,\epsilon}|HT/\delta)$$

$$+ H \sqrt{\log(|\mathcal{Q}_{h,\epsilon}||\mathcal{Q}_{h+1,\epsilon}|HT/\delta) \sum_{i=1}^{t-1} \mathbb{E}_{(s_{i,h},a_{i,h})\sim d_{\rho,h}^{\pi_i}} \left[ \left( \mathbb{E}_{\substack{s_{i,h+1}\sim\mathbb{P}_h(\cdot|s_{i,h},a_{i,h}) \\ a'\sim\widetilde{\pi}_{h+1}^\star(\cdot|s_{i,h+1})}} [l_h(f^\star, f_\epsilon, \xi_{i,h}, \widetilde{\pi}^\star)] \right)^2 \right]}$$

$$\lesssim H^2 \log(\mathcal{N}(\epsilon)HT/\delta), \tag{147}$$

where the last line makes use of the fact that $-x^2 + bx \leqslant b^2/4$.

Moreoever, for any $t \in [T]$, $h \in [H]$, we have

$$Y_{f_\epsilon,h}^t - Y_{f,h}^t = \mathbb{E}_{a'\sim\widetilde{\pi}_{h+1}^\star(\cdot|s_{t,h})} \Bigg[ \left( r_h(s_{t,h},a_{t,h}) + f_{h+1}^\star(s_{t,h+1},a') - f_{\epsilon,h}(s_{t,h},a_{t,h}) \right)^2$$

$$- \left( r_h(s_{t,h},a_{t,h}) + f_{h+1}^\star(s_{t,h+1},a') - f_h(s_{t,h},a_{t,h}) \right)^2 \Bigg]$$

$$= \mathbb{E}_{a'\sim\widetilde{\pi}_{h+1}^\star(\cdot|s_{t,h})} \Bigg[ \left( 2r_h(s_{t,h},a_{t,h}) + 2f_{h+1}^\star(s_{t,h+1},a') - f_{\epsilon,h}(s_{t,h},a_{t,h}) - f_h(s_{t,h},a_{t,h}) \right)$$

$$\cdot \left( f_h(s_{t,h},a_{t,h}) - f_{\epsilon,h}(s_{t,h},a_{t,h}) \right) \Bigg] \leqslant 4H\epsilon. \tag{148}$$

Combining (147) and (148), we have with probability at least $1 - \delta$, for any $t \in [T]$ and $f \in \mathcal{Q}$,

$$\sum_{i=1}^{t-1}\sum_{h=1}^{H} \left( -Y_{f,h}^i \right) \leqslant \sum_{i=1}^{t-1}\sum_{h=1}^{H} \left( -Y_{f_\epsilon,h}^i \right) + 4H^2\epsilon T$$

$$\overset{(52)}{\leqslant} C_2 H^3 \log(\mathcal{N}(\epsilon)HT/\delta) + 4H^2\epsilon T, \tag{149}$$

where $C_2 > 0$ is an absolute constant.

By (71) we have

$$\mathcal{L}_t(f^\star, \widetilde{\pi}^\star) \leqslant C_2 H^3 \log(\mathcal{N}(\epsilon)HT/\delta) + 4H^2\epsilon T + \frac{4H^2 T \log|\mathcal{A}|}{B}. \tag{150}$$

**Combining the two bounds.** Combining (144) and (150), we have for any $t \in [T]$,

$$\mathcal{L}_t(f^\star, \widetilde{\pi}^\star) - \mathcal{L}_t(f_t, \pi_t) \leqslant -\frac{1}{2} \sum_{i=1}^{t-1}\sum_{h=1}^{H} \mathbb{E}_{(s_{i,h},a_{i,h})\sim d_{\rho,h}^{\pi_i}} [\ell_h(f_t, s_{i,h}, a_{i,h}, \pi_t)]$$

$$+ CH^3 \left( \log\left(\mathcal{N}\left(\epsilon/B\right)\right) + \log(TH/\delta) + \epsilon T + \frac{T\log|\mathcal{A}|}{BH} \right) \tag{151}$$

for some absolute constant $C > 0$. Letting $\epsilon = \frac{1}{T}$, we obtain the desired result.

## C  Value-incentivized Actor-Critic Method for Discounted MDPs

**Infinite-horizon MDPs.** Let $\mathcal{M} = (\mathcal{S}, \mathcal{A}, P, r, \gamma)$ be an infinite-horizon discounted MDP, where $\mathcal{S}$ and $\mathcal{A}$ denote the state space and the action space, respectively, $\gamma \in [0, 1)$ denotes the discount factor, $P : \mathcal{S} \times \mathcal{A} \mapsto \Delta(\mathcal{S})$ is the transition kernel, and $r : \mathcal{S} \times \mathcal{A} \mapsto [0, 1]$ is the reward function. A policy $\pi : \mathcal{S} \mapsto \Delta(\mathcal{A})$ specifies an action selection rule, where $\pi(a|s)$ specifies the probability of taking action $a$ in state $s$ for each $(s, a) \in \mathcal{S} \times \mathcal{A}$. For any given policy $\pi$, the value function, denoted by $V^\pi : \mathcal{S} \mapsto \mathbb{R}$, is given as

$$\forall s \in \mathcal{S} : \quad V^\pi(s) := \mathbb{E}\left[ \sum_{t=0}^{\infty} \gamma^t r(s_t, a_t) | s_0 = s \right], \tag{152}$$

which measures the expected discounted cumulative reward starting from an initial state $s_0 = s$, where the randomness is over the trajectory generated following $a_t \sim \pi(\cdot|s_t)$ and the MDP dynamic $s_{t+1} \sim P(\cdot|s_t, a_t)$. Given an initial state distribution $s_0 \sim \rho$ over $\mathcal{S}$, we also define $V^\pi(\rho) :=$

$\mathbb{E}_{s \sim \rho}\left[V^{\pi}(s)\right]$ with slight abuse of notation. Similarly, the Q-function of policy $\pi$, denoted by $Q^{\pi}$ : $\mathcal{S} \times \mathcal{A} \mapsto \mathbb{R}$, is defined as

$$\forall(s, a) \in \mathcal{S} \times \mathcal{A}: \quad Q^{\pi}(s, a) := \mathbb{E}\left[\sum_{t=0}^{\infty} \gamma^t r(s_t, a_t) | s_0 = s, a_0 = a\right], \tag{153}$$

which measures the expected discounted cumulative reward with an initial state $s_0 = s$ and an initial action $a_0 = a$, with expectation taken over the randomness of the trajectory. It is known that there exists at least one optimal policy $\pi^{\star}$ that maximizes the value function $V^{\pi}(s)$ for all states $s \in \mathcal{S}$ [Puterman, 2014], whose corresponding optimal value function and Q-function are denoted as $V^{\star}$ and $Q^{\star}$, respectively. We also define the state-action visitation distribution $d_{\rho}^{\pi} \in \Delta(\mathcal{S} \times \mathcal{A})$ induced by policy $\pi$ and initial state distribution $\rho$ as

$$d_{\rho}^{\pi}(s, a) := (1 - \gamma)\mathbb{E}_{s_0 \sim \rho}\left[\sum_{h=0}^{\infty} \gamma^h \Pr(s_h = s, a_h = a | s_0)\right]. \tag{154}$$

### C.1 Algorithm development

Similar as (13), we start with an optimization problem:

$$\max_{f \in \mathcal{Q}, \pi} (1 - \gamma)\mathbb{E}_{s_0 \sim \rho, a \sim \pi(\cdot|s_0)}\left[Q_f(s_0, a)\right] \tag{155}$$

$$\text{s.t. } Q_f(s, a) = r(s, a) + \gamma \cdot \mathbb{E}_{s' \sim P(\cdot|s,a), a' \sim \pi(\cdot|s')}[Q_f(s', a')], \ \forall(s, a) \in \mathcal{S} \times \mathcal{A}.$$

Writing the regularized Lagrangian system of (155) as

$$\max_{f, \pi} (1 - \gamma)\mathbb{E}_{s_0 \sim \rho, a \sim \pi(\cdot|s_0)}\left[Q_f(s_0, a)\right]$$

$$+ \min_{\lambda} \int \lambda(s, a)\left(r(s, a) + \gamma \cdot \mathbb{E}_{s' \sim P(\cdot|s,a), a' \sim \pi(\cdot|s')}[Q_f(s', a')] - Q_f(s, a)\right) + \frac{\beta(s, a)}{2}\lambda(s, a)^2 ds da. \tag{156}$$

Similar to the finite-horizon case, we use the reparameterization (10) which gives

$$\max_{f, \pi}\left\{(1 - \gamma)\mathbb{E}_{s_0 \sim \rho, a \sim \pi(\cdot|s_0)}[Q_f(s_0, a)] - \int \frac{1}{2\beta(s, a)}\mathbb{E}_{s' \sim P(\cdot|s,a), a' \sim \pi(\cdot|s')}\left[\left(r(s, a) + \gamma Q_f(s', a') - Q_f(s, a)\right)^2\right]\right. \tag{157}$$

$$\left. - \min_{\rho}\left(r(s, a) + \gamma Q_f(s', a') - g(s, a)\right)^2\right] ds da\right\},$$

which is easier to optimize over both $Q_f$ and $\pi$. The population primal-dual optimization problem (157) prompts us to design the proposed algorithm, by computing the sample version of (157), see Algorithm 2, where we let

$$V_f^{\pi}(s) := \mathbb{E}_{a \sim \pi(\cdot|s)}\left[Q_f(s, a)\right], \quad \text{and} \quad V_f^{\pi}(\rho) := \mathbb{E}_{s \sim \rho}\left[V_f^{\pi}(s)\right]. \tag{158}$$

In Algorithm 2, at iteration $t$, given dataset $\mathcal{D}_{t-1}$ collected from the previous iterations, we define the loss function as follows:

$$\mathcal{L}_t(f, \pi) = \sum_{(s, a, s') \in \mathcal{D}_{t-1}} \mathbb{E}_{a' \sim \pi(\cdot|s')}\left(r(s, a) + \gamma Q_f(s', a') - Q_f(s, a)\right)^2$$

$$- \inf_{g \in \mathcal{Q}} \sum_{(s, a, s') \in \mathcal{D}_{t-1}} \mathbb{E}_{a' \sim \pi(\cdot|s')}\left(r(s, a) + \gamma Q_f(s', a') - g(s, a)\right)^2. \tag{159}$$

We compute (160) in each iteration, which is the sample version of (157), and use the current policy $\pi_t$ to collect new data following the sampling procedure in Algorithm 3, which is also used in Yuan et al. [2023, Algorithm 3], Yang et al. [2024, Algorithm 5], and Yang et al. [2025, Algorithm 7]. Algorithm 3 has an expected iteration number $\mathbb{E}[h + 1] = \frac{1}{1-\gamma}$, and it guarantees $\mathbb{P}(s_h = s, a_h = a) = d_{\rho}^{\pi}(s, a)$ [Yuan et al., 2023] for any $(s, a) \in \mathcal{S} \times \mathcal{A}$ and any policy $\pi$.

**Algorithm 2** Value-incentivized Actor-Critic (VAC) for infinite-horizon discounted MDPs.

---

1: **Input:** regularization coefficient $\alpha > 0$.
2: **Initialization:** dataset $\mathcal{D}_0 := \emptyset$.
3: **for** $t = 1, \cdots, T$ **do**
4:     Update Q-function estimation and policy:

$$(f_t, \pi_t) \leftarrow \arg\max_{f \in \mathcal{Q}, \pi \in \mathcal{P}} \left\{ (1-\gamma)V_f^\pi(\rho) - \alpha \mathcal{L}_t(f, \pi) \right\}. \tag{160}$$

5:     Data collection: sample $(s_t, a_t, s_t') \leftarrow \mathsf{Sampler}(\pi_t, \rho)$, and update the dataset $\mathcal{D}_t = \mathcal{D}_{t-1} \cup \{(s_t, a_t, s_t')\}$.
6: **end for**

---

**Algorithm 3** Sampler for $(s, a) \sim d_\rho^\pi$ and $s' \sim \mathbb{P}(\cdot|s, a)$

---

1: **Input:** policy $\pi$, initial state distribution $\rho$, player index $n$.
2: **Initialization:** $s_0 \sim \rho$, $a_0 \sim \pi(\cdot|s_0)$, time step $h = 0$, variable $X \sim \text{Bernoulli}(\gamma)$.
3: **while** $X = 1$ **do**
4:     Sample $s_{h+1} \sim P(\cdot|s_h, a_h)$
5:     Sample $a_{h+1} \sim \pi(\cdot|s_{h+1})$
6:     $h \leftarrow h + 1$
7:     $X \sim \text{Bernoulli}(\gamma)$
8: **end while**
9: Sample $s_{h+1} \sim P(\cdot|s_h, a_h)$
10: **return** $(s_h, a_h, s_{h+1})$.

---

### C.2 Theoretical guarantees

Same as the finite-horizon setting, we assume the following $d$-dimensional linear MDP model.

**Assumption 8** (infinite-horizon linear MDP). *There exists* unknown *vector $\zeta \in \mathbb{R}^d$ and* unknown *(signed) measures $\mu = (\mu^{(1)}, \cdots, \mu^{(d)})$ over $\mathcal{S}$ such that*

$$r(s, a) = \phi(s, a)^\top \zeta \quad \text{and} \quad P(s'|s, a) = \phi(s, a)^\top \mu(s'),$$

*where $\phi : \mathcal{S} \times \mathcal{A} \rightarrow \mathbb{R}^d$ is a* known *feature map satisfying $\|\phi(s, a)\|_2 \leqslant 1$, and $\max\{\|\zeta\|_2, \|\mu(\mathcal{S})\|_2\} \leqslant \sqrt{d}$, for all $(s, a, s') \in \mathcal{S} \times \mathcal{A} \times \mathcal{S}$.*

Similar as for the finite case, under Assumption 8, we only need to set the Q-function class to be linear and the policy class $\mathcal{P}$ to be the set of log-linear policies.

**Assumption 9** (linear $Q$-function class (infinite-horizon)). *The function class $\mathcal{Q}$ is defined as*

$$\mathcal{Q} := \left\{ f_\theta := \phi(\cdot, \cdot)^\top \theta : \|\theta\|_2 \leqslant \frac{\sqrt{d}}{1 - \gamma}, \|f_\theta\|_\infty \leqslant \frac{1}{1 - \gamma} \right\}.$$

**Assumption 10** (log-linear policy class (infinite-horizon)). *The policy class $\mathcal{P}$ is defined as*

$$\mathcal{P} := \left\{ \pi_\omega : \pi_\omega(s, a) = \frac{\exp\left(\phi(s, a)^\top \omega\right)}{\sum_{a' \in \mathcal{A}} \exp\left(\phi(s, a')^\top \omega\right)} \text{ with } \|\omega\|_2 \leqslant \frac{B\sqrt{d}}{1 - \gamma} \right\}$$

*with some constant $B > 0$.*

We give the regret bound of Algorithm 2 in Theorem 14.

**Theorem 14** (infinite-horizon). *Suppose Assumptions 8-10 hold. We let $B = \frac{T \log|\mathcal{A}|(1-\gamma)}{d}$ in Assumption 10 and set*

$$\alpha = \left( \frac{(1-\gamma)^2}{T \log\left(\log|\mathcal{A}|T/\delta\right)} \log\left(1 + \frac{T^{3/2}}{d(1-\gamma)^2}\right) \right)^{1/2}. \tag{161}$$

*Then for any $\delta \in (0, 1)$, with probability at least $1 - \delta$, the regret of Algorithm 2 satisfies*

$$\mathsf{Regret}(T) = \mathcal{O}\left( \frac{d\sqrt{T}}{(1-\gamma)^2} \sqrt{\log\left(\frac{\log(|\mathcal{A}|)T}{\delta}\right) \log\left(1 + \frac{T^{3/2}}{d(1-\gamma)^2}\right)} \right). \tag{162}$$

Note that

$$\min_{t \in [T]} \left( V^\star(\rho) - V^{\pi_t}(\rho) \right) \leqslant \frac{\mathsf{Regret}(T)}{T},$$

thus Theorem 14 guarantees that the iteration complexity to reach $\epsilon$-accuracy w.r.t. value suboptimality for any $\epsilon > 0$ is $\widetilde{\mathcal{O}}\left( \frac{d^2}{(1-\gamma)^4 \epsilon^2} \right)$, and the total sample complexity is $\widetilde{\mathcal{O}}\left( \frac{d^2}{(1-\gamma)^5 \epsilon^2} \right)$.

## C.3   Proof of Theorem 14

**Notation.**   For notation simplicity, we let $f^\star := Q^\star$ be the optimal Q-function. We let $\Pi := \Delta(\mathcal{A})^{\mathcal{S}}$ denote the set of all policies. We also define transition tuples

$$\xi := (s, a, s') \in \mathcal{S} \times \mathcal{A} \times \mathcal{S} \quad \text{and} \quad \xi_t := (s_t, a_t, s_t') \in \mathcal{S} \times \mathcal{A} \times \mathcal{S}. \tag{163}$$

Given any policy $\pi$ and $f : \mathcal{S} \times \mathcal{A} \to \mathbb{R}$, we define $\mathbb{P}^\pi f$ as

$$\forall (s,a) \in \mathcal{S} \times \mathcal{A} : \quad \mathbb{P}^\pi f(s,a) := r(s,a) + \gamma \mathbb{E}_{s' \sim \mathbb{P}(\cdot|s,a), a' \sim \pi(\cdot|s')} \left[ f(s', a') \right]. \tag{164}$$

We let

$$\Theta := \{ \theta : f_\theta \in \mathcal{Q} \}, \quad \Omega := \left\{ \omega : \|\omega\|_2 \leqslant \frac{B\sqrt{d}}{1 - \gamma} \right\} \tag{165}$$

be the parameter space of $\mathcal{Q}$ and $\mathcal{P}$, respectively.

We'll repeatedly use the following lemma, which is a standard consequence of linear MDP.

**Lemma 15** (Linear MDP $\Rightarrow$ Bellman completeness + realizability (infinite-horizon)). *Under Assumption 8, we have*

- *(realizability) $Q^\star \in \mathcal{Q}$;*

- *(Bellman completeness) $\forall \pi \in \Pi$ and $f \in \mathcal{Q}$, $\mathbb{P}^\pi f \in \mathcal{Q}$.*

We'll also use the following lemma, which bounds the difference between the optimal value function $V^\star(\rho)$ and $\max_{\pi \in \mathcal{P}} V^\pi(\rho)$ — the optimal value over the policy class $\mathcal{P}$, where we let

$$\widetilde{\pi}^\star := \arg\max_{\pi \in \mathcal{P}} V_{f^\star}^\pi(\rho). \tag{166}$$

**Lemma 16** (model error with log linear policies (infinite-horizon)). *Under Assumptions 8-10, we have*

$$\forall s \in \mathcal{S} : \quad 0 \leqslant V^\star(s) - V_{f^\star}^{\widetilde{\pi}^\star}(s) \leqslant \frac{\log |\mathcal{A}|}{B}, \tag{167}$$

*where $B$ is defined in Assumption 10.*

We omit the proofs of the above two lemmas due to similarity to that of the finite-horizon setting.

**Main proof of Theorem 14.**   Given the regret decomposition in (30), we will bound the two terms separately.

**Step 1: bounding term (i).**   Similar to the argument in the finite-horizon setting, invoking Lemma 16, we have

$$V^\star(\rho) - V_{f_t}^{\pi_t}(\rho) \leqslant \frac{\alpha}{1 - \gamma} \left( \mathcal{L}_t(f^\star, \widetilde{\pi}^\star) - \mathcal{L}_t(f_t, \pi_t) \right) + \frac{\log |\mathcal{A}|}{B}. \tag{168}$$

Thus to bound (i), we only need to bound $\mathcal{L}_t(f^\star, \widetilde{\pi}^\star) - \mathcal{L}_t(f_t, \pi_t)$ for each $t \in [T]$. Define $\ell : \mathcal{Q} \times \mathcal{S} \times \mathcal{A} \times \Pi$ as

$$\ell(f, s, a, \pi) := \left( \mathbb{E}_{s' \sim \mathbb{P}(\cdot|s,a), a' \sim \pi(\cdot|s')} \left[ r(s,a) + \gamma f(s', a') - f(s,a) \right] \right)^2. \tag{169}$$

We give the following lemma, whose proof is deferred to Appendix C.4.1.

**Lemma 17.** *Suppose Assumption 8-10 hold. For any $\delta \in (0,1)$, with probability at least $1 - \delta$, for any $t \in [T]$, we have*

$$\mathcal{L}_t(f^\star, \widetilde{\pi}^\star) - \mathcal{L}_t(f_t, \pi_t) \leqslant -\frac{1}{2}\sum_{i=1}^{t-1} \mathbb{E}_{(s_i,a_i)\sim d_\rho^{\pi_i}} \left[\ell(f_t, s_i, a_i, \pi_t)\right]$$

$$+ \frac{C}{(1-\gamma)^2} \cdot \left(d\log\left(\frac{BdT}{(1-\gamma)\delta}\right) + (1-\gamma)\frac{T\log|\mathcal{A}|}{B}\right) \quad (170)$$

*for some absolute constant $C > 0$.*

By (168) and Lemma 17, we have

$$V^\star(\rho) - V_{f_t}^{\pi_t}(\rho) \leqslant \frac{\alpha}{1-\gamma}\left\{-\frac{1}{2}\sum_{i=1}^{t-1} \mathbb{E}_{(s_i,a_i)\sim d_\rho^{\pi_i}}\left[\ell(f_t,s_i,a_i,\pi_t)\right] + \frac{C}{(1-\gamma)^2}\cdot d\log\left(\frac{BdT}{(1-\gamma)\delta}\right)\right\}$$

$$+ \left(\frac{C\alpha T}{(1-\gamma)^2} + 1\right)\frac{\log|\mathcal{A}|}{B},$$

which gives

$$\text{(i)} \leqslant \frac{\alpha}{1-\gamma}\left\{-\frac{1}{2}\sum_{t=1}^{T}\sum_{i=1}^{t-1} \mathbb{E}_{(s_i,a_i)\sim d_\rho^{\pi_i}}\left[\ell(f_t,s_i,a_i,\pi_t)\right] + \frac{CT}{(1-\gamma)^2}\cdot d\log\left(\frac{BdT}{(1-\gamma)\delta}\right)\right\}$$

$$+ \left(\frac{C\alpha T}{(1-\gamma)^2} + 1\right)\frac{T\log|\mathcal{A}|}{B}. \quad (171)$$

**Step 2: bounding term (ii).** For any $\lambda > 0$, we define

$$d_\gamma(\lambda) := d\log\left(1 + \frac{T}{d\lambda(1-\gamma)^2}\right). \quad (172)$$

We use the following lemma to bound (ii), whose proof is deferred to Appendix C.4.2.

**Lemma 18.** *Under Assumption 8, for any $\eta > 0$, we have*

$$\sum_{t=1}^{T}\left|V_{f_t}^{\pi_t}(\rho) - V^{\pi_t}(\rho)\right|$$

$$\leqslant \frac{\eta}{1-\gamma}\cdot\sum_{t=1}^{T}\sum_{i=1}^{t-1} \mathbb{E}_{(s_i,a_i)\sim d_\rho^{\pi_i}}\ell(f_t,s_i,a_i,\pi_t) + \left(\frac{7}{1-\gamma} + \frac{1}{\eta(1-\gamma)}\right)d_\gamma(\lambda) + \frac{3Td\lambda}{2(1-\gamma)}. \quad (173)$$

By Lemma 18, we have

$$\text{(ii)} \leqslant \frac{\eta}{1-\gamma}\cdot\sum_{t=1}^{T}\sum_{i=1}^{t-1} \mathbb{E}_{(s_i,a_i)\sim d_\rho^{\pi_i}}\ell(f_t,s_i,a_i,\pi_t) + \left(\frac{7}{1-\gamma} + \frac{1}{\eta(1-\gamma)}\right)d_\gamma(\lambda) + \frac{3Td\lambda}{2(1-\gamma)}. \quad (174)$$

**Step 3: combining (i) and (ii).** Substituting (171) and (174) into (30), and letting $\eta = \frac{\alpha}{2}$, we have

$$\text{Regret}(T) \leqslant \frac{CT\alpha}{(1-\gamma)^3}\cdot d\log\left(\frac{BdT}{(1-\gamma)\delta}\right) + \left(\frac{C\alpha T}{(1-\gamma)^2} + 1\right)\frac{T\log|\mathcal{A}|}{B}$$

$$+ \left(\frac{7}{1-\gamma} + \frac{2}{\alpha(1-\gamma)}\right)d_\gamma(\lambda) + \frac{3Td\lambda}{2(1-\gamma)}. \quad (175)$$

Setting

$$\lambda = \frac{1}{\sqrt{T}}, \quad \alpha = \left(\frac{(1-\gamma)^2\log\left(1 + \frac{T^{3/2}}{d(1-\gamma)^2}\right)}{T\log\left(\log|\mathcal{A}|T/\delta\right)}\right)^{1/2}, \quad \text{and} \quad B = \frac{T\log|\mathcal{A}|(1-\gamma)}{d} \quad (176)$$

in the above bound, we have with probability at least $1 - \delta$,

$$\mathsf{Regret}(T) \leqslant C' \frac{d\sqrt{T}}{(1-\gamma)^2} \sqrt{\log\left(\frac{\log(|\mathcal{A}|)T}{\delta}\right) \log\left(1 + \frac{T^{3/2}}{d(1-\gamma)^2}\right)}.$$

for some absolute constant $C' > 0$. This completes the proof of Theorem 14.

## C.4 Proof of key lemmas

### C.4.1 Proof of Lemma 17

We bound the two terms $\mathcal{L}_t(f^\star, \widetilde{\pi}^\star)$ and $-\mathcal{L}_t(f_t, \pi_t)$ in the left-hand side of (170) separately. Given $f, f' : \mathcal{S} \times \mathcal{A} \to \mathbb{R}$, data tuple $\xi = (s, a, s')$ and policy $\pi$, we define the random variable

$$l(f, f', \xi, \pi) := r(s, a) + \gamma f(s', a') - f'(s, a), \tag{177}$$

where $a' \sim \pi(\cdot|s')$. Then we have (recall we define $\mathbb{P}^\pi f$ in (164))

$$l(f, \mathbb{P}^\pi f, \xi, \pi) = \gamma\left(f(s', a') - \mathbb{E}_{\substack{s' \sim \mathbb{P}(\cdot|s,a) \\ a' \sim \pi(\cdot|s')}}[f(s', a')]\right). \tag{178}$$

Combining (177) and (178), we deduce that for any $f, f' : \mathcal{S} \times \mathcal{A} \to \mathbb{R}$, $\xi$ and $\pi$,

$$l(f, f', \xi, \pi) - l(f, \mathbb{P}^\pi f, \xi, \pi) = \mathbb{E}_{\substack{s' \sim \mathbb{P}(\cdot|s,a) \\ a' \sim \pi(\cdot|s')}}[l(f, f', \xi, \pi)]. \tag{179}$$

**Bounding $-\mathcal{L}_t(f_t, \pi_t)$.** For any $f \in \mathcal{Q}, \pi$ and $t \in [T]$, we define $X_{f,\pi}^t$ as

$$X_{f,\pi}^t := \mathbb{E}_{a' \sim \pi(\cdot|s_t')}\left[l(f, f, \xi_t, \pi)^2 - l(f, \mathbb{P}^\pi f, \xi_t, \pi)^2\right]. \tag{180}$$

Then we have for any $f \in \mathcal{Q}$:

$$\sum_{i=1}^{t-1} X_{f,\pi}^i = \sum_{i=1}^{t-1} \mathbb{E}_{a' \sim \pi(\cdot|s_i')} l(f, f, \xi_i, \pi)^2 - \sum_{i=1}^{t-1} \mathbb{E}_{a' \sim \pi(\cdot|s_i')} l(f, \mathbb{P}^\pi f, \xi_i, \pi)^2$$

$$\leqslant \sum_{i=1}^{t-1} \mathbb{E}_{a' \sim \pi(\cdot|s_i')} l(f, f, \xi_i, \pi)^2 - \inf_{g \in \mathcal{Q}} \sum_{i=1}^{t-1} \mathbb{E}_{a' \sim \pi(\cdot|s_i')} l(f, g, \xi_i, \pi)^2 \stackrel{(159)}{=} \mathcal{L}_t(f, \pi), \tag{181}$$

where the inequality uses the fact that $\mathbb{P}^\pi f \in \mathcal{Q}$, which is guaranteed by Lemma 15. Therefore, to upper bound $-\mathcal{L}_t(f_t, \pi_t)$, we only need to bound $-\sum_{i=1}^{t-1} X_{f_t, \pi_t}^i$.

Below we use Freedman's inequality (Lemma 2) and a covering number argument to give the desired bound. Repeating a similar argument as the finite-horizon setting, for any $\epsilon > 0$, there exists an $\epsilon$-net $\Theta_\epsilon \subset \Theta$ and an $\epsilon$-net $\Omega_\epsilon \subset \Omega$ such that

$$\log|\Theta_\epsilon| \leqslant d\log\left(1 + \frac{2\sqrt{d}}{(1-\gamma)\epsilon}\right), \quad \text{and} \quad \log|\Omega_\epsilon| \leqslant d\log\left(1 + \frac{2B\sqrt{d}}{(1-\gamma)\epsilon}\right). \tag{182}$$

Let $\mathcal{Q}_\epsilon := \{f_\epsilon = f_{\theta_\epsilon} : \theta_\epsilon \in \Theta_\epsilon\}$, and $\mathcal{P}_\epsilon := \{\pi_\epsilon(a|s) = \frac{\exp(\phi(s,a)^\top \omega_\epsilon)}{\sum_{a' \in \mathcal{A}} \exp(\phi(s,a')^\top \omega_\epsilon)} : \omega_\epsilon \in \Omega_\epsilon\}$. For any $f \in \mathcal{Q}$ and $\pi \in \mathcal{P}$, there exists $f_\epsilon \in \mathcal{Q}_\epsilon$ and $\pi_\epsilon \in \mathcal{P}_\epsilon$ such that

$$\left|X_{f_\epsilon, \pi_\epsilon}^t - X_{f,\pi}^t\right| \leqslant \frac{24\epsilon}{(1-\gamma)^2}. \tag{183}$$

To invoke Freedman's inequality, we calculate the following quantities.

- Assumption 8 ensures that $X_{f,\pi}^t$ is bounded:

$$\forall f \in \mathcal{Q}: \quad |X_{f,\pi}^t| \leqslant \frac{4}{(1-\gamma)^2}. \tag{184}$$

- Repeating the argument for (59), we have

$$\mathbb{E}_{s'_t \sim \mathbb{P}(\cdot | s_t, a_t)} \left[ X^t_{f,\pi} \right] = \left( \mathbb{E}_{\substack{s'_t \sim \mathbb{P}(\cdot | s_t, a_t) \\ a' \sim \pi(\cdot | s'_t)}} \left[ l(f, f, \xi_t, \pi) \right] \right)^2 \overset{(169)}{=} \ell(f, s_t, a_t, \pi). \qquad (185)$$

Define the filtration $\mathcal{F}_t := \sigma(\mathcal{D}_t)$, then we have (recall Algorithm 3 ensures $(s_t, a_t) \sim d_\rho^{\pi_t}$)

$$\forall f \in \mathcal{Q}: \quad \mathbb{E}\left[ X^t_{f,\pi} | \mathcal{F}_{t-1} \right] = \mathbb{E}\left[ \mathbb{E}_{s'_t \sim \mathbb{P}(\cdot | s_t, a_t)} \left[ X^t_{f,\pi} \right] | \mathcal{F}_{t-1} \right] = \mathbb{E}_{(s_t, a_t) \sim d_\rho^{\pi_t}} \left[ \ell(f, s_t, a_t, \pi) \right]. \tag{186}$$

- Furthermore, we have

$$\mathsf{Var}\left[ X^t_{f,\pi} | \mathcal{F}_{t-1} \right]$$
$$\leqslant \mathbb{E}\left[ \left( X^t_{f,\pi} \right)^2 | \mathcal{F}_{t-1} \right]$$
$$= \mathbb{E}\left[ \left( \mathbb{E}_{a' \sim \pi(\cdot | s'_t)} \left[ \left( r(s_t, a_t) + \gamma f(s'_t, a') - f(s_t, a_t) \right)^2 - \gamma^2 \left( f(s'_t, a') - \mathbb{E}_{\substack{s'_t \sim \mathbb{P}(\cdot | s_t, a_t) \\ a' \sim \pi(\cdot | s'_t)}} \left[ f(s'_t, a') \right]^2 \right] \right) \right)^2 \Bigg| \mathcal{F}_{t-1} \right]$$
$$\leqslant \mathbb{E}\Bigg[ \left( r(s_t, a_t) + 2\gamma f(s'_t, a') - f(s_t, a_t) - \mathbb{E}_{\substack{s'_t \sim \mathbb{P}(\cdot | s_t, a_t) \\ a' \sim \pi(\cdot | s'_t)}} \left[ f(s'_t, a') \right] \right)^2$$
$$\cdot \left( r(s_t, a_t) + \gamma \mathbb{E}_{\substack{s'_t \sim \mathbb{P}(\cdot | s_t, a_t) \\ a' \sim \pi(\cdot | s'_t)}} \left[ f(s'_t, a') \right] - f(s_t, a_t) \right)^2 \Bigg| \mathcal{F}_{t-1} \Bigg]$$
$$\leqslant \frac{16}{(1-\gamma)^2} \mathbb{E}_{(s_t, a_t) \sim d_\rho^{\pi_t}} \left[ \ell(f, s_t, a_t, \pi) \right], \quad \forall f \in \mathcal{Q}. \tag{187}$$

where the first equality follows from (177) and (178), and the second inequality follows from Jenson's inequality.

Therefore, by Lemma 2, we have with probability at least $1 - \delta$, for all $t \in [T], f_\epsilon \in \mathcal{Q}_\epsilon, \pi_\epsilon \in \mathcal{P}_\epsilon$:

$$\sum_{i=1}^{t-1} \mathbb{E}_{(s_i, a_i) \sim d_\rho^{\pi_i}} \left[ \ell(f_\epsilon, s_i, a_i, \pi_\epsilon) \right] - \sum_{i=1}^{t-1} X^i_{f_\epsilon, \pi_\epsilon}$$
$$\leqslant \frac{1}{2} \sum_{i=1}^{t-1} \mathbb{E}_{(s_i, a_i) \sim d_\rho^{\pi_i}} \left[ \ell(f_\epsilon, s_i, a_i, \pi_\epsilon) \right] + \frac{C_1}{(1-\gamma)^2} \log(T |\Theta_\epsilon| |\Omega_\epsilon| / \delta)$$
$$\overset{(182)}{\leqslant} \frac{1}{2} \sum_{i=1}^{t-1} \mathbb{E}_{(s_i, a_i) \sim d_\rho^{\pi_i}} \left[ \ell(f_\epsilon, s_i, a_i, \pi_\epsilon) \right] + \frac{C_1}{(1-\gamma)^2} \left( d \log \left( \frac{4Bd}{(1-\gamma)^2 \epsilon^2} \right) + \log(T/\delta) \right), \tag{188}$$

where $C_1 > 0$ is an absolute constant. From (188) we deduce that for all $t \in [T]\ f_\epsilon \in \mathcal{Q}_\epsilon$, and $\pi_\epsilon \in \mathcal{P}_\epsilon$,

$$-\sum_{i=1}^{t-1} X^i_{f_\epsilon, \pi_\epsilon} \leqslant -\frac{1}{2} \sum_{i=1}^{t-1} \mathbb{E}_{(s_i, a_i) \sim d_\rho^{\pi_i}} \left[ \ell(f_\epsilon, s_i, a_i, \pi_\epsilon) \right] + \frac{C_1}{(1-\gamma)^2} \left( d \log \left( \frac{4Bd}{(1-\gamma)^2 \epsilon^2} \right) + \log(T/\delta) \right). \tag{189}$$

Note that for any $t \in [T]$, there exist $\theta_t \in \Theta$ and $\omega_t \in \Omega$ such that $f_t = f_{\theta_t} \in \mathcal{Q}$ and $\pi_t = \pi_{\omega_t} \in \mathcal{P}$. We can choose $\theta_{t,\epsilon} \in \Theta_\epsilon$ and $\omega_{t,\epsilon} \in \Omega_\epsilon$ such that $\|\theta_t - \theta_{t,\epsilon}\|_2 \leqslant \epsilon$ and $\|\omega_t - \omega_{t,\epsilon}\|_2 \leqslant \epsilon$. We let $f_{t,\epsilon} := f_{\theta_{t,\epsilon}} \in \mathcal{Q}_\epsilon$. Then by (189) we have for all $t \in [T]$,

$$-\mathcal{L}_t(f_t, \pi_t)$$
$$\overset{(181)}{\leqslant} -\sum_{i=1}^{t-1} X^i_{f_t, \pi_t}$$
$$\overset{(183)}{\leqslant} -\sum_{i=1}^{t-1} X^i_{f_{t,\epsilon}, \pi_t} + \frac{24 T \epsilon}{(1-\gamma)^2}$$

$$\overset{(189)}{\leqslant} -\frac{1}{2}\sum_{i=1}^{t-1}\mathbb{E}_{(s_i,a_i)\sim d_\rho^{\pi i}}\left[\ell(f_{t,\epsilon},s_i,a_i,\pi_{t,\epsilon})\right] + \frac{C_1}{(1-\gamma)^2}\left(d\log\left(\frac{4Bd}{(1-\gamma)^2\epsilon^2}\right) + \log(T/\delta)\right) + \frac{24T\epsilon}{(1-\gamma)^2}$$

$$\leqslant -\frac{1}{2}\sum_{i=1}^{t-1}\mathbb{E}_{(s_i,a_i)\sim d_\rho^{\pi i}}\left[\ell(f_t,s_i,a_i,\pi_t)\right] + \frac{C_1}{(1-\gamma)^2}\left(d\log\left(\frac{4Bd}{(1-\gamma)^2\epsilon^2}\right) + \log\left(\frac{T}{\delta}\right)\right) + \frac{36T\epsilon}{(1-\gamma)^2},$$
$$(190)$$

where the last line follows from (183) and (185).

**Bounding $\mathcal{L}_t(f^\star,\widetilde{\pi}^\star)$.** For any $f\in\mathcal{Q}$ and $t\in[T]$, we define

$$Y_f^t := \mathbb{E}_{a'\sim\widetilde{\pi}^\star(\cdot|s_t')}\left[l(f^\star,f,\xi_t,\widetilde{\pi}^\star)^2 - l(f^\star,\widetilde{f}^\star,\xi_t,\widetilde{\pi}^\star)^2\right], \quad\text{where}\quad \widetilde{f}^\star := \mathbb{P}^{\widetilde{\pi}^\star}f^\star. \qquad (191)$$

Note that for any tuple $\xi = (s,a,s')$, we have

$$\left|l(f^\star,f^\star,\xi,\widetilde{\pi}^\star)^2 - l(f^\star,\widetilde{f}^\star,\xi,\widetilde{\pi}^\star)^2\right| = \left|l(f^\star,f^\star,\xi,\pi^\star) + l(f^\star,\widetilde{f}^\star,\xi,\widetilde{\pi}^\star)\right|\left|l(f^\star,f^\star,\xi,\widetilde{\pi}^\star) - l(f^\star,\widetilde{f}^\star,\xi,\widetilde{\pi}^\star)\right|$$

$$\leqslant \frac{4}{1-\gamma}\left|\mathbb{E}_{\substack{s'\sim\mathbb{P}(\cdot|s,a)\\a'\sim\widetilde{\pi}^\star(\cdot|s')}}\left[l(f^\star,f^\star,\xi,\widetilde{\pi}^\star)\right]\right|, \qquad (192)$$

where the last line follows from (179). Furthermore, we have

$$\mathbb{E}_{\substack{s'\sim\mathbb{P}(\cdot|s,a)\\a'\sim\widetilde{\pi}^\star(\cdot|s')}}\left[l(f^\star,f^\star,\xi,\widetilde{\pi}^\star)\right] \overset{(177)}{=} \mathbb{E}_{\substack{s'\sim\mathbb{P}(\cdot|s,a)\\a'\sim\widetilde{\pi}^\star(\cdot|s')}}\left[r(s,a) + \gamma f^\star(s',a') - f^\star(s,a)\right]$$

$$= r(s,a) + \gamma\mathbb{E}_{s'\sim\mathbb{P}(\cdot|s,a)}\left[V_{f^\star}^{\widetilde{\pi}^\star}(s')\right] - f^\star(s,a)$$

$$= \gamma\mathbb{E}_{s'\sim\mathbb{P}(\cdot|s,a)}\left[V_{f^\star}^{\widetilde{\pi}^\star}(s')\right] - \gamma\mathbb{E}_{s'\sim\mathbb{P}(\cdot|s,a)}\left[V^{\pi^\star}(s')\right], \qquad (193)$$

where the last line uses Bellman's optimality equation

$$r(s,a) + \gamma\mathbb{E}_{s'\sim\mathbb{P}(\cdot|s,a)}\left[V^{\pi^\star}(s')\right] - f^\star(s,a) = 0. \qquad (194)$$

By Lemma 16, we have

$$\mathbb{E}_{s'\sim\mathbb{P}(\cdot|s,a)}\left[V^{\pi^\star}(s')\right] - \frac{\log|\mathcal{A}|}{B} \leqslant \mathbb{E}_{s'\sim\mathbb{P}(\cdot|s,a)}\left[V_{f^\star}^{\widetilde{\pi}^\star}(s')\right] \leqslant \mathbb{E}_{s'\sim\mathbb{P}(\cdot|s,a)}\left[V^{\pi^\star}(s')\right]. \qquad (195)$$

Plugging the above inequality into (193) and (192), we have

$$\left|l(f^\star,f^\star,\xi,\widetilde{\pi}^\star)^2 - l(f^\star,\widetilde{f}^\star,\xi,\widetilde{\pi}^\star)^2\right| \leqslant \frac{4\gamma}{1-\gamma}\frac{\log|\mathcal{A}|}{B}. \qquad (196)$$

The above bound (196) implies that

$$\mathcal{L}_t(f^\star,\widetilde{\pi}^\star) = \sum_{i=1}^{t-1}\mathbb{E}_{a'\sim\pi^\star(\cdot|s_i')}l(f^\star,f^\star,\xi_i,\widetilde{\pi}^\star)^2 - \inf_{g\in\mathcal{Q}}\sum_{i=1}^{t-1}\mathbb{E}_{a'\sim\pi^\star(\cdot|s_i')}l(f^\star,g,\xi_i,\widetilde{\pi}^\star)^2$$

$$\leqslant \sup_{f\in\mathcal{Q}}\sum_{i=1}^{t-1}\left(-Y_f^i\right) + \frac{4\gamma T}{1-\gamma}\frac{\log|\mathcal{A}|}{B}, \qquad (197)$$

where we also use the definitions of $Y_f^t$, $\widetilde{f}^\star$ (c.f. (191)), and $\mathcal{L}_t$ (c.f. (159)). Thus to bound $\mathcal{L}_t(f^\star,\widetilde{\pi}^\star)$, below we bound the sum $\sum_{i=1}^{t-1}Y_f^i$ for any $f\in\mathcal{Q}$ and $t\in[T]$. To invoke Freedman?s inequality, we calculate the following quantities.

• Repeating the argument for (59), we have

$$\mathbb{E}_{s_t'\sim\mathbb{P}(\cdot|s_t,a_t)}\left[Y_f^t\right] = \left(\mathbb{E}_{\substack{s_t'\sim\mathbb{P}(\cdot|s_t,a_t)\\a'\sim\widetilde{\pi}^\star(\cdot|s_t')}}\left[l(f^\star,f,\xi_t,\widetilde{\pi}^\star)\right]\right)^2, \qquad (198)$$

which implies

$$\forall f\in\mathcal{Q}: \quad \mathbb{E}\left[Y_f^t|\mathcal{F}_{t-1}\right] = \mathbb{E}_{(s_t,a_t)\sim d_\rho^{\pi t}}\left[\left(\mathbb{E}_{\substack{s_t'\sim\mathbb{P}(\cdot|s_t,a_t)\\a'\sim\widetilde{\pi}^\star(\cdot|s_t')}}\left[l(f^\star,f,\xi_t,\widetilde{\pi}^\star)\right]\right)^2\right]. \qquad (199)$$

- We have

$$\mathsf{Var}\left[Y_f^t|\mathcal{F}_{t-1}\right] \leqslant \mathbb{E}\left[\left(Y_f^t\right)^2|\mathcal{F}_{t-1}\right]$$

$$= \mathbb{E}\left[\left(\mathbb{E}_{a'\sim\widetilde{\pi}^\star(\cdot|s_t')}\left[\left(r(s_t,a_t)+\gamma f^\star(s_t',a')-f(s_t,a_t)\right)^2\right.\right.\right.$$

$$\left.\left.\left. -\gamma^2\left(f^\star(s_t',a')-\mathbb{E}_{\substack{s_t'\sim\mathbb{P}(\cdot|s_t,a_t)\\a'\sim\widetilde{\pi}^\star(\cdot|s_t')}}\left[f^\star(s_t',a')\right]\right)^2\right]\right)^2\Bigg|\mathcal{F}_{t-1}\right]$$

$$\leqslant \mathbb{E}\left[\left(r(s_t,a_t)+2\gamma f^\star(s_t',a')-f(s_t,a_t)-\mathbb{E}_{\substack{s_t'\sim\mathbb{P}(\cdot|s_t,a_t)\\a'\sim\widetilde{\pi}^\star(\cdot|s_t')}}\left[f^\star(s_t',a')\right]\right)^2\right.$$

$$\left.\cdot\left(r(s_t,a_t)+\gamma\mathbb{E}_{\substack{s_t'\sim\mathbb{P}(\cdot|s_t,a_t)\\a'\sim\widetilde{\pi}^\star(\cdot|s_t')}}\left[f^\star(s_t',a')\right]-f(s_t,a_t)\right)^2\Bigg|\mathcal{F}_{t-1}\right]$$

$$\leqslant \frac{16}{(1-\gamma)^2}\mathbb{E}_{(s_t,a_t)\sim d_\rho^{\pi_t}}\left[\left(\mathbb{E}_{\substack{s_t'\sim\mathbb{P}(\cdot|s_t,a_t)\\a'\sim\widetilde{\pi}^\star(\cdot|s_t')}}\left[l(f^\star,f,\xi,\widetilde{\pi}^\star)\right]\right)^2\right], \tag{200}$$

where the first line uses (by (178))

$$l(f^\star,\widetilde{f}^\star,\xi_t,\pi^\star) = \gamma\left(f^\star(s_t',a')-\mathbb{E}_{\substack{s_t'\sim\mathbb{P}(\cdot|s_t,a_t)\\a'\sim\widetilde{\pi}^\star(\cdot|s_t')}}\left[f^\star(s_t',a')\right]\right), \tag{201}$$

where $a'\sim\widetilde{\pi}^\star(\cdot|s_t')$, and the second inequality uses Jenson's inequality.

- Last but not least, it's easy to verify that

$$|Y_f^t| \leqslant \frac{4}{(1-\gamma)^2}. \tag{202}$$

Invoking Lemma 2, and setting $\eta$ in Lemma 2 as

$$\eta = \min\left\{\frac{(1-\gamma)^2}{4}, \sqrt{\frac{\log(|\Theta_\epsilon|T/\delta)}{\sum_{i=1}^{t-1}\mathsf{Var}\left[Y_f^i|\mathcal{F}_{i-1}\right]}}\right\}$$

for each $f_\epsilon\in\mathcal{Q}_\epsilon$, we have with probability at least $1-\delta$,

$$\forall f_\epsilon\in\mathcal{Q}_\epsilon, t\in[T]:\quad \sum_{i=1}^{t-1}\left(-Y_{f_\epsilon}^i+\mathbb{E}_{(s_i,a_i)\sim d_\rho^{\pi_i}}\left[\left(\mathbb{E}_{\substack{s_i'\sim\mathbb{P}(\cdot|s_i,a_i)\\a'\sim\widetilde{\pi}^\star(\cdot|s_i')}}\left[l(f^\star,f_\epsilon,\xi_i,\widetilde{\pi}^\star)\right]\right)^2\right]\right)$$

$$\lesssim \frac{1}{1-\gamma}\sqrt{\log(|\Theta_\epsilon|T/\delta)\sum_{i=1}^{t-1}\mathbb{E}_{(s_i,a_i)\sim d_\rho^{\pi_i}}\left[\left(\mathbb{E}_{\substack{s_i'\sim\mathbb{P}(\cdot|s_i,a_i)\\a'\sim\widetilde{\pi}^\star(\cdot|s_i')}}\left[l(f^\star,f_\epsilon,\xi_i,\widetilde{\pi}^\star)\right]\right)^2\right]}$$

$$+\frac{1}{(1-\gamma)^2}\log(|\Theta_\epsilon|T/\delta). \tag{203}$$

Reorganizing the above inequality, we have for any $f_\epsilon\in\mathcal{Q}_\epsilon, t\in[T]$:

$$\sum_{i=1}^{t-1}\left(-Y_{f_\epsilon}^i\right) \lesssim \frac{1}{(1-\gamma)^2}\log(|\Theta_\epsilon|T/\delta)-\sum_{i=1}^{t-1}\mathbb{E}_{(s_i,a_i)\sim d_\rho^{\pi_i}}\left[\left(\mathbb{E}_{\substack{s_i'\sim\mathbb{P}(\cdot|s_i,a_i)\\a'\sim\widetilde{\pi}^\star(\cdot|s_i')}}\left[l(f^\star,f_\epsilon,\xi_i,\widetilde{\pi}^\star)\right]\right)^2\right]$$

$$+\frac{1}{1-\gamma}\sqrt{\log(|\Theta_\epsilon|T/\delta)\sum_{i=1}^{t-1}\mathbb{E}_{(s_i,a_i)\sim d_\rho^{\pi_i}}\left[\left(\mathbb{E}_{\substack{s_i'\sim\mathbb{P}(\cdot|s_i,a_i)\\a'\sim\widetilde{\pi}^\star(\cdot|s_i')}}\left[l(f^\star,f_\epsilon,\xi_i,\widetilde{\pi}^\star)\right]\right)^2\right]}$$

$$\lesssim \frac{1}{(1-\gamma)^2}\log(|\Theta_\epsilon|T/\delta), \tag{204}$$

where the last line makes use of the fact that $-x^2 + bx \leqslant b^2/4$.

Moreoever, for any $t \in [T]$, we have

$$Y_{f_\epsilon}^t - Y_f^t$$

$$= \mathbb{E}_{a' \sim \widetilde{\pi}^\star(\cdot|s_t')} \left[ \left( r(s_t, a_t) + \gamma f^\star(s_t', a') - f_\epsilon(s_t, a_t) \right)^2 - \left( r(s_t, a_t) + \gamma f^\star(s_t', a') - f(s_t, a_t) \right)^2 \right]$$

$$= \mathbb{E}_{a' \sim \widetilde{\pi}^\star(\cdot|s_t')} \left[ \left( 2r(s_t, a_t) + 2\gamma f^\star(s_t', a') - f_\epsilon(s_t, a_t) - f(s_t, a_t) \right) \cdot \left( f(s_t, a_t) - f_\epsilon(s_t, a_t) \right) \right] \leqslant \frac{4\epsilon}{1-\gamma},$$
(205)

where the last inequality uses $|f(s, a) - f_\epsilon(s, a)| \leqslant \|\phi(s, a)\|_2 \|\theta - \theta_\epsilon\|_2 \leqslant \epsilon$. Combining (204) and (205), we have with probability at least $1 - \delta$, for any $t \in [T]$ and $f \in \mathcal{Q}$,

$$\sum_{i=1}^{t-1} \left( -Y_f^i \right) \leqslant \frac{C_2}{(1-\gamma)^2} \log(|\Theta_\epsilon|T/\delta) + \frac{4\epsilon T}{1-\gamma}$$

$$\overset{(182)}{\leqslant} \frac{C_2}{(1-\gamma)^2} \left( d \log \left( 1 + \frac{2\sqrt{d}}{(1-\gamma)\epsilon} \right) + \log(T/\delta) \right) + \frac{4\epsilon T}{1-\gamma}, \qquad (206)$$

where $C_2 > 0$ is an absolute constant.

By (197) we have

$$\mathcal{L}_t(f^\star, \widetilde{\pi}^\star) \leqslant \frac{C_2}{(1-\gamma)^2} \left( d \log \left( 1 + \frac{2\sqrt{d}}{(1-\gamma)\epsilon} \right) + \log(T/\delta) \right) + \frac{4T}{1-\gamma} \left( \epsilon + \frac{\log |\mathcal{A}|}{B} \right). \quad (207)$$

**Combining the two bounds.** Combining (190) and (207), we have for any $t \in [T]$,

$$\mathcal{L}_t(f^\star, \widetilde{\pi}^\star) - \mathcal{L}_t(f_t, \pi_t) \leqslant -\frac{1}{2} \sum_{i=1}^{t-1} \mathbb{E}_{(s_i, a_i) \sim d_\rho^{\pi_i}} \left[ \ell(f_t, s_i, a_i, \pi_t) \right]$$

$$+ \frac{C}{(1-\gamma)^2} \left( d \log \left( \frac{Bd}{(1-\gamma)\epsilon} \right) + \log \left( \frac{T}{\delta} \right) + T\epsilon + (1-\gamma) \frac{T \log |\mathcal{A}|}{B} \right)$$
(208)

for some absolute constant $C > 0$. Letting $\epsilon = \frac{1}{T}$, we obtain the desired result.

### C.4.2 Proof of Lemma 18

First note that for any policy $\pi$ and $f : \mathcal{S} \times \mathcal{A} \to \mathbb{R}$, we have

$$V_f^\pi(\rho) = \mathbb{E}_{\substack{s_0 \sim \rho, a_h \sim \pi(\cdot|s_h) \\ s_{h+1} \sim \mathbb{P}(\cdot|s_h, a_h), \forall h \in \mathbb{N}}} \left[ \sum_{h=0}^\infty \left( \gamma^h V_f^\pi(s_h) - \gamma^{h+1} V_f^\pi(s_{h+1}) \right) \right]$$

$$= \mathbb{E}_{\substack{s_0 \sim \rho, a_h \sim \pi(\cdot|s_h) \\ s_{h+1} \sim \mathbb{P}(\cdot|s_h, a_h), \forall h \in \mathbb{N}}} \left[ \sum_{h=0}^\infty \gamma^h \left( Q_f(s_h, a_h) - \gamma V_f^\pi(s_{h+1}) \right) \right], \qquad (209)$$

and

$$V^\pi(\rho) = \mathbb{E}_{\substack{s_0 \sim \rho, a_h \sim \pi(\cdot|s_h) \\ s_{h+1} \sim \mathbb{P}(\cdot|s_h, a_h), \forall h \in \mathbb{N}}} \left[ \sum_{h=0}^\infty \gamma^h r(s_h, a_h) \right]. \qquad (210)$$

The above two expressions (209) and (210) together give that

$$V_f^\pi(\rho) - V^\pi(\rho) = \mathbb{E}_{\substack{s_0 \sim \rho, a_h \sim \pi(\cdot|s_h) \\ s_{h+1} \sim \mathbb{P}(\cdot|s_h, a_h), \forall h \in \mathbb{N}}} \left[ \sum_{h=0}^\infty \gamma^h \left( Q_f(s_h, a_h) - r(s_h, a_h) - \gamma V_f^\pi(s_{h+1}) \right) \right]$$

$$= \frac{1}{1-\gamma} \mathbb{E}_{(s,a) \sim d_\rho^\pi} \Big[ \underbrace{Q_f(s, a) - r(s, a) - \gamma \mathbb{P} V_f^\pi(s, a)}_{:=\mathcal{E}(f, s, a, \pi)} \Big], \qquad (211)$$

where we define

$$\mathbb{P}V_f^\pi(s, a) := \mathbb{E}_{s' \sim \mathbb{P}(\cdot|s,a)} \left[ V_f^\pi(s') \right], \tag{212}$$

and

$$\mathcal{E}(f, s, a, \pi) := Q_f(s, a) - r(s, a) - \gamma \mathbb{P}V_f^\pi(s, a). \tag{213}$$

By Assumption 8, for any $f \in \mathcal{Q}$, there exists $\theta_f \in \Theta$ such that $f(s, a) = \langle \theta_f, \phi(s, a) \rangle$. Thus we have

$$\mathcal{E}(f, s, a, \pi) = \phi(s, a)^\top \underbrace{\left( \theta_f - \zeta - \int_{\mathcal{S}} V_f^\pi(s') d\mu(s') \right)}_{W(f, \pi)}, \tag{214}$$

where $W(f, \pi)$ satisfies

$$\forall f \in \mathcal{Q}, \pi \in \Pi: \quad \|W(f, \pi)\|_2 \leqslant \frac{3}{1 - \gamma} \sqrt{d} \tag{215}$$

under Assumption 8. We define

$$x(\pi) := \frac{1}{1 - \gamma} \mathbb{E}_{(s,a) \sim d_\rho^\pi} \left[ \phi(s, a) \right]. \tag{216}$$

Then we have

$$V_f^\pi(\rho) - V^\pi(\rho) = \frac{1}{1 - \gamma} \mathbb{E}_{(s,a) \sim d_\rho^\pi} \left[ \mathcal{E}(f, s, a, \pi) \right] = \langle x(\pi), W(f, \pi) \rangle. \tag{217}$$

For all $t \in [T]$, we define

$$\Lambda_t(\lambda) := \lambda I_d + \sum_{i=1}^{t-1} x(\pi_i) x(\pi_i)^\top, \ \forall \lambda > 0, \tag{218}$$

where $I_d$ is the $d \times d$ identity matrix. Then by Lemma 4, we have

$$\sum_{i=1}^{t} \min \left\{ \|x(\pi_i)\|_{\Lambda_i(\lambda)^{-1}}, 1 \right\} \leqslant 2 \log \left( \det \left( I_d + \frac{1}{\lambda} \sum_{i=1}^{t-1} x(\pi_i) x(\pi_i)^\top \right) \right). \tag{219}$$

Further, we could use Lemma 5 to bound the last term in (219), and obtain

$$\forall t \in [T]: \quad \sum_{i=1}^{t} \min \left\{ \|x(\pi_i)\|_{\Lambda_i(\lambda)^{-1}}, 1 \right\} \leqslant 2 d_\gamma(\lambda), \tag{220}$$

where in the last line, we use the definition of $d_\gamma(\lambda)$ (c.f. (172)) and the fact that

$$\|x(\pi)\|_2 \leqslant \frac{1}{1 - \gamma}, \tag{221}$$

which is ensured by Assumption 8.

Observe that

$$\sum_{t=1}^{T} \left| V_{f_t}^{\pi_t}(\rho) - V^{\pi_t}(\rho) \right| \overset{(211)}{=} \frac{1}{1 - \gamma} \sum_{t=1}^{T} \left| \mathbb{E}_{(s,a) \sim d_\rho^{\pi_t}} \left[ \mathcal{E}(f_t, s, a, \pi_t) \right] \right|$$

$$\overset{(214)}{=} \sum_{t=1}^{T} |\langle x(\pi_t), W(f_t, \pi_t) \rangle|$$

$$= \underbrace{\sum_{t=1}^{T} |\langle x(\pi_t), W(f_t, \pi_t) \rangle| \, \mathbf{1} \left\{ \|x(\pi_t)\|_{\Lambda_t(\lambda)^{-1}} \leqslant 1 \right\}}_{(a)}$$

$$+ \underbrace{\sum_{t=1}^{T} |\langle x(\pi_t), W(f_t, \pi_t) \rangle| \, \mathbf{1} \left\{ \|x(\pi_t)\|_{\Lambda_t(\lambda)^{-1}} > 1 \right\}}_{(b)}, \tag{222}$$

where $\mathbf{1}\{\cdot\}$ is the indicator function.

To give the desired bound, we will bound (a) and (b) separately.

**Bounding (a).** We have for any $\lambda > 0$,

$$\text{(a)} \leq \sum_{t=1}^{T} \|W(f_t, \pi_t)\|_{\Lambda_t(\lambda)} \|x(\pi_t)\|_{\Lambda_t(\lambda)^{-1}} \mathbf{1}\left\{\|x(\pi_t)\|_{\Lambda_t(\lambda)^{-1}} \leq 1\right\}$$

$$\leq \sum_{t=1}^{T} \|W(f_t, \pi_t)\|_{\Lambda_t(\lambda)} \min\left\{\|x(\pi_t)\|_{\Lambda_t(\lambda)^{-1}}, 1\right\}. \tag{223}$$

$\|W(f_t, \pi_t)\|_{\Lambda_t(\lambda)}$ can be bounded as follows:

$$\|W(f_t, \pi_t)\|_{\Lambda_t(\lambda)} \leq \sqrt{\lambda} \cdot \frac{3\sqrt{d}}{1-\gamma} + \left(\sum_{i=1}^{t-1} |\langle x(\pi_i), W(f_t, \pi_t)\rangle|^2\right)^{1/2}, \tag{224}$$

where we use (215), (218) and the fact that $\sqrt{a+b} \leq \sqrt{a} + \sqrt{b}$ for any $a, b \geq 0$.

(223) and (224) together give

$$\text{(a)} \leq \sum_{t=1}^{T} \left(\sqrt{\lambda} \cdot \frac{3\sqrt{d}}{1-\gamma} + \left(\sum_{i=1}^{t-1} |\langle x(\pi_i), W(f_t, \pi_t)\rangle|^2\right)^{1/2}\right) \min\left\{\|x(\pi_t)\|_{\Lambda_t(\lambda)^{-1}}, 1\right\}$$

$$\leq \underbrace{\left(\sum_{t=1}^{T} \lambda \cdot \frac{9d}{(1-\gamma)^2}\right)^{1/2} \left(\sum_{t=1}^{T} \min\left\{\|x(\pi_t)\|_{\Lambda_t(\lambda)^{-1}}, 1\right\}\right)^{1/2}}_{\text{(a-i)}}$$

$$+ \underbrace{\left(\sum_{t=1}^{T} \sum_{i=1}^{t-1} |\langle x(\pi_i), W(f_t, \pi_t)\rangle|^2\right)^{1/2} \left(\sum_{t=1}^{T} \min\left\{\|x(\pi_t)\|_{\Lambda_t(\lambda)^{-1}}, 1\right\}\right)^{1/2}}_{\text{(a-ii)}}, \tag{225}$$

where in the second inequality we use Cauchy-Schwarz inequality and the fact that

$$\forall t \in [T]: \quad \min\left\{\|x(\pi_t)\|_{\Lambda_t(\lambda)^{-1}}, 1\right\}^2 \leq \min\left\{\|x(\pi_t)\|_{\Lambda_t(\lambda)^{-1}}, 1\right\}. \tag{226}$$

(a-i) in (225) could be bounded as follows:

$$\text{(a-i)} \overset{(220)}{\leq} 3\sqrt{\frac{\lambda dT}{(1-\gamma)^2} \cdot 2d_\gamma(\lambda)}. \tag{227}$$

To bound (a-ii), note that for any $\pi, \pi' \in \Pi$, we have

$$|\langle x(\pi'), W(f, \pi)\rangle|^2 = \frac{1}{(1-\gamma)^2} \left|\mathbb{E}_{(s,a)\sim d_\rho^{\pi'}}\left[Q_f(s,a) - r(s,a) - \gamma \mathbb{P}V_f^\pi(s,a)\right]\right|^2$$

$$\leq \frac{1}{(1-\gamma)^2} \mathbb{E}_{(s,a)\sim d_\rho^{\pi'}}\left[\ell(f, s, a, \pi)\right], \tag{228}$$

where the inequality follows from Jenson's inequality, and recall $\ell(f, s, a, \pi)$ is defined in (169). Combining (228) and (220), we could bound (a-ii) in (225) as follows:

$$\text{(a-ii)} \leq \frac{1}{1-\gamma}\left(2d_\gamma(\lambda) \sum_{t=1}^{T}\sum_{i=1}^{t-1} \mathbb{E}_{(s_i,a_i)\sim d_\rho^{\pi_i}}\ell(f_t, s_i, a_i, \pi_t)\right)^{1/2}. \tag{229}$$

Plugging (227) and (229) into (225), we have

$$\text{(a)} \leq \frac{3}{1-\gamma}\sqrt{\lambda dT \cdot 2d_\gamma(\lambda)} + \frac{1}{1-\gamma}\left(2d_\gamma(\lambda) \sum_{t=1}^{T}\sum_{i=1}^{t-1} \mathbb{E}_{(s_i,a_i)\sim d_\rho^{\pi_i}}\ell(f_t, s_i, a_i, \pi_t)\right)^{1/2}. \tag{230}$$

**Bounding (b).** By Assumption 8 and (217), we have

$$\forall \pi \in \Pi : \quad |\langle x(\pi), W(f, \pi) \rangle| \leqslant \frac{2}{1 - \gamma}. \tag{231}$$

Combining the above inequality with (220), we have

$$(\mathrm{b}) \leqslant \frac{4}{1 - \gamma} d_\gamma(\lambda). \tag{232}$$

**Combining (a) and (b).** Plugging (230) and (232) into (222), we have

$$\sum_{t=1}^{T} \left| V_{f_t}^{\pi_t}(\rho) - V^{\pi_t}(\rho) \right|$$

$$\leqslant \frac{3}{1 - \gamma} \sqrt{\lambda dT \cdot 2 d_\gamma(\lambda)} + \frac{1}{1 - \gamma} \left( 2 d_\gamma(\lambda) \sum_{t=1}^{T} \sum_{i=1}^{t-1} \mathbb{E}_{(s_i, a_i) \sim d_\rho^{\pi_i}} \ell(f_t, s_i, a_i, \pi_t) \right)^{1/2} + \frac{4}{1 - \gamma} d_\gamma(\lambda). \tag{233}$$

The first term in the right hand side of (233) could be bounded as

$$\frac{3}{1 - \gamma} \sqrt{\lambda dT \cdot 2 d_\gamma(\lambda)} \leqslant \frac{3}{2(1 - \gamma)} \left( \lambda dT + 2 d_\gamma(\lambda) \right), \tag{234}$$

and the second term in the right hand side of (233) could be bounded as

$$\frac{1}{1 - \gamma} \left( 2 d_\gamma(\lambda) \sum_{t=1}^{T} \sum_{i=1}^{t-1} \mathbb{E}_{(s_i, a_i) \sim d_\rho^{\pi_i}} \ell(f_t, s_i, a_i, \pi_t) \right)^{1/2}$$

$$\leqslant \frac{d_\gamma(\lambda)}{\eta(1 - \gamma)} + \frac{\eta}{1 - \gamma} \cdot \sum_{t=1}^{T} \sum_{i=1}^{t-1} \mathbb{E}_{(s_i, a_i) \sim d_\rho^{\pi_i}} \ell(f_t, s_i, a_i, \pi_t), \tag{235}$$

for any $\eta > 0$, where in both (234) and (235), we use the fact that $\sqrt{ab} \leqslant \frac{a+b}{2}$ for any $a, b \geqslant 0$. Substituting (234) and (235) into (233) and reorganizing the terms, we have

$$\sum_{t=1}^{T} \left| V_{f_t}^{\pi_t}(\rho) - V^{\pi_t}(\rho) \right|$$

$$\leqslant \frac{\eta}{1 - \gamma} \cdot \sum_{t=1}^{T} \sum_{i=1}^{t-1} \mathbb{E}_{(s_i, a_i) \sim d_\rho^{\pi_i}} \ell(f_t, s_i, a_i, \pi_t) + \left( \frac{7}{1 - \gamma} + \frac{1}{\eta(1 - \gamma)} \right) d_\gamma(\lambda) + \frac{3T d\lambda}{2(1 - \gamma)}. \tag{236}$$

This gives the desired result.

