# OpenReview forum: "Exploration from a Primal-Dual Lens: Value-Incentivized Actor-Critic Methods for Sample-Efficient Online RL"
_NeurIPS.cc/2025/Conference — NeurIPS 2025 poster_

### Official Review · Reviewer_bBXP · 2025-06-26

**Clarity:** 2
**Significance:** 1
**Originality:** 2
**Rating:** 4
**Confidence:** 3

**Summary:**

The paper addresses the need for sample efficient online RL algorithms in the presence of general function approximation. The authors first provide an alternate interpretation of the Maximize-to-Explore (MEX) algorithm (the model-free variant) as the Lagrangian of a constrained optimization problem with the Bellman optimality equation as the constraint. By changing the constraint to the Bellman consistency equation, they develop an actor-critic algorithm which is more versatile than the inner-loop subroutine of MEX. The authors prove theoretical regret guarantees of their VAC algorithm for linear MDPs, including finite and infinite horizon. They do not provide experiments.

**Questions:**

My questions follow from the above comments so please feel free to respond to related points at the same time.
1. Can you compare your theoretical guarantees with the results of existing algorithms?
2. Can you provide additional details explaining why the MEX subproblem cannot be implemented easily with standard toolkits?
3. Can you explain how the results in the linear MDP setting can be extended to broader settings?

Small edits:
* Line 207: incomplete sentence
* Line 246: "but with practical implementation generalizable to arbitrary function approximator" I am not sure what this means.
* Line 49: The paper argues that MEX uses the *optimal* value function in the loss term. However, this terminology is confusing as MEX is not using $V^*$ (line 149), but rather $\max_{a \in \mathcal{A}} Q_{f, h} (\cdot, a)$, i.e., the state value function derived from $Q_{f, h}$.

**Ethical Concerns:**

["NO or VERY MINOR ethics concerns only"]

**Final Justification:**

The paper makes a meaningful contribution by providing an actor-critic formulation of the MEX algorithm, which is computationally tractable and theoretically backed. While their initial work only considered the linear case, in the rebuttal they show that it can be extended beyond this setting.

**Limitations:**

yes

**Quality:**

2

**Strengths And Weaknesses:**

Strengths
1. The interpretation of MEX as a constrained optimization problem is useful and opens avenues for potential future algorithmic innovation.
1. An actor-critic algorithm is more versatile and potentially easier to implement than a double-loop algorithm.
2. The paper is well-organized and clearly written.

Weaknesses
1. The paper lacks comparison with theoretical guarantees of other methods.
2. Since the main advantage of VAC lies in its algorithmic practicality, it would be appropriate to provide experiments and implementation details to strengthen this claim. This is especially since we do not have theoretical guarantees about its performance in nonlinear (neural network) settings compared to existing algorithms.
3. The paper is not clear about the weaknesses of MEX and why it requires nonstandard implementation. While I understand that MEX has a double-loop structure, I do not see why the inner-loop would be computationally intractable (line 194).
4. The biggest weakness is that the theoretical analysis applies only to linear MDPs, for which we already have good methods for balancing exploration and exploitation, whereas the work is motivated by developing practical solutions for RL settings with general function approximation. For example, the original MEX paper considers low-GEC MDPs, a broader class of problems that include linear MDPs, linear mixture MDPs, MDPs of low witness rank, MDPs of low Bellman eluder dimension, and MDPs of bilinear class. While I recognize that an actor-critic method can be more challenging to analyze than the original double-loop method, the results of this paper would be significantly strengthened if a broader theoretical setting were considered.

---

> ### Author Rebuttal · Authors · 2025-07-31
>
> ## Response to Reviewer bBXP
>
> Thank you for your time reviewing our paper. If our responses resolve your questions, we'd appreciate your consideration in raising the score.
>
> >**W1/Q1: The paper lacks comparison with theoretical guarantees of other methods...Can you compare your theoretical guarantees with the results of existing algorithms?**
>
> Thank you for your question.
>
> - **Our regret bound is near-optimal.** As stated in line 244-246 of our paper, our $\tilde{O}(dH^2\sqrt{T})$ regret bound, compared to the minimax lower bound $\tilde{O}(dH^{3/2}\sqrt{T})$, is *near-optimal* up to a factor of $\sqrt{H}$.
>
> - **We have the same regret bound as MEX under linear MDP.** MEX doesn't explicitly provide a regret bound for the linear MDP setting, and assumes a finite function class. Addapting their Corollary 5.2 to linear MDPs, and through a covering number argument on the function class, their regret can be shown to have the same order as ours ($\tilde{O}(dH^2\sqrt{T})$) (in their Corollary 5.2, $B_l=O(H)$, $d_{GEC}(1/\sqrt{HK})=\tilde{O}(Hd)$, and $\log(|\mathcal{H}|)=\tilde{O}(Hd)$).
>
> - **Extending to general FA setting, our regret also matches MEX.** Under similar assumptions as in MEX, we can give a regret bound same as MEX under general funcion approximation. See our response to Weakness 4/ Question 3 you arise for more details.
>
> We'll add these discussion on our revised paper. Thank you for your suggestion.
>
> >**W2: regarding experiments and implementation details**
>
> Thank you for your suggestion. First, We conducted experiments on two difficult MuJoCo continuous control tasks: Walker2d and Ant. We benchmark VAC against the state-of-the-art Soft Actor-Critic (SAC) algorithm [Haarnoja et al., 2018]. The average return and standard deviation of the algorithms over 3 random seeds after 1000,000 training steps are summerized in the following table:
>
> | env / alg | SAC | VAC (ours, $\alpha=1000$) | VAC (ours, $\alpha=500$) |
> |---|---|---|---|
> | Ant-v4 | 3163.86 $\pm$ 268.90 | 3974.44 $\pm$ 390.95 | **5134.55** $\pm$ 424.42 |
> | Walker2D-v4 | 4394.68 $\pm$ 181.42 | **4609.53** $\pm$ 333.98 | 4434.75 $\pm$ 117.34 |
>
> The results demonstrate VAC outperforms the strong SAC baseline.
>
> - **Implementation details**: we use a SAC style implementation with a different Q-learning objective. To be specific:
>
>     - We perform alternating single gradient (Adam) steps to update actor (policy) and critic (Q-function) by solving an approximation version of our objective (17). Same as the implementation of SAC and MEX (see their Section 7.2), we drop the second term in the expression of $L_t(f,\pi)$ (c.f. Eq.(18)) and use a standard TD error. And we use $1/n\sum_{i=1}^n Q_{f,1}(s,a_i)$ ($a_i\sim\pi(\cdot|s)$) to approximate the value-incentivized regularization term $V_{f}^\pi(s)$ (in our experiment, we set $n=1$, which is good enough).
>
>     - Non-linear FA: we use standard MLPs to parameterize both actor and critic, with 2 hidden layers of 256 neurons each, using the ReLU activation.
>
> - **Small theory-to-practice gap:** We'd like to highlight that VAC's theoretical formulation natually yields an actor-critic algorithm, leading to a very small theory-to-practice gap. This can be contrasted with prior work like MEX, where the practical actor-critic implementation used in their experiments (see their Section 7.2) differs significantly from their proposed theoretical bi-level objective.
>
> >**W3/Q2: regrading MEX's implementation difficulty**
>
> Thank you for pointing out the implementation gap in MEX. Below we elaborate on why the inner “maximize over actions” sub‐problem in (7) is not readily handled by off‑the‑shelf toolkits, and how in practice the authors end up sidestepping it in their code.
>
> - **The inner maximization $\max_{a} Q(s,a)$ can be intractable itself.** The MEX objective in Eq.(7) requires computing $\max_{a}Q_{f,h+1}(s_{h+1},a')$ for every transition $(s_h,a_h,s_{h+1})$ in the dataset at every gradient step. This is a signficant challenge in many settings:
>     - Large discrete action spaces: In domains like large language models, the action space (e.g., all possible tokens or sentences) is enormous. An exhaustive search for the maximum is computationally infeasible.
>     - Continuous action spaces: If $Q(s,\cdot)$ is non-convex, finding the global maximum is an NP-hard problem. Even with convex Q-functions, this requires running a separate optimization procedure for every data point in the batch, which is computationally expensive.
>
> - **The bilevel optimization problem is intractable to solve.** To minimize the objective $L_t(f)$ given in our Eq.(7) thourgh gradient-based methods, we need to compute $\nabla_\theta [\max_{a} Q(s,a;\theta)]$, which could be intractable and could not be handled by standard toolkits. Note that simply computing $a^\star:=\arg\max_{a} Q(s,a)$ and computing $\nabla_\theta Q(s,a^\star;\theta)$ can lead to wrong gradients of the Q-function, because $\nabla_\theta [\max_{a} Q(s,a;\theta)]$ does not always equal to $\nabla_\theta  Q(s,a^\star;\theta)$, where $a^\star:=\arg\max_a\nabla_\theta  Q(s,a;\theta)$. A simple example is when
>     $$Q(a;\theta)=\begin{cases} a,\quad\text{when $a\leq \theta$},\\
>     0, \quad\text{when $a> \theta$}. \end{cases}$$
> Then when $\theta>0$, $\nabla_\theta [\max_{a} Q(s,a;\theta)]=1$, but $\nabla_\theta  Q(s,a^\star;\theta)=0$.
>
> - **The original MEX Implementation sidesteps this problem.** Crucially, the authors of MEX themselves acknowledge this difficulty in their practical implementation, and circumvent the bilvel optimization entirely. Instead, they resort to an actor-critic framework, creating a significant disconnect between their theoretically analyzed objective and  practical implementation.
>
>     We'll include the above discussion in our revised paper. Thank you for your comment.
>
> >**W4/Q3: regarding extending the theory to the general FA setting**
>
> Thank you for your question. Our analysis is readily extended to the general function approximation setting under similar assumptions made in MEX.
>
> - Assumptions:
>     1. (Assumption 3.1 in MEX) The Q-function class $\mathcal{Q}$ satisfies $Q^\star\in\mathcal{Q}$ (realizability), and for any $\pi\in\mathcal{P}$, $f\in\mathcal{Q}$, $P^{\pi} f\in\mathcal{Q}$ (Bellman completeness).
>
>     From Lemma 10 in our paper we can see that Assumption 1 here holds under linear MDPs.
>
>     2. We set policy class $\mathcal{P}$ as
>     $$ P_h :=[\pi_h:\pi_h(s,a)=\frac{\exp(BQ_h(s,a))}{\sum_{a'}\exp(BQ_h(s,a'))}, \forall Q_h\in\mathcal{Q}_h]$$
>     with some constant $B>0$.
>
>     3. (Generalized Eluder coefficient (GEC), Assumption 4.2 in MEX) Given any $\lambda>0$, there exists $d(\lambda)\in R_+$ such that for any sequence $f_t$ in  Q and $\pi_t$ in P ($t=1,...,T$), we have $$\sum_{t=1}^T (V_{f_t}^{\pi_t}(\rho)-V_{f_t}(\rho))\leq \inf_{\eta>0}\eta\sum_{t=1}^T\sum_{i=1}^{t-1}\sum_{h=1}^H E_{(s_i,a_i)\sim d_{\rho,h}^{\pi_i}}\ell_h(f_t,s_i,a_i,\pi_t)+d(\lambda)/\eta +\sqrt{d(\lambda)HT}+\lambda HT$$ for each $\lambda>0$, we denote the smallest $d(\lambda)$ that makes the above hold as $d_{GEC}(\lambda)$.
>
>     From Lemma 4 we can see that under linear MDPs, Assumption 3 here holds with $d_{GEC}(\lambda)\lesssim Hd$.
>
>     4. $\mathcal{Q}$ has finite $\epsilon$-covering number.
>
>     MEX directly assumes $|\mathcal{Q}|$ is finite. Under linear assumptoins, we bound the covering number of $\mathcal{Q}$ in the proof of Theorem 1.
>
> - Results: under the above assumptions, setting $\alpha =\tilde{O}\left(\frac{d_{GEC}(\sqrt{H/T})}{TH^3}\right)$, $B=T\log|A|/H$, with high probability, the regret of Algorithm 1 satisfies
> $$\text{Regret}(T)=\tilde{O}\left(H^{3/2}\sqrt{Td_{GEC}(\sqrt{H/T})}\right).$$
>
> Under linear MDPs, the above regret reduces to (22) in our Thorem 1. Besides, this bound also matches (is slightly tighter than) the bound given in Corollary 5.2 of MEX.
>
> - Proof idea: note that the assumptions we make here (and in MEX) are verified under linear MDPs in our paper in the proof of Theorem 1. We could give the above result following the same logic of the proof of Theorem 1  without the need to prove the parts guaranteed by the new assumptions.
>
> We'll write the extended results as a section in the appendix of our revised paper if we were accepted. Thank you for raising this point.
>
> > **Small edits (line 207,246,49).**
>
> Thank you for your careful review.
>
> - We'll correct the imcomplete sentence in line 207.
>
> - For line 246, we apologize for the unclear phrasing. We meant that while our theoretical analysis is provided for the linear setting, the VAC algorithm itself is compatable with general function approximation, unlike most other works guided by the principle of optimism in the face of uncertainty (see our discussion in Section 1). We will clarify this in the text.
>
> - In line 49, our "optimal value function" refers to the value function derived by acting greedily with respect to the current Q-function estimate (i.e., $\max_a Q_{f,h}(\cdot, a)$), not the true optimal value function of the MDP. We will revise the terminology throughout the paper.
>
> ---
>
> Reference:
>
> [Haarnoja et al., 2018] T Haarnoja et al. Soft actor-critic: Off-policy maximum entropy deep reinforcement learning with a stochastic actorSoft Actor-Critic Algorithms and Applications. ICML 2018.

---

> > ### Author Response · Authors · 2025-08-04
> >
> > Dear Reviewer bBXP,
> >
> > As the author-reviewer discussion period will end soon, we would like to check whether our responses have properly addressed your concerns? If so, could you please kindly consider increasing your initial score accordingly? Certainly, we are more than happy to answer your further questions.
> >
> > Thank you for your time and effort in reviewing our work!
> >
> > Best Regards, Authors

---

> > ### Comment · Reviewer_bBXP · 2025-08-04
> >
> > Thank you to the authors for your clarifications and additional work, which should be incorporated into the final draft. I think the additional detail that the current MEX implementation is an actor-critic algorithm is useful context. I think this work should be accepted and will update my score accordingly.

---

### Official Review · Reviewer_2BfU · 2025-06-30

**Clarity:** 3
**Significance:** 3
**Originality:** 3
**Rating:** 5
**Confidence:** 3

**Summary:**

This paper presents a method for online reinforcement learning called Value-incentivized Actor-Critic (VAC). The paper starts with revisiting the recent Maximize-to-Explore (MEX) framework, demonstrating how it can be derived from a primal-dual Lagrangian perspective. Motivated by this interpretation, they introduce VAC, an actor-critic algorithm optimizing an objective that integrates policy optimization and Q-function estimation without requiring complex bilevel optimization.

Theoretically, the paper provides rigorous guarantees, demonstrating that VAC achieves near-optimal regret of order $O(d H^2 \sqrt{T})$ in the finite-horizon linear Markov decision process (MDP) setting, as well as regret guarantees for the infinite-horizon discounted setting. Here, $T$ is the number of episodes, $H$ is the length of each episode and $d$ is the dimension of the feature map

**Questions:**

- Initially, the dual variables $\lambda_h$ introduced in Equation (10) (and correspondingly the functions $g_h$) are unrestricted (for the sake of intuition). Later, in the transition from Equation (11) to Equation (12), you explicitly constrain these dual functions $g_h$ to belong to the function class $\mathcal{Q}_h$. Could you elaborate on why this particular restriction on the dual variables is appropriate?

- You say that "MEX only admits an actor-critic implementation for $\alpha=0$," which implies that a conceptual actor-critic structure suddenly emerges at $\alpha=0$. However, even at $\alpha=0$, MEX still optimizes only a Q-function without explicitly introducing an actor (policy). Could you clarify explicitly what you mean by "admits actor-critic implementation" at $\alpha=0$? Does this statement imply explicitly adding a separate actor policy component at $\alpha=0$? If not, how does the $\alpha=0$ setting alone introduce an actor-critic structure conceptually?

**Ethical Concerns:**

["NO or VERY MINOR ethics concerns only"]

**Final Justification:**

- The authors' rebuttal clearly addressed my main concerns, particularly regarding intuition for exploration and the main ideas behind Lemmas 3 and 4.
- Although the theoretical regret result does not improve upon the state-of-the-art, I place significant weight on the practical potential and simplicity of the proposed method.

I believe this is a good paper that should be accepted, and I have updated my score accordingly.

**Limitations:**

As far as I concern, the authors have adequately addressed the limitations of their work, and the paper does not have any potential societal impact.

**Quality:**

3

**Strengths And Weaknesses:**

**Strengths:**

- The paper presents a novel algorithm for online RL that does not require bonus computation for exploration nor nested optimization. Thus, it naturally extends to practical function approximation.
- The paper rigorously shows that their algorithm achieves optimal regret up to a factor of $\sqrt{H}$.
- The paper is well-organized, provides proper background, gives clear intuition for the MEX approach, explicitly explains its drawbacks, and shows how the proposed method overcomes these challenges.

**Weaknesses**
- In terms of theoretical guarantees, the paper does not present fundamentally new results. Near-optimal (and better) regret bounds have already been established for stochastic linear MDPs (He et al., 2023). With that being said, I do believe there is value in an algorithm like the proposed one, which shows greater potential for practical use.

- Lemmas 3 and 4, which are central to the analysis, are presented without sufficient intuition or explanation regarding how their results are derived

- At least to me, the paper does not provide sufficient intuition or explanation regarding what explicitly drives exploration in the proposed method

---

> ### Author Rebuttal · Authors · 2025-07-31
>
> ## Response to Reviewer 2BfU
>
> Thank you for your insightful comments and positive feedback! If our clarifications below address your concerns, we'd appreciate your consideration of increasing your score.
>
> > **W1: In terms of theoretical guarantees, the paper does not present fundamentally new results. Near-optimal (and better) regret bounds have already been established for stochastic linear MDPs (He et al., 2023). With that being said, I do believe there is value in an algorithm like the proposed one, which shows greater potential for practical use.**
>
> We fully agree with your assessment. Our primary goal was not to establish a new state-of-the-art regret bound, as near-optimal rates for linear MDPs are indeed known. Instead, our main contribution is the design of the Value-Incentivized Actor-Critic (VAC) framework, which we believe exploration without explicit bonus estimation can be widely applicable to any nonlinear function, therefore, holding significantly greater practical promise than prior methods that achieve similar guarantees.
>
> > **W2: Lemmas 3 and 4, which are central to the analysis, are presented without sufficient intuition or explanation regarding how their results are derived.**
>
> Thank you for raising this point. We'll add the following discussion in our revised paper:
>
> - Lemma 3 bounds the "one-step regret" within the loss function itself: $L_t(f^\star, \tilde\pi^\star) - L_t(f_t, \pi_t)$. The proof strategy is to bound $-L_t(f_t, \pi_t)$ and $L_t(f^\star, \tilde\pi^\star)$ separately. The key tool for proving both parts is Freedman's Inequality (Lemma 5) --- a concentration inequality for martingales. Freedman's Inequality tells us that when summing up a sequence of random numbers where the expectation of the next number, given the past, is zero (this is a martingale difference sequence), the sum is unlikely to be very large. The bound it provides depends on the sum of the variances of these random numbers.
>
>     We construct specific random variables $X$s (Eq.(52)) and $Y$s (Eq.(76)) for bounding $-L_t(f_t, \pi_t)$ and $L_t(f^\star, \tilde\pi^\star)$, resp., and show $-L_t(f_t, \pi_t)$ and $L_t(f^\star, \tilde\pi^\star)$ are sums of $X$s and $Y$s, reps. We then prove $X$s and $Y$s form two martingale sequences, bound their variance, and then apply Freedman's inequality, leading to the results in (75) and (97). Lemma 3 follows from combining these two results.
>
> - Lemma 4 bounds the sum of $V_{f_t}^{\pi_t}(\rho)-V^{\pi_t}(\rho)$ --- the estimation errors of critic $f_t$ for the current policy $\pi_t$, over all episodes $T$.
>
>     - Step 1: Decomposing the value difference into single-step errors (Eq.(99)-(103)). We first rewrite $V_f(\rho)-V^\pi(\rho)$as the sum of the expected one-step Bellman errors $\mathcal{E}_{h}(f,s,a,\pi)$ over $H$ steps.
>
>     - Step 2: using linearity to turn errors into inner products (Eq.(104)-(107)). We leverage the linear MDP assumptions to rewrite the Bellman error $\mathcal{E}_{h}(f,s,a,\pi)$ as the inner product of the feature vector $\phi_h(s,a)$ and a "weight error" vector $W_h(f,\pi)$ that captures how far off our learned parameters are from the "true" parameters that would make the Bellman error zero.
>
>     - Step 3: bounding the sum of inner products.The above reformulation converts our problem to a classic problem in online learning and linear bandits, and allows us to bound $\sum_{t=1}^T |V_{f_t}^{\pi_t}(\rho)-V^{\pi_t}(\rho)|$ with the classic self-normalizing technique [Abbasi-Yadkori et al., 2011].
>
>
> > **W3: the paper does not provide sufficient intuition or explanation regarding what explicitly drives exploration in the proposed method.**
>
> Thank you for your comment. The term $V_f^\pi(\rho)$ in our objective (17) -- $\sup_{f\in \mathcal{Q}, \pi\in\mathcal{P}} \{V_f^\pi(\rho)-\alpha L_t(f,\pi)\}$ -- incentivizes exploration. This exploration mechanism is conceptually inherited from the reward-biasing framework [Kumar and Becker, 1982]. Below is an intuitive explanation on it.
>
> - **The exploration driver (the value-incentivized term) $V_f^\pi(\rho)$:** This term, which we aim to maximize, encourages the agent to find a joint policy-critic pair $(\pi,f)$ that promises the highest possible expected return. This is the "optimism" principle at the heart of our method.
>
> - **The exploitation driver (the loss term) $L_t(f, \pi)$:** $L_t(f, \pi)$ is the Bellman consistency error calculated on the data collected so far. It penalizes any Q-function $f$ that is inconsistent with the observed transitions and rewards. It acts as a "reality check", pulling the Q-function towards values that are grounded in the data.
>
> - **The interplay between these two terms.** The Bellman error $L_t(f, \pi)$ doesn't put much supervision on under-explored state-action pairs. This gives our algorithm freedom: to maximize the overall objective, the critic ($f$) tends to hypothesize an optimistically high Q-value for under-explored state-action pairs, which incentivizes the actor ($\pi$) to increase the probability of selecting under-explored state-action pairs in order to maximize the overall expected value $V_f^\pi$, leading to exploration. If the optimistic guess was wrong, the newly collected data will create a large Bellman error in the next iteration, and the $L_t$ term will pull the Q-value back down to reality. If the guess was right, the high value is confirmed, and the agent has successfully found a better policy.
>
> We'll add this discussion to our revised paper. Thank you for your feedback.
>
> >**Q1: regarding restriction on the dual variables from (11) to (12)**
>
> Thank you for your insightful question.
>
> - To answer your question, our interpretation begins with the MEX objective. From Eq.(9) we can directly compute that the optimal $\lambda_h^\star$ is $$\frac{Q_h(s,a)- (r_h(s,a)+E_{s'\sim P_h}[\max_{a'} Q_{f,h+1}(s',a')])}{\beta}.$$ Comparing this and (10), we immediately know that the corresponding optimal $g^\star_h$ is $$g^\star_h(s,a)=r_h(s,a)+E_{s'}[\max_{a'} Q_{f,h+1}(s',a')],$$ the Bellamn optimality target. This indicates that the approximation error incurred by restricting $g_h\in\mathcal{Q}_h$ is fundamentally linked to the inherent Bellman optimality error of the function class $\mathcal{Q}_h$. Particularly, MEX assumes $\{\mathcal{Q}_h\}$ satisfies Bellman completeness in their Assumption 3.1, which guarantees the $g^\star_h\in\mathcal{Q}_h$, and this approximation error becomes 0.
>
> - Moreover, the same logic extends directly to our proposed VAC method. Following an identical derivation for the Lagrangian in Eq. (14), the optimal dual function in our case becomes the Bellman consistency target for the policy $\pi$, where the max operator is replaced by an expectation: $$g_h^\star(s,a)=r_h(s,a)+E_{s'\sim P_h, a'\sim\pi_{h+1}}[Q_{f,h+1}(s',a')].$$
> Therefore, the parameterization for $g_h$ is also highly reasonable for VAC because it ties the quality of the dual approximation to a core property of the function class: its ability to represent Bellman updates. Especially, under our assumptions, $\mathcal{Q}_h$ satisfies Bellman completeness (see our Lemma 10), and this approximation error is guaranteed to be 0.
>
> We'll add the above discussion to our updated paper. Thank you for your comment.
>
> >**regarding the meaning of "MEX only admits an actor-critic implementation for $\alpha=0$"**
>
> Thank you for this very insightful question and for your careful reading of our paper. We agree that our statement in lines 222-224 is confusing and inaccurate.
> We appreciate the opportunity to clarify our meaning and correct the text.
>
> - Our intention with the $\alpha=0$ case was based on a mathematical equivalence. When $\alpha=0$, the MEX objective becomes $\sup_f [E_{s\sim\rho} \max_a Q_{f,1}(s,a)]$, which is equivalent to $\sup_f \sup_{\pi} [E_{s\sim\rho} Q_{f,1}^\pi(s,a)]$. This admits an actor-critic implementation, and the actor is the greedy policy.
>
> - We found our original characterization of $\alpha=0$ was incorrect. When $\alpha=0$, the Bellman consistency loss term Eq.(7) is completely ignored, which encourages finding a Q-function with the highest possible values, unconstrained by data. This corresponds to pure optimistic exploration, not "no exploration". We'll replace line 222-224 with the following revised text:
>
>     >*In contrast, MEX does not admit an actor-critic implementation for any $\alpha>0$ since their data loss term requires the optimal value function, while the data loss term $L_t(f, \pi)$ is policy-dependent in VAC.*
>
> Finally, we also want to stress the implementation challenges of MEX that motivate our work. The core difficulty lies in the bilevel optimization structure of the MEX objective (Eq.(7)), which may be intractable for both solving $\max_a Q(s,a)$ and computing the gradients. MEX acknowledges the difficulty, and their practical sidesteps its own theoretical objective by resorting to an actor-critic framework.
>
> ---
>
> References:
>
> [Abbasi-Yadkori et al., 2011] Abbasi-Yadkori, Y., Pál, D., & Szepesvári, C. Improved algorithms for linear stochastic bandits. NeurIPS, 2011.
>
> [Kumar and Becker, 1982]  Kumar and A. Becker. A new family of optimal adaptive controllers for markov chains. IEEE Transactions on Automatic Control, 1982.

---

> > ### Comment · Reviewer_2BfU · 2025-08-04
> >
> > Thank you for your detailed clarifications. Your response adequately addresses my main concerns. I believe this is a good paper that should be accepted, and I will update my score accordingly.

---

> > > ### Author Response · Authors · 2025-08-04
> > >
> > > Dear Reviewer 2BfU,
> > >
> > > Thank you very much for your thoughtful review and supportive decision!
> > >
> > > Best Regards, Authors

---

### Official Review · Reviewer_rUfp · 2025-07-01

**Clarity:** 3
**Significance:** 3
**Originality:** 3
**Rating:** 5
**Confidence:** 3

**Summary:**

The paper introduces VAC, a primal–dual actor-critic algorithm for online RL that merges exploration and exploitation into a single differentiable objective. It enforces data consistency and rewards high-value policies, achieving provably near-optimal regret in linear MDPs, while offering scalability to general function approximators

**Questions:**

Major:
1. The robustness of the algorithm to the violation of assumptions. In particular, quality of linear approximation.
2. Bellman consistency regularization is interesting but results in bi-level. If the regularizer is not useful in the first place, the entire work you did would not be so useful at the end. Please discuss benefits of the formulation that you simplied.

Minor:
1. In 1st paragraph of Sec 3: If f is a Q, what is Q_f?
2: In Eq (10): Is this parameterization always possible?
3. In Eq (10): what is \delta_h(s,a)? I guess that is r()_maxQ in Eq (9).
4. a possible typo: exploration-> exploitation in line 168

**Ethical Concerns:**

["NO or VERY MINOR ethics concerns only"]

**Final Justification:**

Although the readability of the paper could have been improved, I like the dual perspective on the problem. After the rebuttal, I would like to keep the same rating as is.

**Limitations:**

1. Works only with linear MDP, in which the quality of approximation can be arbitrarily bad with some features.
2. Empirical validation is completely missing.

**Quality:**

3

**Strengths And Weaknesses:**

Strengths
1. Avoids bi-level optimization via primal-dual approach
2. Exploration as part of optimization via primal dual perspective; critic's value function serves as a dual variable incentivizing the actor's exploration.
3. Theoretical results with sublinear regret bounds


Weaknesses
1. Limited to linear MDP and learning good features can be challenging. This is particularly concerning as the same features are used for linear approximation of reward and dynamics. I can take the reward but dynamics model has a high complexity. Finding the same features for reward and dynamics all together can be more challenging than just for dynamics.

Minor: Abstract can be much improved (Introduction can be too): the first half of abstract is too general (you used valuable space for something not essential)

---

> ### Author Rebuttal · Authors · 2025-07-31
>
> ## Response to Reviewer rUfp
>
> >**W1: regarding feature learning of linear MDP**
>
> Thank you for your comment. While we agree that the assumption of a joint linear representation for both rewards and dynamics (Assumption 1) is a significant one, we would like to offer a few points of clarification:
>
> - **validity and flexibility:** In the tabular setting where the state and action spaces are finite, the assumption holds as we could set the features $\phi_h(s, a)$ to be one-hot vectors,  demonstrating that our framework is a valid extension of a fundamental RL scenario. More discussion on the linear MDP examples can be found in [Jen et al., 2020].
> Note that despite being linear, the Markov transition model $P_h(·|s, a)$ can still have infinite degrees of freedom as the measure $\mu_h$ in (20) is unknown.
>
> - **Standard assumption for theoretical analysis:** As we remark around Assumption 1 in our paper, this is a standard and widely-used assumption in the theoretical analysis of online RL with function approximation, as established in seminal works like [Jin et al., 2020].
>
> - **Extension to general FA:** We want to highlight that our framework is not fundamentally limited to linear MDPs. Our analysis can be readily extended to general function approximation settings, similar to the approach in MEX [Liu et al., 2024]. By replacing the linear MDP assumptions with more general conditions—namely, realizability and Bellman completeness of the Q-function class, and a bound on the Generalized Eluder Coefficient (GEC)—our VAC algorithm achieves a similar $\tilde{O}(\sqrt{T})$ regret. More details about this can be found in our response to W4/Q3 of Reviewer bBXP.
>
> We'll add more discussions to our revised paper. Thank you for raising this point.
>
> >**W2 (minor): abstract and introduction can be improved**
>
> Thank you for your valuable feedback. We will revise the abstract and introduction to be more direct in our updated paper.
>
> >**Q1-major: The robustness of the algorithm to the violation of assumptions. In particular, quality of linear approximation.**
>
> - Our analysis can be extended to handle the misspecification scenario, where the true rewards and dynamics are only approximately linear. The key is to analyze how this approximation error propagates through our algorithm and affects the final regret following a similar idea as in [Xie et al.,2021] or [Yuan et al.,2023].
>
> - As we mentioned in our previous response, Our analysis is readily extended to the general function approximation setting under similar assumptions made in MEX. See also our response to W4/Q3 of Reviewer bBXP for more details on this.
>
> >**Q2-major: Bellman consistency regularization results in bi-level...discuss benefits of the formulation that you simplified.**
>
> Thank you for this insightful question. Below is our clarification on the motivation and advantages of our approach.
>
> - **The "Bellman consistency regularization" is a principled data-fitting loss.** First, we'd like to clarify the role of the term in question. While our derivation in Section 3 frames it as a penalty term in a Lagrangian, in the final objective (Eq. (17)), it functions as a data-fitting loss term, not an exploration regularizer. This loss term ensures that the learned Q-function $f$ remains consistent with the observed environment dynamics. The actual term that incentivizes exploration is the value-incentivizing term $V^\pi_f(\rho)$ in Eq.(17).
>
> - **The benefit of our simplification: from intractable bilevel to tractable saddle-point optimization.** We want to stress that this loss is actually standard and commonly used in the literature, see  [Xie et al.,2021], [Dai et al., 2018] for example. Notably, unlike MEX's the bi-level loss function (EQ.(7)), where $\max_a Q(s,a)$ is inside the square and the maximum needs to me computed for all states in the dataset, which could be intractable, our loss $L_t(f,\pi)$ in (18) leads to a tractable saddle-point problem. To see this, we could write $L_t(f,\pi)$ as $\sup_{g\in \mathcal{Q}} \tilde L_t(f,\pi,g)$ for some function $\tilde L_t$. Then our optimization objective (17) becomes
> $$\sup_{f,\pi}\inf_{g}\{V_f^\pi(\rho)-\alpha \tilde{L}_t(f,\pi,g)\}.$$
> This problem is well-studied in the RL literature. For example, Algorithm 1 in [Dai et al., 2018] gives a feasible mirror descent type of method to solve this.
>
> We'll add more discussion on this in our revised paper. Thank you for raising this.
>
> >**Q1-minor: In 1st paragraph of Sec 3: If $f$ is a $Q$, what is $Q_f$? 2: In Eq (10): Is this parameterization always possible?**
>
> - As defined in lines 163, $Q_f=\{Q_{f,h}\}_{h\in[H]}$ with $Q_{f,h} = f_h$, i.e., $Q_f=f$.
>
> - Yes, the reparameterization in Eq (10) is always possible. This is a constructive reparameterization trick (following [Baird, 1995] and [Dai et al., 2018], as cited on line 185). For any $\lambda_h$, one can define a corresponding $g_h$ via $g_h(s,a) = Q_{f,h}(s,a) - \beta\lambda_h(s,a)$.
>
> >**Q2-minor: In Eq (10): what is $\delta_h(s,a)$? I guess that is r+maxQ in Eq (9).**
>
> There's no $\delta_h$ in (10). We guess you mean $\delta_h$ apprearing in 187-189, where we specify (11) holds for any $\delta_h(s,a)$, and be setting $\delta_h(s,a)=r_h(s,a)+\max_{a}Q_{f,h+1}(s',a)$ in (11), we obtain (12).
>
> >**Q3-minor: a possible typo: exploration-> exploitation in line 168.**
>
> Thank you for your careful reviewing. Here "exploration" is indeed the correct term, and we see how this point connects to your insightful question in Q2-major about the role of the objective's components. Specifically, line 168 describes the two distinct parts of the MEX objective in Eq. (6):
>
> - The first term, $\sup_{f\in Q} E_{s_1\sim \rho} [\max_a Q_{f,1}(s_1,a)]$, promotes exploration. By optimizing over the function f to find an optimistically high Q-function, the objective incentivizes visiting states and actions that could have high value. This is the core principle of "Maximize to Explore."
>
> - The second term, $-\alpha L_t(f)$, is the data-fitting loss derived from the regularization term of the Lagrangian (9). Its role is to ensure the optimistic Q-function remains consistent with the observed data transitions, as discussed in our response to Q2-major.
>
> >**Limitation 1: works only with linear MDP, in which the quality of approximation can be arbitrarily bad with some features.**
>
> Thank you for raising this point. We agree the performance of any method using linear function approximation is fundamentally tied to the quality of the chosen features. Meanwhile, we want to emphasize that the VAC algorithm itself is not restricted to linear models, and our algorithm is compatible with general function approximation. Furthermore, as we mention in our previous responses, we can relax the linear MDP assumptions and obtain a $\tilde{O}(\sqrt{T})$ regret bound for general function approximation setting under similar assumptions as MEX. We plan to write this extension as a section in the appendix in our revised paper. Thank you for your comment.
>
> >**Limitation 2: Empirical validation is completely missing.**
>
> We conducted experiments on two difficult MuJoCo continuous control tasks: Walker2d and Ant. We benchmark VAC against the state-of-the-art Soft Actor-Critic (SAC) algorithm [Haarnoja et al., 2018]. The average return and standard deviation of the algorithms over 3 random seeds after 1000,000 training steps are summarized in the following table:
>
> | env / alg | SAC | VAC (ours, $\alpha=1000$) | VAC (ours, $\alpha=500$) |
> |---|---|---|---|
> | Ant-v4 | 3163.86 $\pm$ 268.90 | 3974.44 $\pm$ 390.95 | **5134.55** $\pm$ 424.42 |
> | Walker2D-v4 | 4394.68 $\pm$ 181.42 | **4609.53** $\pm$ 333.98 | 4434.75 $\pm$ 117.34 |
>
> The results demonstrate VAC outperforms the strong SAC baseline.
>
> - **Implementation details**: we use a SAC style implementation with a different Q-learning objective. To be specific:
>
>     - We perform alternating single gradient (Adam) steps to update actor (policy) and critic (Q-function) by solving an approximation version of our objective (17). Same as the implementation of SAC and MEX (see their Section 7.2), we drop the second term in the expression of $L_t(f,\pi)$ (c.f. Eq.(18)) and use a standard TD error. And we use $1/n\sum_{i=1}^n Q_{f,1}(s,a_i)$ ($a_i\sim\pi(\cdot|s)$) to approximate the value-incentivized regularization term $V_{f}^\pi(s)$ (in our experiment, we set $n=1$, which is good enough).
>
>     - Non-linear FA: we use standard MLPs to parameterize both actor and critic, with 2 hidden layers of 256 neurons each, using the ReLU activation.
>
> - **Small theory-to-practice gap:** We'd like to highlight that VAC's theoretical formulation natrually yields an actor-critic algorithm, leading to a very small theory-to-practice gap. This can be contrasted with prior work like MEX, where the practical actor-critic implementation used in their experiments (see their Section 7.2) differs significantly from their proposed theoretical bi-level objective.
>
> ---
>
> [Jin et al., 2020] C. Jin, Z. Yang, Z. Wang, and M. I. Jordan. Provably efficient reinforcement learning with linear function approximation. In Conference on learning theory, PMLR 2020.
>
> [Liu et al., 2024] Z. Liu, M. Lu, W. Xiong, H. Zhong, H. Hu, S. Zhang, S. Zheng, Z. Yang, and Z. Wang. Maximize to explore:
> One objective function fusing estimation, planning, and exploration. NeurIPS 2024.
>
> [Xie et al.,2021] T Xie, CA Cheng, N Jiang, P Mineiro, A Agarwal. Bellman-consistent Pessimism for Offline Reinforcement Learning, NeurIPS 2021.
>
> [Yuan et al.,2023] R Yuan, SS Du, RM Gower, A Lazaric, L Xiao. Linear Convergence of Natural Policy Gradient Methods with Log-Linear Policies.

---

> > ### Comment · Reviewer_rUfp · 2025-08-03
> >
> > I appreciate the additional computations using MuJoCo, which show a good performance over SAC.
> >
> > While I am still concerned about the linear model with dynamics, this is not a new concern and hence I would like to keep my original rating as is.

---

### Official Review · Reviewer_1Y1J · 2025-07-04

**Clarity:** 3
**Significance:** 3
**Originality:** 3
**Rating:** 5
**Confidence:** 3

**Summary:**

The paper revisits optimistic exploration in online RL through a primal-dual lens and proposes a Value-Incentivized Actor-Critic (VAC) algorithm. VAC merges exploration and exploitation in a single, first-order-optimizable objective that favors high-value policies while staying consistent with data. It sidesteps the bilevel optimization hurdle faced by Maximize-to-Explore. Under linear MDPs, VAC achieves near-optimal sub-linear regret in both finite- and infinite-horizon cases.

**Questions:**

Question 1: I am still not really clear how the proposed method improves the bound in MEX from the order of each variable (e.g., d, H, and T)? Or could the authors provide more intuition for the improvement?

**Ethical Concerns:**

["NO or VERY MINOR ethics concerns only"]

**Paper Formatting Concerns:**

No conern

**Quality:**

3

**Strengths And Weaknesses:**

Strength 1: The theoretical results are sound and solid. The presentation is good to follow and clear to understand.

Strength 2:  It is innovative to incorporate the MEX idea to the actor-critic framework and help readers understand how to incentivize exploration from the primal-dual perspective.

Strength 3: The derived sample complexity bound in this paper is nearly optimal and tight, which also matches the results in the other paper.

Weakness 1: From my understanding, one advantage of MEX is to use a single objective for the optimization to trade-off between exploration and exploitation in RL, which is implementation-friendly in the deep RL framework, which is also verified by experiments on Mujoco tasks. However, this paper does not study the empirical performance on the proposed method, which limits the potential of the board application of the proposed method empirically.

---

> ### Author Rebuttal · Authors · 2025-07-31
>
> ## Response to Reviewer 1Y1J
>
> Thank you for your positive evaluation and insightful questions! Below we address your main concern.
>
> >**Weakness 1: From my understanding, one advantage of MEX is to use a single objective for the optimization to trade-off between exploration and exploitation in RL, which is implementation-friendly in the deep RL framework, which is also verified by experiments on Mujoco tasks. However, this paper does not study the empirical performance on the proposed method, which limits the potential of the board application of the proposed method empirically.**
>
> Thank you for your valuable feedback.
> We conducted experiments on two difficult MuJoCo continuous control tasks: Walker2d and Ant. We benchmark VAC against the state-of-the-art Soft Actor-Critic (SAC) algorithm [Haarnoja et al., 2018]. The average return and standard deviation of the algorithms over 3 random seeds after 1000,000 training steps are summarized in the following table:
>
> | env / alg | SAC | VAC (ours, $\alpha=1000$) | VAC (ours, $\alpha=500$) |
> |---|---|---|---|
> | Ant-v4 | 3163.86 $\pm$ 268.90 | 3974.44 $\pm$ 390.95 | **5134.55** $\pm$ 424.42 |
> | Walker2D-v4 | 4394.68 $\pm$ 181.42 | **4609.53** $\pm$ 333.98 | 4434.75 $\pm$ 117.34 |
>
> The results demonstrate VAC outperforms the strong SAC baseline.
>
> - **Implementation details**: we use a SAC style implementation with a different Q-learning objective. To be specific:
>
>     - We perform alternating single gradient (Adam) steps to update actor (policy) and critic (Q-function) by solving an approximation version of our objective (17). Same as the implementation of SAC and MEX (see their Section 7.2), we drop the second term in the expression of $L_t(f,\pi)$ (c.f. Eq.(18)) and use a standard TD error. And we use $1/n\sum_{i=1}^n Q_{f,1}(s,a_i)$ ($a_i\sim\pi(\cdot|s)$) to approximate the value-incentivized regularization term $V_{f}^\pi(s)$ (in our experiment, we set $n=1$, which is good enough).
>
>     - Non-linear FA: we use standard MLPs to parameterize both actor and critic, with 2 hidden layers of 256 neurons each, using the ReLU activation.
>
> - **Small theory-to-practice gap:** We'd like to highlight that VAC's theoretical formulation naturally yields an actor-critic algorithm, leading to a very small theory-to-practice gap. This can be contrasted with MEX, where the practical actor-critic implementation used in their experiments (see their Section 7.2) differs significantly from their proposed theoretical bi-level objective which could not be implemented end-to-end in deep RL frameworks for continuous environments in MuJoCo. Instead of using their proposed algorithm, they resort to an actor-critic framework in the experiments that completely circumvent the bilevel optimization.
>
> >**Question 1: I am still not really clear how the proposed method improves the bound in MEX from the order of each variable (e.g., d, H, and T)? Or could the authors provide more intuition for the improvement?**
>
> Thank you for your question. As stated in our introduction, our contribution is not improving the regret bound of MEX, but to circumvent the intractable bi-level optimization problem and propose a provable sample-efficient algorithm that's easy-to-implement.
>
> - **VAC and MEX have the same near-optimal regret bound.** MEX doesn't explicitly provide a regret bound for the linear MDP setting, and assumes a finite function class. Adapting their Corollary 5.2 to linear MDPs, and through a covering number argument on the function class, their regret can be shown to have the same order as ours ($\tilde{O}(dH^2\sqrt{T})$) (in their Corollary 5.2, $B_l=O(H)$, $d_{GEC}(1/\sqrt{HK})=\tilde{O}(Hd)$, and $\log(|\mathcal{H}|)=\tilde{O}(Hd)$). Both our regret is *near-optimal* up to a factor of $\sqrt{H}$ compared to the minimax lower bound $\tilde{\Omega}(d\sqrt{H^3T})$, as is stated in line 244-246 of our paper.
>
> - **MEX's bilevel optimization objective is hard/intractable to solve.** We want to stress that MEX objective is hard/intractable to solve, and this motivates us to design an easy-to-implement actor-critic algorithm.
>
>     - **The inner maximization $\max_{a} Q(s,a)$ can be intractable itself.** The MEX objective in Eq.(7) requires computing max_{a}Q_{f,h+1}(s_{h+1},a') for every transition $(s_h,a_h,s_{h+1})$ in the dataset at every gradient step. This is a significant challenge in many settings:
>     - Large discrete action spaces: In domains like large language models, the action space (e.g., all possible tokens or sentences) is enormous. An exhaustive search for the maximum is computationally infeasible.
>     - Continuous action spaces: If $Q(s,\cdot)$ is non-convex, finding the global maximum is an NP-hard problem. Even with convex Q-functions, this requires running a separate optimization procedure for every data point in the batch, which is computationally expensive.
>     - **The bilevel optimization problem is intractable to solve.**  To minimize the objective $L_t(f)$ given in our Eq.(7) through gradient-based methods, we need to compute $\nabla_\theta [\max_{a} Q(s,a;\theta)]$, which could be intractable and could not be handled by standard toolkits. Note that simply computing $a^\star:=\arg\max_{a} Q(s,a)$ and computing $\nabla_\theta Q(s,a^\star;\theta)$ can lead to wrong gradients of the Q-function, because $\nabla_\theta [\max_{a} Q(s,a;\theta)]$ does not always equal to $\nabla_\theta  Q(s,a^\star;\theta)$, where $a^\star:=\arg\max_a\nabla_\theta  Q(s,a;\theta)$. A simple example is when
>     $$Q(a;\theta)=\begin{cases} a,\quad\text{when $a\leq \theta$},\\
>     0, \quad\text{when $a> \theta$}. \end{cases}$$
> Then when $\theta>0$, $\nabla_\theta [\max_{a} Q(s,a;\theta)]=1$, but $\nabla_\theta  Q(s,a^\star;\theta)=0$.
>
>    - **The original MEX Implementation sidesteps this problem.** Crucially, the authors of MEX themselves acknowledge this difficulty in their practical implementation, and circumvent the bi-level optimization entirely. Instead, they resort to an actor-critic framework, creating a significant disconnect between their theoretically analyzed objective and their practical implementation.
>
> In contrast to MEX, our VAC algorithm Value-Incentivized Actor-Critic (VAC) algorithm elegantly resolves these practical challenges, leading to a method that is both theoretically sound and straightforward to implement.
>
> ---
>
> Reference:
>
> [Haarnoja et al., 2018] T Haarnoja et al. Soft actor-critic: Off-policy maximum entropy deep reinforcement learning with a stochastic actorSoft Actor-Critic Algorithms and Applications. ICML 2018.

---

> > ### Comment · Reviewer_1Y1J · 2025-08-01
> > **Reply**
> >
> > Thanks for the reply, which addresses my concerns. Hence, I will keep my opinion for acceptence.

---

### Decision · Program_Chairs · 2025-09-17

**Decision:**

Accept (poster)

**Comment:**

The authors study a setting that is standard in RL theory (linear MDPs). Although their regret bounds exhibit no improvement over existing ones (in fact, statistically optimal algorithms for this setting are known), they achieve so with an algorithm that has a primal-dual structure (this allows to draw connections with practical actor-critic algorithms) *and* is computationally efficient. The reviewers agree on the value of this contribution. The reviewers' suggestions and the new experiments should be included in the final version.